# Precise modulation of BRG1 levels reveals features of mSWI/SNF dosage sensitivity

Yota Hagihara [1,2,3,7], Chao Zhang [1,2,3,6,7] & Yi Zhang [1,2,3,4,5] ✉

Mammalian switch/sucrose nonfermentable (mSWI/SNF) complex regulates chromatin accessibility and frequently shows alterations due to mutation in cancer and neurological diseases. Inadequate expression of mSWI/SNF in heterozygous mice can lead to developmental defects, indicating dosage-sensitive effects of mSWI/SNF. However, how its dosage affects function has remained unclear. Using a targeted protein degradation system, we investigated its dosage-sensitive effects by precisely controlling protein levels of BRG1, the ATPase subunit of the mSWI/SNF complex. We found that binding of BRG1 to chromatin exhibited a linear response to the BRG1 protein level. Although chromatin accessibility at most promoters and insulators was largely unaffected by BRG1 depletion, 44% of enhancers, including 84% of defined superenhancers, showed reduced accessibility. Notably, half of the BRG1-regulated enhancers, particularly superenhancers, exhibited a buffered response to BRG1 loss. Consistently, transcription exhibited a predominantly buffered response to changes in BRG1 levels. Collectively, our findings demonstrate a genomic feature-specific response to BRG1 dosage, shedding light on the dosage-sensitive effects of mSWI/SNF complex defects in cancer and other diseases.

Packaging of DNA into chromatin restricts the access of transcription factors (TFs) to their cognate binding sites[1–3]. ATP-dependent chromatin remodelers facilitate TF binding by sliding, disassembling or ejecting nucleosomes, thereby generating open chromatin at regulatory elements such as promoters, enhancers and insulators[1–4]. In mammals, chromatin remodelers are categorized into four major families based on their ATPase subunits: SWI/SNF (also known as BRG1/BRM-associated factor (BAF)); imitation switch (ISWI); chromodomain helicase DNA-binding (CHD); and INO80 (ref. 1). Although these remodelers can co-occupy genomic regions, each chromatin remodeler also selectively remodels particular genomic loci and affects the binding of specific TFs[5–7]. The regulatory elements, which are detected by transposases and nucleases, are generally considered to be stable once opened[8,9]. However, recent studies using chemical inhibitors or

acute degradation have shown that chromatin remodelers are continuously required to maintain chromatin accessibility[10–12], revealing the dynamic nature of nucleosome positioning. These approaches overcome the limitations of conventional genetic depletion methods, in which remodelers persist for days and are subject to multiple layers of feedback regulation. As a result, they enable more direct investigation of the primary roles of chromatin remodelers[10–17].

The mammalian SWI/SNF (mSWI/SNF) complex is a multisubunit assembly derived from 29 genes[18]. It exists in three distinct forms, canonical BAF (cBAF), polybromo-associated BAF (PBAF) and non-canonical BAF (ncBAF), which share two mutually exclusive ATPase subunits, BRG1 (encoded by *SMARCA4*) and BRM (*SMARCA2*). Copy number variations of remodelers, including mSWI/SNF, are poorly tolerated in humans[19,20]. Moreover, mutations in genes encoding mSWI/

[1]Howard Hughes Medical Institute, Boston Children's Hospital, Boston, MA, USA. [2]Program in Cellular and Molecular Medicine, Boston Children's Hospital, Boston, MA, USA. [3]Division of Hematology/Oncology, Department of Pediatrics, Boston Children's Hospital, Boston, MA, USA. [4]Department of Genetics, Harvard Medical School, Boston, MA, USA. [5]Harvard Stem Cell Institute, Boston, MA, USA. [6]Present address: State Key Laboratory of Genetic Evolution & Animal Models, Kunming Institute of Zoology, Chinese Academy of Sciences, Kunming, China. [7]These authors contributed equally: Yota Hagihara, Chao Zhang. ✉e-mail: yzhang@genetics.med.harvard.edu

SNF subunits occur in more than 20% of cancers[21–23] and often destabilize the complex[24,25]. Notably, heterozygous mutations have also been linked to neurodevelopmental syndromes such as Coffin–Siris and Nicolaides–Baraitser[26–28], indicating sensitivity to the dosage of mSWI/SNF[20,29–32]. Consistently, *Brg1*-heterozygous mice exhibit exencephaly, cardiac developmental defects and increased susceptibility to mammary tumors[33–36]. Similar dosage-sensitive effects have also been observed in *Caenorhabditis elegans*, with functional differences in cell migration and growth depending on SWI/SNF dosage[37,38]. Collectively, these results suggest a conserved dosage-sensitive function of mSWI/SNF. However, the molecular basis of this function remains poorly understood.

In this study, we took advantage of the dTAG system[39,40], which enables precise control of protein dosage[39,40], to investigate BRG1 dosage sensitivity in mouse embryonic stem (mES) cells. By modulating BRG1 levels, we found that BRG1 binding was linearly sensitive to its dosage, regardless of TFs or histone modifications, suggesting that the genome-wide distribution of mSWI/SNF is independent of these factors. Chromatin accessibility at half of BRG1-bound enhancers and at small subsets of promoters and insulators depended on BRG1. At BRG1-bound but BRG1-independent promoters and insulator, NFYA and CTCF may contribute to maintenance of accessibility. Among BRG1-dependent enhancers, approximately half exhibited a buffered response to BRG1 dosage. Notably, superenhancers (SEs) exhibited a buffered pattern, whereas weak enhancers were more sensitive to reduction in BRG1 levels. Although BRG1 degradation had a limited impact on transcription, downregulated genes were predominantly buffered genes. We further confirmed the conservation of BRG1-dosage-sensitive accessibility regulation and its role in *MYC* expression in human lung epithelial cells. Overall, our results reveal the genomic features of BRG1 dosage sensitivity, providing insight into its dosage-sensitive defects in cancers and other diseases.

## Results

### BRG1 binding exhibits a linear response to its dosage

In mES cells, *Brg1* encodes the major catalytic subunit of mSWI/SNF, as *Brm* is only weakly expressed[12,41] (Extended Data Fig. 1a). To manipulate BRG1 levels, we used the dTAG system[39,40,42] to perform targeted degradation of the BRG1 protein by introducing an *FKBP12–F36V* degron into the *Brg1* locus, resulting in *FKBP12–F36V Brg1*-knock-in cells. In the presence of heterobifunctional molecule dTAG13, the degron directs the fusion protein to the proteasome for degradation[39] (Extended Data Fig. 1b,c), a process that is completed within 60 min (Extended Data Fig. 1d). Addition of the degron tag did not affect the ability of BRG1 to form complexes with other mSWI/SNF components, including ARID1A (cBAF), PBRM1 (PBAF) and BRD9 (ncBAF), as evidenced by our finding that the immunoprecipitation efficiency of *Brg*1-knock-in cells was comparable to that of the untagged wild-type cells (Extended Data Fig. 1e). As a previous study found that mSWI/SNF inhibition for 24 h mirrored the effects of genetic manipulation[12], we treated cells with dTAG13 for 24 h to modulate BRG1 levels in a dose-dependent manner

(Fig. 1a,b). Immunostaining confirmed a population-wide decrease in BRG1 levels (Extended Data Fig. 1f). These results demonstrated that we had successfully established a system that allowed precise control of BRG1 dosage for study of its dosage-sensitive function.

To this end, we first investigated the genome-wide binding profile of BRG1 by performing a cleavage under targets and release using nuclease (CUT&RUN) assay using *Brg1*-knock-in cells (Extended Data Fig. 1g). BRG1 binding was enriched at promoters (23.9%, marked by histone H3 trimethylated at Lys4 (H3K4me3)), enhancers (65.0%, marked by H3 monomethylated at Lys4 and H3 acetylated at Lys27 (H3K27ac)) and insulators (11.1%, marked by CTCF) (Fig. 1c,d), consistent with its known roles at both proximal and distal regulatory elements[6,16,43]. The presence of BRG1 at insulators was consistent with its reported interaction with CTCF and ncBAF complex localization patterns[43–45]. These results confirm that the tagged BRG1 mirrors the endogenous binding profile and primarily targets promoters, enhancers and insulators.

The nature of the mechanisms underlying the target selectivity of remodelers remains a fundamental question. TFs and histone modifications have been reported to recruit mSWI/SNF, likely through physical interactions[46–50]. Given that TFs exhibit nonlinear binding profiles and preferentially occupy high-affinity targets at low concentrations[40,51–53], we hypothesized that the binding profile of BRG1 would follow that of TFs, assuming that its recruitment was determined by physical interactions with TFs. In this case, high-affinity and low-affinity binding sites would exhibit differential dosage sensitivity. To test this hypothesis, we profiled BRG1 binding across five dTAG13 concentrations, corresponding to different BRG1 levels (Fig. 1a,b). Numbers of BRG1-binding sites were reduced by almost half following 0.3 nM dTAG13 treatment (78% of protein remaining), and further depletion nearly abolished detectable peaks (10 nM dTAG13, 12% of protein remaining) (Fig. 1b,e). Principal component analysis (PCA) confirmed a global shift in BRG1 binding as dTAG13 concentrations increased (Fig. 1f). Overall, BRG1-binding signals decreased in a gradual, dose-dependent manner (Fig. 1g).

Next, we investigated whether BRG1 binding at different types of regulatory element (promoters, enhancers and insulators) would exhibit distinct dosage sensitivity. Notably, all three classes showed similar gradual reductions in binding (Fig. 1h). To further analyze the kinetics of BRG1 binding, we generated two hypothetical models: a linear response model (with a proportional relationship between BRG1 dosage and binding signals); and a buffered response model (showing limited changes in binding with a subtle decrease in BRG1 levels) (Fig. 1i). We then calculated the delta values between the distances of observed signals and those of the two models. A positive delta value indicated proximity to the linear model, whereas a negative delta value indicated closeness to the buffered model (Fig. 1j and Methods). Of all BRG1-binding peaks, 92.2% followed the linear response model (Fig. 1k). As previous studies have implicated OCT4 and H3K27ac in BRG1 recruitment[48–50], we assessed BRG1 binding at co-occupied sites. Both sites cobound by OCT4 and BRG1 and those co-occupied by H3K27ac and BRG1 also exhibited linear responses to BRG1 dosage

**Fig. 1 | Chromatin binding of BRG1 exhibits a linear response to its dosage. a**, Western blot analysis of BRG1 in *Brg1*-knock-in (KI) and wild-type (WT) mES cells after treatment with the indicated concentrations of dTAG13 for 24 h. Data are representative of six independent experiments. **b**, Quantification of relative BRG1 abundance following dTAG13 treatment. BRG1 expression was normalized to that of tubulin and then to that of DMSO-treated (0 nM) *Brg1*-KI mES cells within each biological replicate. Data represent the mean ± s.e.m. from six independent experiments. **c**, Heatmap showing profiles of protein and histone modifications (CUT&RUN and ChIP–seq) for regions associated with different genomic elements (bottom). The top panel shows aggregate coverage plots with mean enrichment at promoters (red), enhancers (cyan) and insulators (orange). **d**, Representative genome browser views of BRG1 binding at a promoter, enhancer and insulator. **e**, Numbers of BRG1 binding peaks identified at different dTAG13 concentrations. **f**, PCA of BRG1 CUT&RUN data across varying dTAG13 concentrations. **g**, Aggregate coverage plot showing mean BRG1 enrichment at different dTAG13 concentrations. **h**, Aggregate coverage plot of BRG1 enrichment at promoters, enhancers and insulators across different dTAG13 concentrations. **i**, Diagram illustrating models of linear (black) and buffered (green) dosage-sensitive responses. **j**, BRG1 signals at representative linear (red) and sensitive (blue) regions (left). The relative signal was calculated by normalizing with 0 nM signals. Genome browser view showing BRG1 binding at regions with linear and buffered responses (right). The genome locations of the browser view are indicated at the bottom of the panels. The linear and buffered models are illustrated in the diagram. **k**, Histogram of delta values for all 20,326 BRG1-binding sites. The red line indicates delta = 0. Enh., enhancer; Ins., insulator; Pro., promoter.

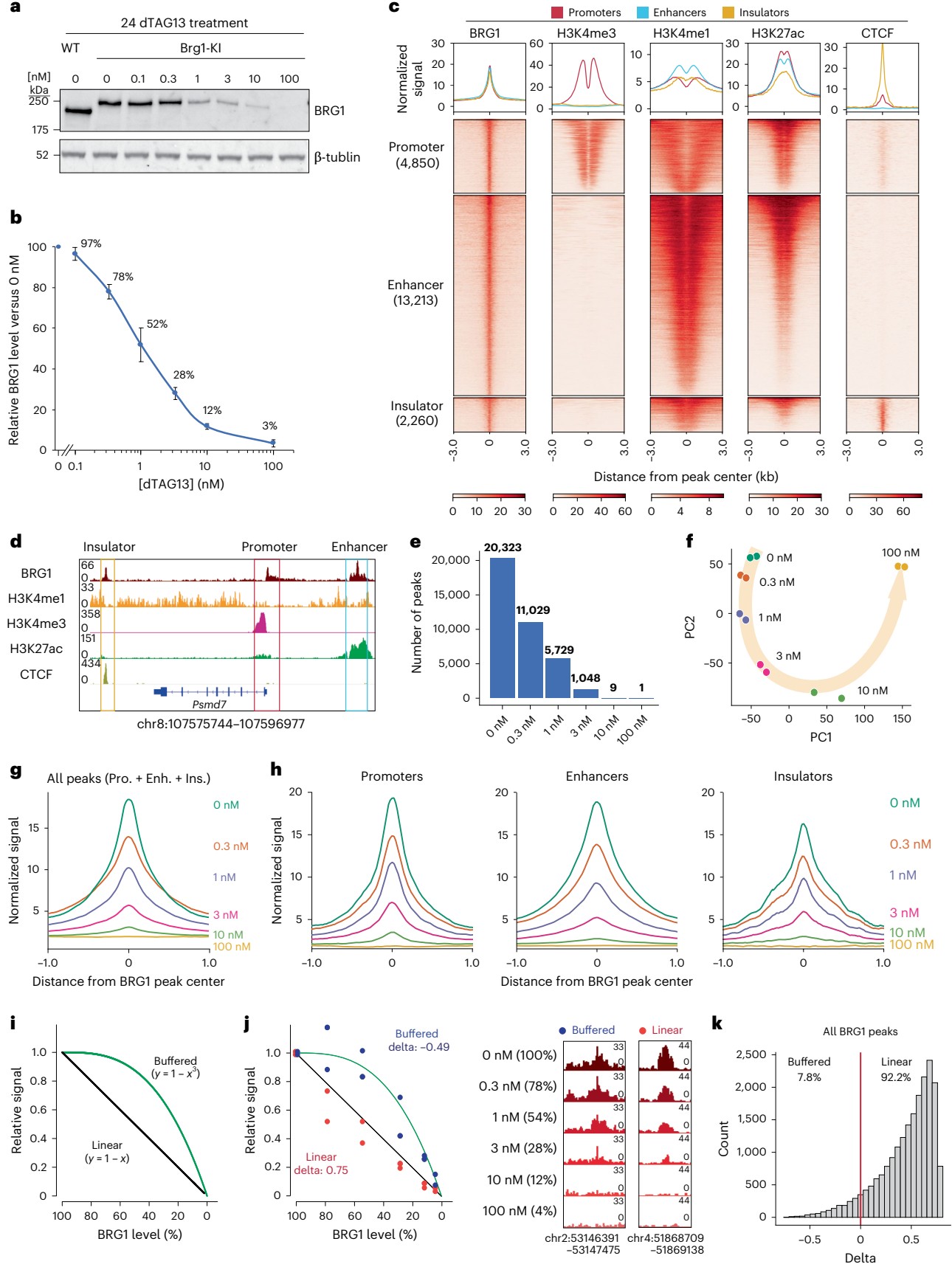

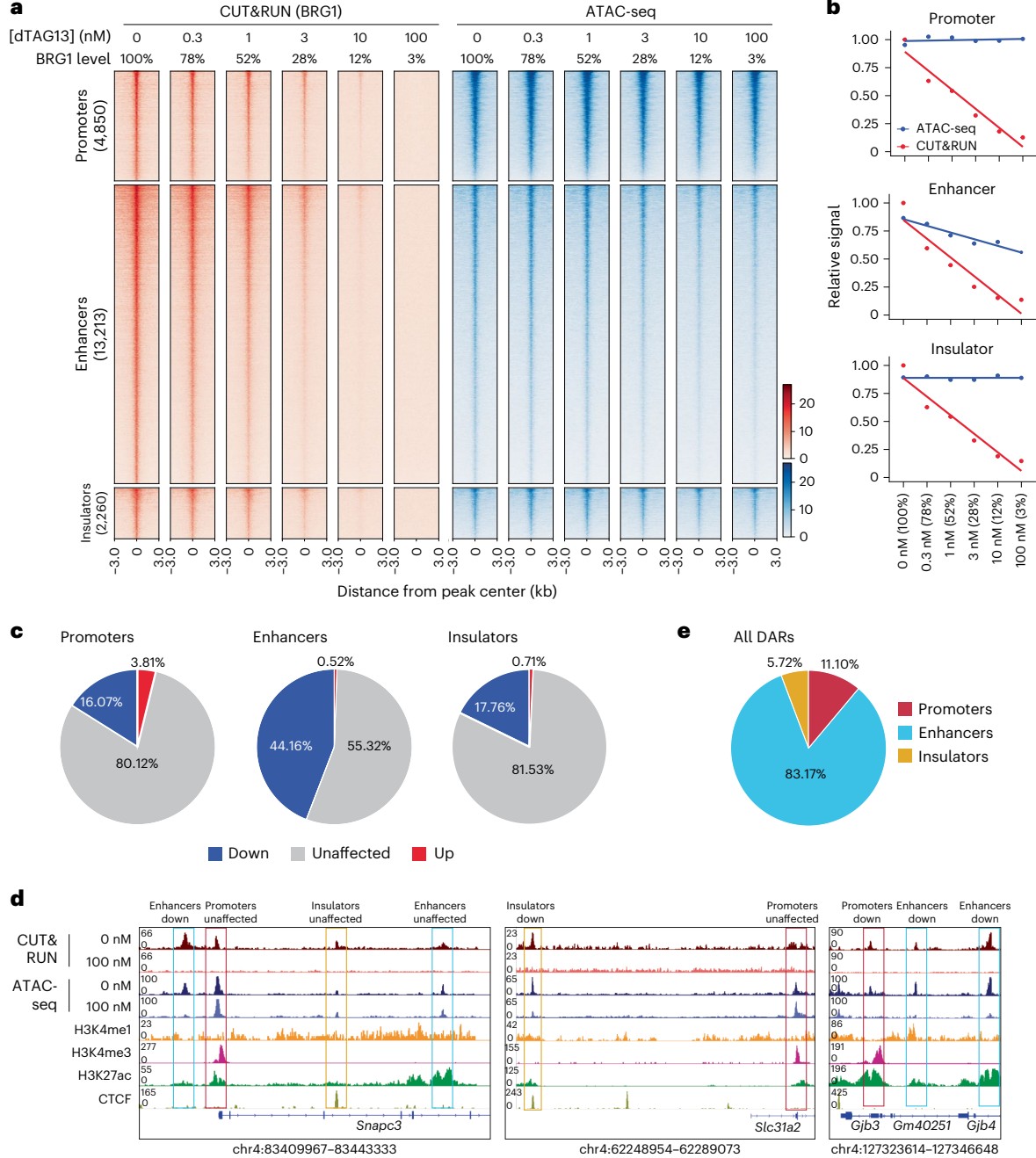

**Fig. 2 | Chromatin accessibilities at different regulatory elements exhibit varying dependencies on BRG1. a**, Heatmap showing BRG1 binding (red) and ATAC-seq (blue) for regions falling into genomic elements at the indicated dTAG13 concentration. **b**, Line plots depicting relative BRG1 binding (red) and ATAC-seq (blue) at promoters, enhancers and insulators. **c**, Pie chart depicting the percentages of significant changes in ATAC-seq signals at BRG1 target sites in promoters, enhancers and insulators. **d**, Genome browser view showing BRG1 binding and ATAC-seq signals across regions containing promoters, enhancers and insulators. **e**, Pie charts depicting the relative distributions of significant changes in ATAC-seq signals at promoters, enhancers and insulators.

(Extended Data Fig. 1h), suggesting that BRG1 binding at these loci does not heavily rely on recruitment by TFs or histone modifications.

## Variable BRG1 dependency in accessibility among regulatory elements

Given that the major function of chromatin remodeler is to regulate chromatin accessibility, we next assessed the effects of BRG1 depletion on chromatin accessibility using transposase-accessible chromatin with sequencing (ATAC-seq) at different BRG1 levels, focusing on BRG1-binding sites. Numbers of differentially accessible regions (DARs) increased progressively with decreasing BRG1 levels (Extended

Data Fig. 2a), with most DARs showing reduction in or loss of accessibility (Extended Data Fig. 2b). Notably, whereas chromatin accessibility at enhancers declined gradually in response to BRG1 depletion, it remained largely unchanged at promoters and insulators (Fig. 2a,b). Specifically, 44.2% of BRG1-bound enhancers exhibited decreased accessibility, compared to only 16.1% of promoters and 17.8% of insulators (Fig. 2c,d). Of all DARs, 83.2% were located at enhancers, whereas only 11.1% and 5.7% were found at promoters and insulators, respectively (Fig. 2e). A previous *Brg1*-knockout study showed a similar trend, with enhancers affected more than promoters and insulators[48] (Extended Data Fig. 2c). Collectively, these results indicate that BRG1 has a more

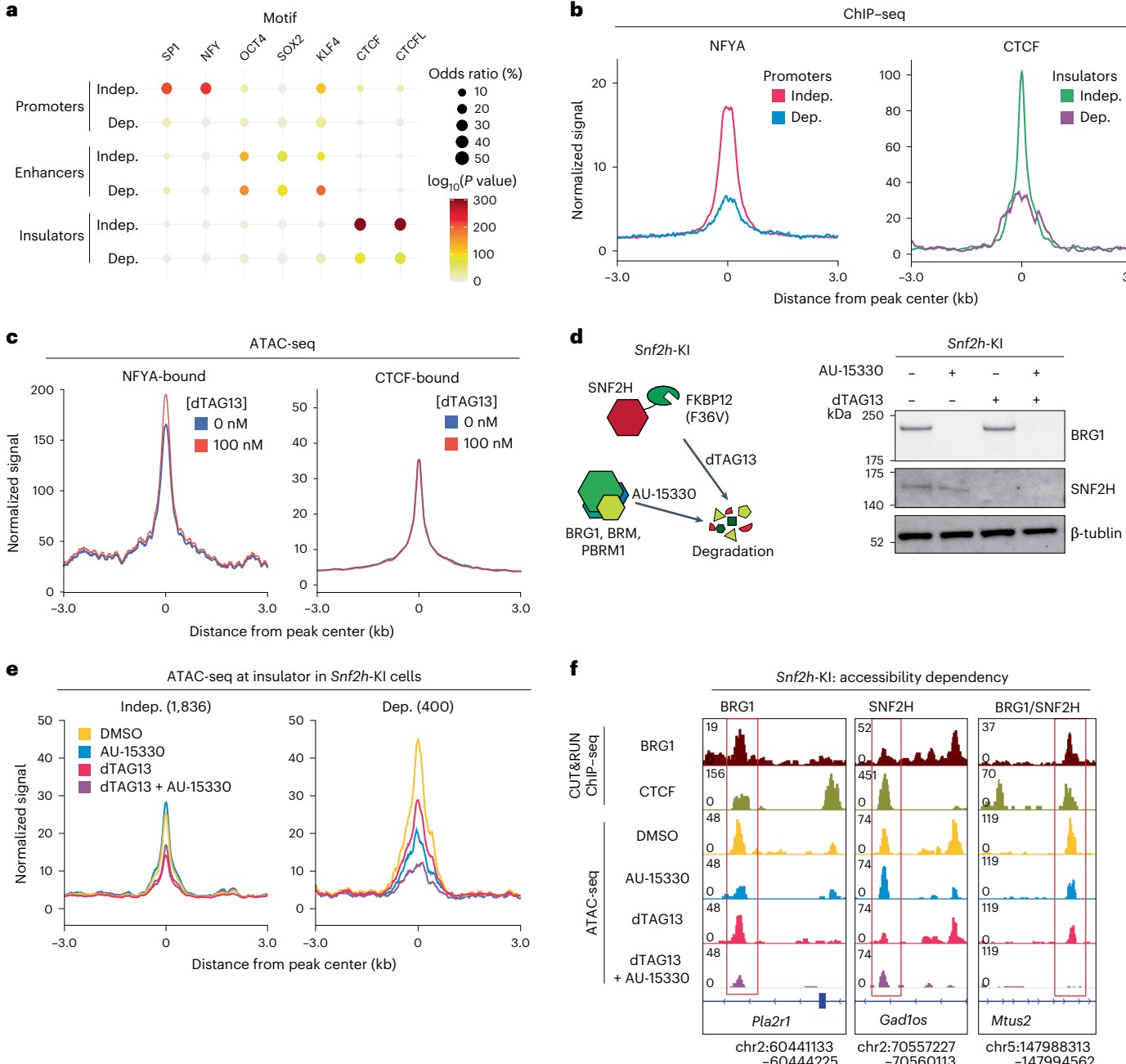

**Fig. 3 | BRG1 and SNF2H cooperatively regulate insulator elements. a**, TF motif enrichment at BRG1-independent (Indep.) and BRG1-dependent (Dep.) ATAC-seq peaks in promoters, enhancers and insulators. The color scale represents the −log$_{10}$(P value), and the dot size corresponds to the effect size measured as an odds ratio. Two-sided Fisher's exact test was performed. **b**, Aggregate coverage plot showing mean NFYA and CTCF ChIP−seq signals at BRG1-dependent and BRG1-independent regulatory elements. **c**, Aggregate coverage plot showing mean ATAC-seq signals at NFYA and CTCF-bound regions. **d**, Schematic of SNF2H and BRG1 degradation system (left). SNF2H is fused with FKBP12−F36V (Snf2h-KI) to enable degradation by dTAG13. BRG1, BRM and PBRM1 are degraded by AU-15330. Western blot analysis of BRG1 and SNF2H in Snf2h-KI mES cells after treatment with 1 μM AU-15330 and/or 100 nM dTAG13 for 24 h. Data are representative of three independent experiments. **e**, Aggregate coverage plot of mean ATAC-seq peaks at BRG1-dependent and BRG1-independent insulators in mES cells after 24 h of AU-15330 and/or dTAG13 treatment. **f**, Representative genome browser view showing regions regulated by BRG1 and SNF2H and those coregulated by BRG1 and SNF2H (BRG1/SNF2H).

important role in maintaining the accessibility of enhancers than that of promoters and insulators.

## Compensatory mechanisms maintain accessibility in the absence of BRG1

Next, we explored the factors that might determine BRG1 dependency. To this end, we first compared BRG1-binding signals and found that their levels alone did not distinguish BRG1-dependent and BRG1-independent regions (Extended Data Fig. 3a). We next performed motif enrichment analyses to explore whether TFs contributed to the maintenance of chromatin accessibility in the absence of BRG1. The NFY and SP1 binding motifs were significantly enriched at independent promoters, whereas CTCF and CTCFL motifs were enriched at independent insulators (Fig. 3a). By contrast, no specific motifs were uniquely enriched at independent enhancers; for instance, the OCT4 motif was identified in both dependent and independent enhancers

(Fig. 3a). Reanalysis of public chromatin immunoprecipitation followed by high-throughput sequencing (ChIP–seq) datasets revealed that NFYA and CTCF preferentially bound to BRG1-independent promoters and insulators, respectively (Fig. 3b and Extended Data Fig. 3b). Conversely, OCT4 was more enriched at dependent enhancers (Extended Data Fig. 3b,c). Chromatin accessibility at OCT4-binding sites was decreased following BRG1 depletion, consistent with previous results demonstrating that BRG1 is necessary for maintenance of the accessibility of OCT4-binding sites in mES cells[48] (Extended Data Fig. 3d).

The distribution of the Polycomb repressive complex (PRC) is directed by mSWI/SNF[14]. As well as enrichment of PRC1 and PRC2 components (RING1B and SUZ12, respectively) at promoters, we found increased binding of these components at BRG1-dependent promoters following BRG1 loss (Extended Data Fig. 3e). Although Polycomb proteins themselves do not directly affect chromatin accessibility[54,55], these results indicate that BRG1 depletion triggers broader chromatin reorganization at BRG1-dependent regulatory elements.

On the other hand, chromatin accessibility at NFYA-binding and CTCF-binding sites remained unchanged after BRG1 depletion (Fig. 3c), suggesting that these factors contribute to the maintenance of chromatin accessibility at BRG1-independent regions. Consistently, reanalysis of *Nfya*-knockdown datasets showed that chromatin accessibility was specifically decreased at NFYA-bound BRG1-independent promoters[56] (Extended Data Fig. 3f). These results, together with the observation that NFYA was mainly enriched at the BRG1-independent promoters, suggest that NFYA plays a critical part in maintaining open chromatin states at BRG1-independent promoters.

CTCF can be recruited by SNF2H, the ATPase subunit of the ISWI complex[7,57–59]. As SNF2H and BRG1 have overlapping binding profiles[5], we hypothesized that SNF2H might compensate for BRG1 at insulators. To test this hypothesis, we established *Snf2h*-dTAG mES cells, in which SNF2H could be degraded by dTAG13 and BRG1 could be degraded by AU-15330, a proteolysis-targeting chimera (PROTAC) that targets both BRG1 and BRM[60] (Fig. 3d). Cells were treated for 24 h, consistent with our previous experiments. As expected, AU-15330 alone had a minor effect on BRG1-independent insulators. By contrast, SNF2H depletion alone led to reduced accessibility at both BRG1-dependent and BRG1-independent insulators, with a greater effect observed at BRG1-independent insulators (Fig. 3e,f). Simultaneous depletion of SNF2H and BRG1 further decreased accessibility at BRG1-dependent insulators but not at independent insulators (Fig. 3e,f and Extended Data Fig. 3g). BRG1-dependent insulators showed slightly higher enrichment of TFs in motif and ChIP–seq analyses, suggesting that BRG1-dependent insulators were occupied by more regulatory factors than BRG1-independent insulators (Extended Data Fig. 3b,h). Overall, these results indicate that the SNF2H–CTCF axis has a dominant role in regulation of accessibility at BRG1-independent insulators, whereas both SNF2H and BRG1 contribute to the accessibility of BRG1-dependent insulators.

### Enhancer activity defines dosage sensitivity to BRG1

Next, we characterized the dosage response properties of BRG1-dependent regulatory elements with respect to chromatin accessibility. To investigate the kinetics of chromatin accessibility following BRG1 depletion, we applied the linear and buffered response models described above (Fig. 1i and Methods). Overall, BRG1-dependent regulatory elements displayed an approximately equal distribution between linear and buffered responses (50.7% linear versus 49.3% buffered) (Fig. 4a, first panel). Whereas enhancers showed slight preference for a linear response (Fig. 4a, third panel), both promoters and insulators exhibited a clear preference for a buffered response (Fig. 4a, second and fourth panels). This indicates that lower levels of BRG1 are enough to maintain accessibility at promoters and insulators, but higher levels are required for maintaining enhancer accessibility.

As the major function of BRG1 is to modulate enhancer accessibility, we further explored the dosage sensitivity of BRG1-dependent enhancers by dividing them into five groups based on their sensitivity: group 1 (G1) represented the most buffered enhancers and group 5 (G5) the most linearly sensitive enhancers (Fig. 4b). When BRG1 levels were depleted to half their original value, G1 enhancers retained 80% of their original accessibility, whereas G5 enhancers retained less than 40% (Extended Data Fig. 4a). The BRG1 binding kinetics showed similar dosage sensitivity in the different groups (Fig. 4b and Extended Data Fig. 4b). As shown in representative regions, whereas BRG1 binding predominantly followed a linear model (Fig. 4c, red dots), enhancer accessibility exhibited different linear and buffered responses (Figs. 1k and 4c, blue dots), indicating that regulation of chromatin accessibility is not totally dependent on BRG1 binding.

Next, we examined the regulatory landscape in each BRG1-dependent enhancer group using ChromHMM. We found that dosage sensitivity was correlated with enhancer activity; buffered-response enhancers (G1) were more enriched for strong enhancers compared to linear-response enhancers (G5) (Fig. 4d). The temporal dynamics of accessibility after mSWI/SNF inhibition showed no substantial differences among groups[16] (Extended Data Fig. 4c), indicating that dosage sensitivity cannot be explained by differences in the speed of the response to mSWI/SNF inhibition. We then analyzed the enrichment of histone modifications and transcriptional regulators across the five groups. Although TFs including NANOG were found in all groups, G1 enhancers exhibited higher levels of H3K27ac and acetyltransferase p300 (Fig. 4e and Extended Data Fig. 4d), both of which are markers of active enhancers[61]. These features suggest that G1 has features of SEs, which are characterized by broader chromatin domains and higher levels of H3K27ac and MED1 (ref. 62). Consistently, MED1 was specifically enriched at G1 compared to other groups (Fig. 4e). Among the 231 SEs defined in mES cells[62,63], 97 (42%) were classified as G1 (Fig. 4f,g). Furthermore, 195 (84.4%) of SEs were among the BRG1-dependent enhancers (G1 to G5), highlighting the critical role of BRG1 in regulation of SE accessibility. Collectively, these results show that the dosage sensitivity of enhancers is more closely correlated with enhancer activity than with specific TFs. SEs, in particular, tend to exhibit buffered responses to BRG1 dosage.

### ChromBPNet reveals sequence basis for BRG1 dosage sensitivity

To gain deeper insights into the sequence features underlying dosage sensitivity, we trained ChromBPNet on ATAC-seq data from control (0 nM dTAG13) and BRG1-depleted (100 nM dTAG13) conditions to predict chromatin accessibility[64] (Extended Data Fig. 5a). The model achieved high accuracy, with a Pearson's correlation of $r = 0.84$ across BRG1-bound peaks (Extended Data Fig. 5b).

Next, we applied the TF-MODISCO motif discovery algorithm to identify motifs with high contribution scores[64,65]. Motifs of SP1 and MAZ were enriched in buffered enhancers, whereas OCT4, SOX2 and KLF4 were enriched in linear enhancers (Extended Data Fig. 5c). Marginal footprinting analysis confirmed preferential binding of OCT4 and SOX2 in linear enhancers (Extended Data Fig. 5d). Notably, buffered enhancers exhibited significantly higher GC content than linear enhancers (Extended Data Fig. 5e), suggesting that sequence composition may shape enhancer properties and influence dosage sensitivity.

We then hypothesized that motif abundance within enhancers would contribute to dosage sensitivity. We quantified the numbers of high-score sites[66] as putative TF binding sites. G1 enhancers exhibited the highest number of high-score sites. BRG1 depletion led to a reduction in the number of high-score sites; however, G1 enhancers retained more sites than G5 enhancers (Extended Data Fig. 5f,g). These results suggest that motif composition and abundance within enhancers are critical for their response to BRG1 dosage.

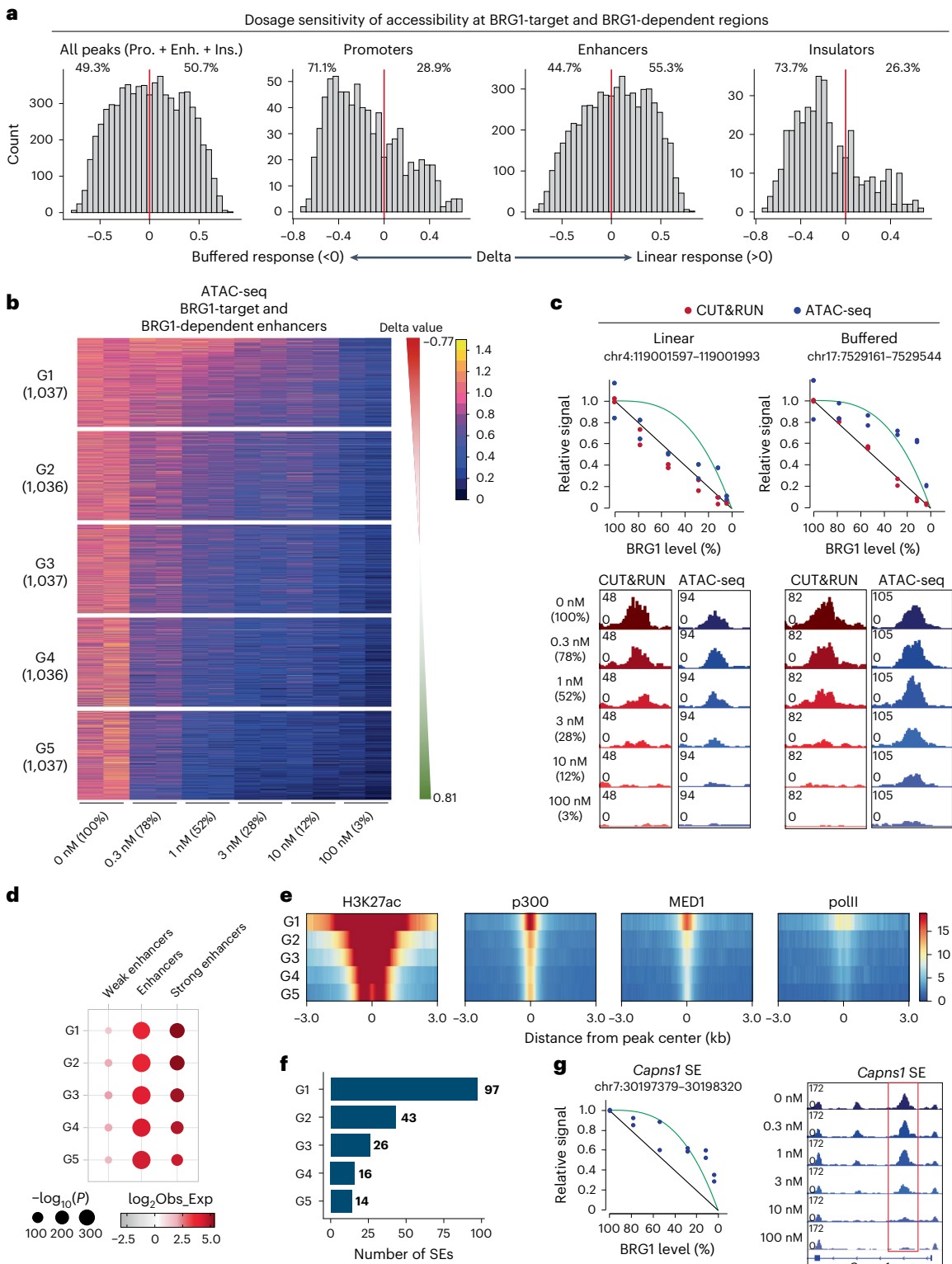

**Fig. 4 | BRG1 dosage sensitivity of enhancer accessibility is associated with enhancer activity. a**, Histograms of delta values for BRG1-dependent regulatory elements, including all elements, promoters, enhancers and insulators. The red line indicates delta = 0. **b**, Heatmap showing BRG1-dependent enhancer regulatory elements clustered into five groups based on delta values, ranging from buffered to linear response regions. **c**, Representative genome browser views showing linear and buffered responses to BRG1 dosage. Red dots represent CUT&RUN signals under different dTAG13 concentrations, and blue dots represent ATAC-seq signals (top) under different dTAG13 concentrations. Genome browser view showing BRG1 binding at regions with linear and buffered responses (bottom). The linear and buffered models are illustrated

in the diagram. **d**, Enrichment analysis of different element types per group, with enrichment calculated against all consensus regions. Dot size represents −log$_{10}$(P value), and the color scale corresponds to the effect size measured as an odds ratio. Two-sided Fisher's exact test was used to compare enrichment to background regions. Obs_Exp represents observed divided by expected frequencies. **e**, Heatmap showing relative enrichment of chromatin marks and TF binding across different groups of ATAC-seq peaks. **f**, Number of SEs in each group. **g**, Genome browser view showing ATAC-seq signals at BRG1-bound *Capns1* SE at different dTAG13 concentrations. The linear (black) and buffered (green) models are illustrated in the diagram.

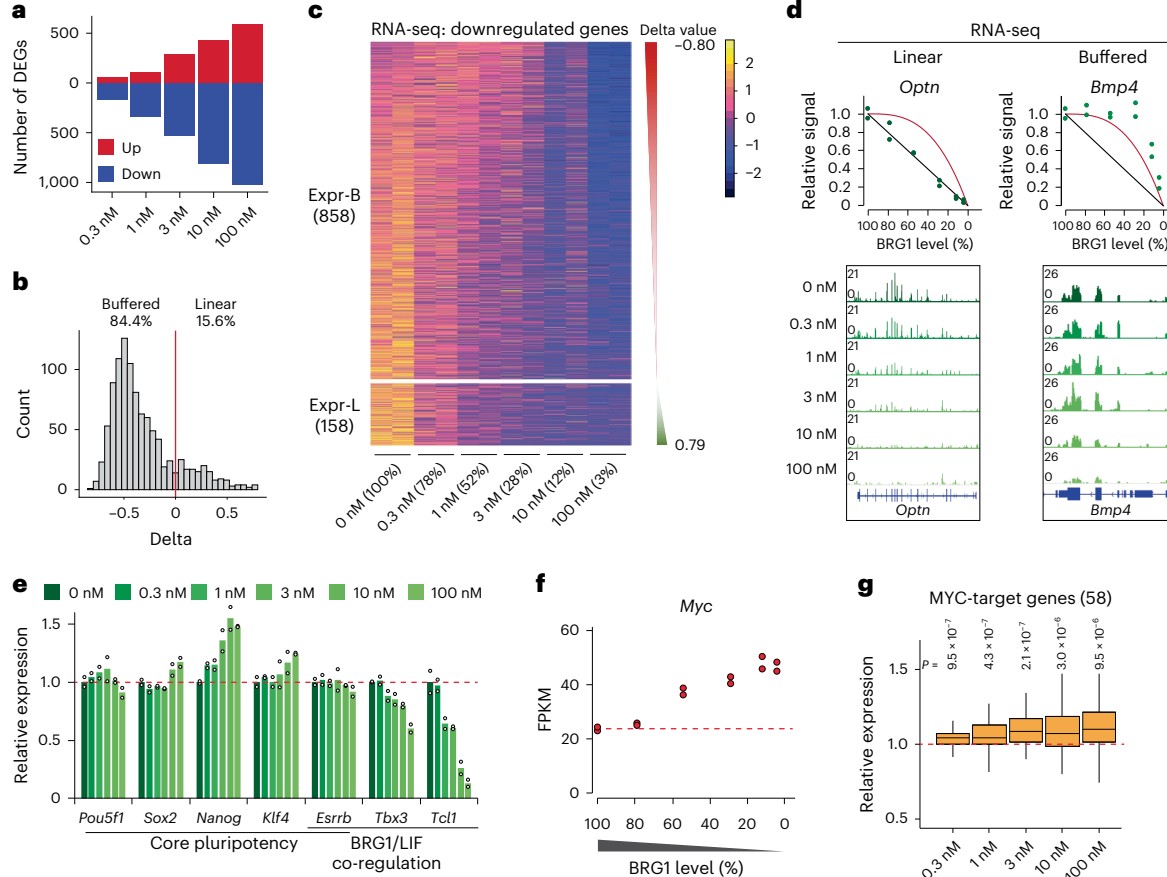

**Fig. 5 | Transcription of the majority of genes exhibits buffered response to BRG1 dosage. a**, Numbers of DEGs at different dTAG13 concentrations (fold change >1.5). **b**, Histogram of delta values for all 1,016 DEGs. The red line indicates delta = 0. **c**, Heatmap showing BRG1-dependent genes, clustered into two groups based on delta values: negative or buffered (Expr-B) and positive or linear (Expr-L). **d**, Representative genes whose expression exhibited linear or buffered responses to BRG1 depletion. Green dots represent mean expression levels. Diagrams illustrating the linear (black) and buffered (red) models are shown. **e**, Relative gene expression of pluripotency factors at different BRG1 levels. Expression levels are normalized to the expression at 0 nM dTAG13

(DMSO), and the red dashed line represents the mean level at 0 nM. Data are from two independent experiments. **f**, *Myc* expression at different BRG1 levels. Data are from two independent experiments. **g**, Box plot showing expression changes of MYC-target genes under different concentrations of dTAG13. Boxes indicate the 25th to 75th percentiles, the median is shown as a horizontal black line, and whiskers extend up to 1.5 times the interquartile range from the quartiles. Expression levels are normalized to the expression at 0 nM (DMSO). Data represent two independent experiments, and *P* values were calculated with Wilcoxon's *t*-test (two-sided, versus 0 nM).

## Dosage-sensitive transcriptional regulation by BRG1

As chromatin accessibility changes can affect transcription, we next examined how varying BRG1 levels influenced gene expression. Transcriptome analysis revealed a gradual increase in numbers of differentially expressed genes (DEGs) with decreasing BRG1 levels (Fig. 5a and Extended Data Fig. 6a). Despite this trend, the overall number of DEGs was relatively low, with 589 (5.2%) upregulated and 1,016 (9.0%) downregulated genes among the 11,303 expressed genes (fragments per kilobase of transcript per million mapped reads (FPKM) > 1). Genes with BRG1-dependent promoters were downregulated with decreased BRG1 dosage, whereas genes with BRG1-independent promoters showed no major change (Extended Data Fig. 6b). Enhancer-proximal genes followed a similar pattern, although with milder effects; only BRG1-dependent enhancer target genes showed modest but significant reductions in expression (Extended Data Fig. 6c). Moreover, genes associated with linearly sensitive enhancers (G5) exhibited greater effects than those linked to buffered enhancers (G1) (Extended Data Fig. 6d). These results indicate that promoter regulation exerts a stronger transcriptional impact than enhancer regulation.

To further dissect transcriptional responses, we modeled all downregulated genes into buffered or linear response categories

(Fig. 5b,c). Notably, the majority of the downregulated genes (84.4%) exhibited buffered responses, with limited numbers of genes (15.6%) showing linear responses to BRG1 dosage (Fig. 5b). Buffered response genes retained more than 75% expression at half BRG1 levels, whereas expression of linear response genes dropped below 50% (Fig. 5d and Extended Data Fig. 6e). Downregulated genes were enriched for terms related to proliferation and the MAPK pathway, implying compromised self-renewal programs in mES cells (Extended Data Fig. 6f). Whereas core pluripotency genes including *Oct4* (also known as *Pou5f1*) remained unchanged, we observed downregulation of *Tbx3* and *Tcl1*, targets of BRG1 and LIF-STAT3 (refs. 67–69; Fig. 5e), linking BRG1 dosage to maintenance of mES cell identity.

To assess the phenotypic consequences, we cultured mES cells long-term in varying concentrations of dTAG13. Proliferation defects became noticeable at 10 nM dTAG13, with severe impairments at 100 nM[41] (Extended Data Fig. 7a). Alkaline phosphatase staining, a hallmark of undifferentiated states, revealed decreased naive mES cell populations at 1 nM, with higher concentrations resulting in loss of pluripotency (Extended Data Fig. 7b). These results indicate that maintenance of pluripotency is more sensitive to BRG1 dosage than proliferation. We also observed a BRG1-dosage-sensitive inverse

correlation in expression of oncogene *Myc* and Polycomb-target *Hox* genes (Fig. 5f and Extended Data Fig. 7c), consistent with their dysregulation upon BRG1 perturbation[14,55,70]. Notably, *Myc* levels were significantly increased at dTAG13 concentrations of 1 nM or higher, accompanied by activation of MYC-target genes (Fig. 5g), reinforcing *Myc* as a dosage-sensitive target of BRG1.

## BRG1 dosage-sensitive regulation is conserved in human cells
To determine whether the BRG1 dosage-sensitive features observed in mES cells were conserved in other cell types, we investigated BRG1 dosage sensitivity in the BEAS-2B cell line, a normal human bronchial epithelial cell line widely used in lung carcinogenesis studies[71–73]. Approximately 35% of lung cancers, including non-small-cell lung cancers, harbor mutations in genes encoding mSWI/SNF subunits[22,74]. Conditional knockout of *Brg1* and/or *Brm* in mouse lungs leads to cancer development and progression[75], suggesting that dosage of mSWI/SNF components is critical for lung cell homeostasis. Unlike mES cells, BEAS-2B cells express both BRG1 and BRM (Extended Data Fig. 8a), which exhibit cooperative and antagonistic functions[76]. To simplify the BRG1 dosage analysis, we generated *BRM*-knockout BESA-2B cells (Extended Data Fig. 8b). Knockout of *BRM* did not alter cell morphology (Extended Data Fig. 8c), BRG1 binding (Extended Data Fig. 8d) or chromatin accessibility (Extended Data Fig. 9c; comparison of ATAC-seq results for wild-type versus *BRM*-knockout, 0 nM), indicating that the remaining mSWI/SNF complexes containing only BRG1 could maintain cell and chromatin states similar to those of wild-type cells.

We depleted BRG1 using AU-15330, a PROTAC targeting BRG1 and BRM[60] that enables rapid BRG1 degradation within 2 h (Extended Data Fig. 8e), and observed dosage-dependent degradation after the treatment (Fig. 6a). Population-wide BRG1 depletion was confirmed by immunostaining (Extended Data Fig. 8f,g). Consistent with a previous report[60], we observed degradation of PBAF-specific subunit PBRM1 (Extended Data Fig. 8h). These results indicate that AU-15330 allows precise control of BRG1 levels, similar to the dTAG system.

We then investigated changes in chromatin accessibility in response to varying BRG1 levels. As in mES cells, numbers of DARs progressively increased with BRG1 degradation levels (Extended Data Fig. 9a), with most DARs showing reduced accessibility (Extended Data Fig. 9b). PCA revealed that wild-type and *BRM*-knockout cells were clustered together under both control (0 nM AU-15330) and depletion (1,000 nM AU-15330) conditions, with a global shift correlated with AU-15330 concentration (Fig. 6b). BRG1 depletion predominantly affected enhancer accessibility (Fig. 6c), with more than 90% of BRG1-bound enhancers showing reduced accessibility (Fig. 6d,e). By contrast, only about half of the BRG1-bound promoters and insulators were affected (Fig. 6d,e). These differences among regulatory elements were consistent with our observations in mES cells. However, the numbers and proportions of affected regions were higher across all regulatory elements in BEAS-2B cells (Figs. 2c and 6d), suggesting a greater dependence on mSWI/SNF for chromatin accessibility in this cell type.

To understand the role of BRG1 in chromatin regulation, we compared BRG1-dependent and BRG1-independent regulatory elements (Extended Data Fig. 9c,d). BRG1-independent promoters exhibited greater H3K4me3 enrichment, a hallmark of strong promoters (Extended Data Fig. 9c). Motif enrichment analysis revealed that both BRG1-dependent promoters and enhancers, as well as BRG1-independent enhancers, were enriched for AP-1 components (FOS and FOSL1) and AP-1-associated factor ATF3 motifs, consistent with their cooperative functions[77,78] (Extended Data Fig. 9e). Notably, BRG1-independent promoters were enriched for NFY and SP1 motifs, as observed in mES cells (Fig. 3a), suggesting a conserved regulatory mechanism across different cell types.

To explore the dosage sensitivity of these changes, we divided regions with decreased chromatin accessibility into buffered and linear response categories (Fig. 6f). Promoters and insulators exhibited mostly buffered responses, whereas enhancers showed both linear and buffered responses (Fig. 6g). We further classified enhancers based on their dosage sensitivity (G1–G5 groups) (Extended Data Fig. 9f). Buffered enhancers retained more ATAC-seq signal than linear enhancers at moderate BRG1 levels (Fig. 6h and Extended Data Fig. 9g). G1 buffered enhancers displayed strong H3K27ac enrichment, whereas G5 linear enhancers had weaker enrichment (Fig. 6i). Of the 276 SEs identified based on MED1 signals, 209 (75.7%) were affected by BRG1 depletion. Notably, SEs were preferentially enriched in the G1 group (Fig. 6j), consistent with our findings in mES cells.

*MYC* has a critical role in lung cancer[79], and we found its expression to be highly sensitive to BRG1 dosage: it was upregulated 1.9-fold at 38% BRG1 (Fig. 6k). By contrast, the expression of oncogene *KRAS* was unaffected (Extended Data Fig. 9h). Near-complete depletion of mSWI/SNF led to growth defects, whereas partial depletion had a limited impact (Extended Data Fig. 9i).

Collectively, our results reveal conserved mSWI/SNF dosage sensitivity in regulation of chromatin accessibility, cellular homeostasis and *MYC* expression. Modest reductions in BRG1 levels led to upregulation of *MYC* without compromising proliferation, whereas further depletion impaired proliferation. These findings support the proposition that mSWI/SNF functions as a tumor suppressor while remaining essential for cancer cell growth[21,22,80] and further suggest that its dosage-sensitive regulatory effects contribute to cancer progression, particularly via control of *MYC* oncogene activation.

## Discussion
By precisely controlling protein levels of BRG1, the essential ATPase subunit of the mSWI/SNF complex, we investigated the dosage-sensitive effects of BRG1 on chromatin binding, chromatin accessibility and transcription. We found that BRG1 binding was linearly sensitive to its dosage, whereas chromatin accessibility of BRG1 targets exhibited either linear or buffered sensitivity depending on the type of regulatory element, and the effects on transcription were largely buffered (Fig. 7). Promoters and insulators typically showed buffered responses, whereas enhancers displayed both buffered and

**Fig. 6 | Conserved dosage sensitivity in human lung epithelial cells. a**, Western blot analysis of BRG1 in *BRM*-knockout (KO) BEAS-2B cells after treatment with the indicated concentrations of AU-15330 for 24 h (top). Data are presented as the mean ± s.e.m. from four independent experiments. Quantification of relative BRG1 abundance following AU-15330 treatment (bottom). BRG1 expression relative to that of tubulin was normalized to DMSO-treated (0 nM) *BRM*-KO BEAS-2B cells. **b**, PCA of ATAC-seq data across different AU-15330 concentrations. Cells were treated with AU-15330 for 24 h. **c**, Pie chart depicting relative percentages of significantly changed ATAC-seq signals at promoters, enhancers and insulators. **d**, Pie charts depicting percentages of significantly changed ATAC-seq signals at BRG1 target sites in promoters, enhancers and insulators. **e**, Heatmap showing profiles of protein and histone modifications (CUT&RUN and ChIP–seq[89]) (red) and ATAC-seq (blue) for regions falling into genomic elements at the indicated dTAG13 concentration (bottom). The top panel shows aggregate coverage plots with mean enrichment at promoters (red), enhancers (cyan) and insulators (orange). **f**, Histogram of delta values for BRG1-dependent regulatory elements (promoters + enhancers + insulators). **g**, Histograms of delta values for BRG1-dependent promoters, enhancers and insulators. The red line indicates delta = 0. **h**, Chromatin accessibility changes for each of the five groups of BRG1-dependent enhancers in BEAS-2B cells. Lines show the mean of each group, shaded areas indicate the s.e.m. **i**, Heatmap showing relative enrichment of chromatin marks and MED1 binding across different groups of ATAC-seq peaks. **j**, Number of SEs in each group. **k**, *MYC* expression relative to that of *GAPDH* at different BRG1 levels. The level was further normalized by 0 nM AU-15330 conditions. Data represent three independent replicates, each with three technical replicates. Bars indicate the average.

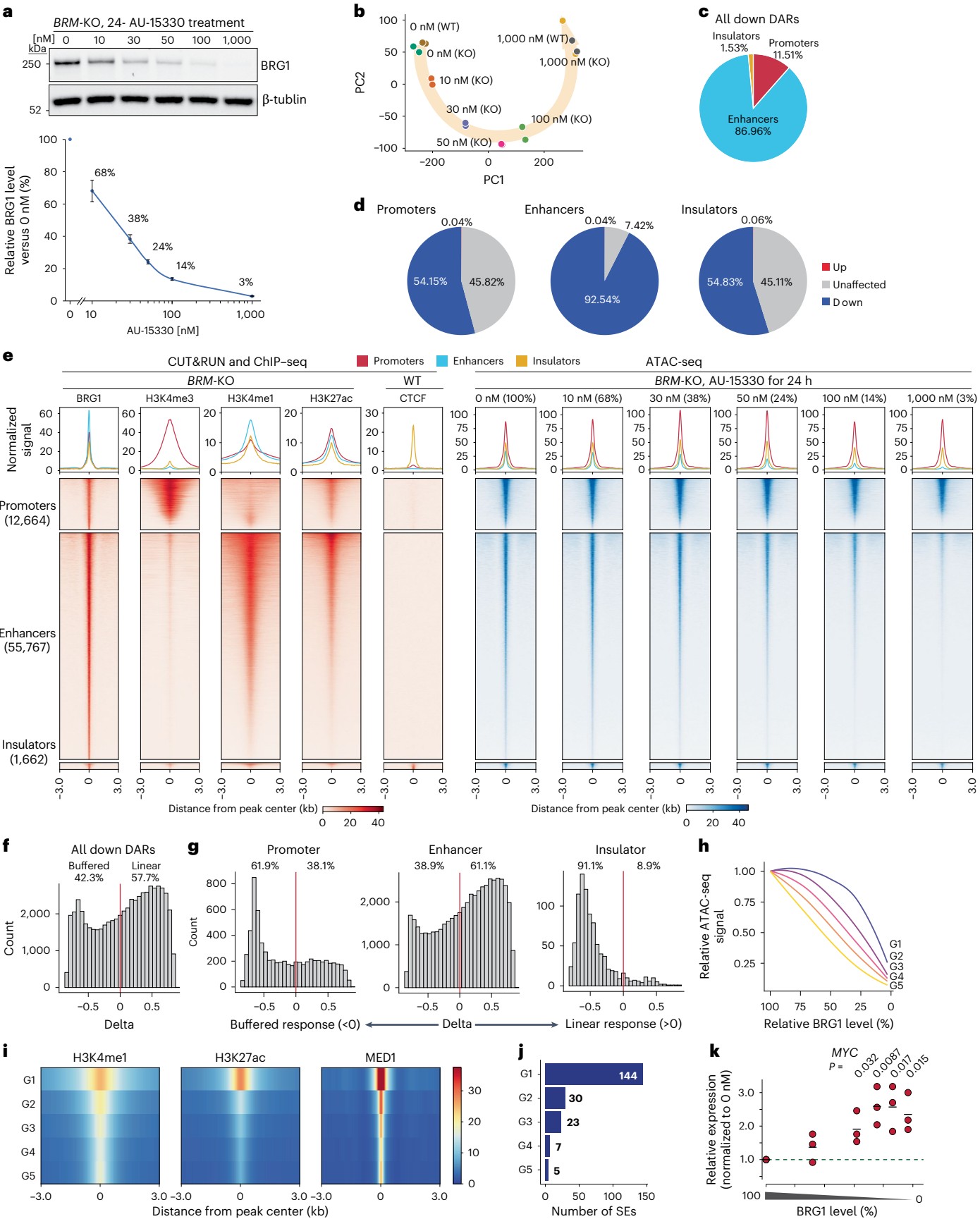

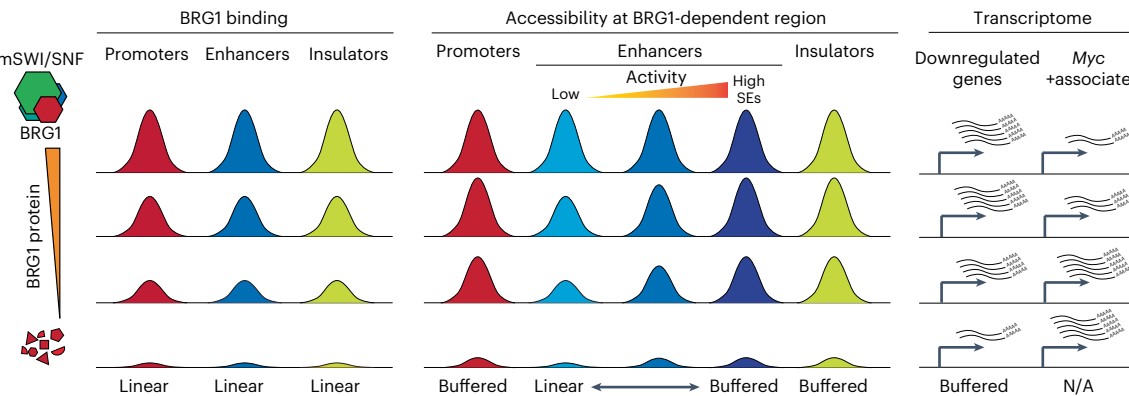

**Fig. 7 | Model illustrating differential responses of genomic regions to BRG1 dosage.** Model illustrating the dosage sensitivity of BRG1 binding, chromatin accessibility and transcriptional responses. Chromatin accessibility response is further categorized by type of regulatory element and enhancer activity, reflecting different patterns of dosage sensitivity across different genomic regions. Red, blue and yellow peaks represent BRG1-bound promoters, enhancers and insulators, respectively. Each row indicates a different BRG1 dosage level, with the bottom row representing the BRG1-depleted condition. Regions labeled 'Linear' exhibited changes in signals that were correlated proportionally with the BRG1 level, whereas those labeled 'Buffered' showed limited changes at certain levels of BRG1 depletion. '*Myc*+associate' denotes *Myc* and MYC-target gene expression. N/A, not applicable.

linear sensitivities, depending on their activity level (Fig. 7). These findings highlight the complexity of dosage-sensitive gene regulation and underscore the critical role of BRG1 in maintaining precise control of chromatin architecture and transcription.

The response of enhancers to BRG1 dosage showed variable sensitivity, with strong enhancers showing buffered responses. This observation was consistent with those of previous studies of the temporal dynamics following mSWI/SNF inhibition, in which strong enhancers responded slowly and weak enhancers responded rapidly[11,12]. Despite these similarities, our data indicate that the dosage sensitivity and temporal dynamics are not entirely the same. One possible explanation for this is that the temporal dynamics are influenced by the chromatin state at the time of inhibitor treatment, whereas dosage sensitivity reflects the degree of enhancer dependency on mSWI/SNF and contributions of other regulatory factors. A crowded chromatin environment may hinder rapid changes, whereas more accessible regions may close gradually via spontaneous nucleosome sliding, which is assumed to take place over minutes to hours[81]. Furthermore, weak enhancers may depend more on BRG1 to maintain their accessibility. In buffered regions, the effects of BRG1 may be mediated through histone modifications or other regulatory mechanisms, such that only low levels of BRG1 are required.

We also found distinct patterns of BRG1 dependency among regulatory elements. Approximately one-third of the BRG1-bound regulatory elements (7,258 of 20,326) exhibited BRG1-dependent accessibility. The enrichment of NFYA and CTCF at BRG1-independent promoters and insulators, respectively, suggests that other factors may compensate for mSWI/SNF depletion. For instance, SNF2H appeared to function predominantly at BRG1-bound insulators, although some sites required both SNF2H and BRG1. Similarly, EP400/TIP60 has been reported to compensate for mSWI/SNF loss at promoters[16], suggesting that cross-talk between mSWI/SNF and other remodelers might be specific to certain types of genomic element. This compensatory role of other remodelers might contribute to the BRG1 dosage sensitivity of accessibility regulation, as both promoters and insulators were enriched for buffered response to BRG1 dosage. In addition, we observed accumulation of Polycomb proteins at BRG1-dependent promoters following BRG1 loss, and expression of *Hox* genes, which are known to be Polycomb targets, was negatively correlated with BRG1 levels. These results suggest that Polycomb proteins may be redistributed in response to BRG1 dosage, as suggested by a study of acute BRG1 depletion[14]. A recent study also found that mSWI/SNF-dependent promoters were enriched for TATA-box elements and flanked by mSWI/

SNF-bound enhancers, whereas GC-rich housekeeping genes were less affected[82]. These findings raise the question of whether differences in dependency arise from compensatory mechanisms or distinct regulatory processes governing chromatin accessibility.

Our results show that the BRG1 dosage sensitivity of enhancer accessibility is linked to enhancer activity. This differential dosage sensitivity response to intermediate BRG1 levels, with decreased accessibility for typical enhancers but maintenance of accessibility for SEs, is intriguing as SEs are often associated with oncogene regulation in cancers[63]. A similar pattern, with typical enhancers but not SEs being affected, has been observed in *SMARCB1*-defective rhabdoid tumors, in which mSWI/SNF stability is compromised[24,25]. In these contexts, residual mSWI/SNF activity is critical for expression of oncogenes, including MYC-target genes[83]. This suggests that selective enhancer dysregulation in response to altered mSWI/SNF dosage may be a general mechanism contributing to oncogenesis. Notably, the role of BRG1 in *MYC* regulation appears to be context-dependent, promoting *MYC* expression in leukemia and prostate cancer[16,60,84-86] but repressing it in cardiomyocytes and lung cancer cells[70,87,88]. Our finding that *MYC* is sensitive to a decrease in BRG1 dosage is consistent with observations of BRG1 disruption or rescue in lung cancer cell lines[70,74]. These studies suggest that BRG1 dosage sensitivity may have a key role in lung cancer biology. In addition, dosage-sensitive regulation of mSWI/SNF is complicated by the presence of redundant ATPase subunits, BRG1 and BRM. In both mES cells and *BRM*-knockout BEAS-2B cells, BRG1 alone supported chromatin accessibility, and its depletion alone was sufficient to reduce accessibility of SEs. By contrast, both BRG1 and BRM need to be depleted to impair SEs in HAP1 cells[11], indicating functional redundancy. Thus, the BRG1 dosage sensitivity observed in this study may reflect a broader principle of mSWI/SNF dosage-sensitive regulation across diverse cell types.

In conclusion, our study reveals molecular mechanisms underlying BRG1 dosage sensitivity with different effects on chromatin binding, accessibility and transcription. These findings deepen our understanding of the function of mSWI/SNF in stem cell biology and emphasize the importance of precise chromatin regulation in maintenance of cellular identity. The insights presented here represent a foundation for future exploration of how alterations in mSWI/SNF dosage affect accessibility and transcription in development, cancer and other disease contexts.

## Online content

Any methods, additional references, Nature Portfolio reporting summaries, source data, extended data, supplementary information,

acknowledgements, peer review information; details of author contributions and competing interests; and statements of data and code availability are available at https://doi.org/10.1038/s41588-025-02305-z.

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

## Methods

This study complied with all relevant ethical regulations and was approved by the Harvard Committee on Microbiological Safety, which serves as the Institutional Biosafety Committee for Harvard Medical School, under ID 22-079-A03. All animal experiments were conducted in accordance with the National Institutes of Health Guide for the Care and Use of Laboratory Animals and approved by the Institutional Animal Care and Use Committee of Boston Children's Hospital and Harvard Medical School under protocol number IS00000270-9.

### Cell culture, establishment of cell lines and chemical treatment

The mES cells were derived from blastocysts of C57BL/6J mice (Jax, 000664) and cultured on 0.1% gelatin-coated plates under 2i/LIF conditions. Cells were maintained in DMEM (Gibco, 11960069) supplemented with 15% fetal bovine serum (FBS, Sigma-Aldrich, F6178), 2 mM GlutaMAX (Gibco, 35050061), 1 mM sodium pyruvate (Gibco, 11360), 1× MEM nonessential amino acids (Gibco, 11140050), 0.084 mM 2-mercaptoethanol (Gibco, 21985023), 100 U ml$^{-1}$ penicillin–streptomycin (Gibco, 15140122), 1000 IU ml$^{-1}$ LIF (Millipore, ESG1107), 0.5 µM PD0325901 (Tocris, 4192) and 3 µM CHIR99021 (Tocris, 4423). BEAS-2B cells were purchased from the American Type Culture Collection (ATCC, CRL-3588). Cells were maintained in RPMI1640 (Gibco, 11875093) supplemented with 10% FBS and 100 U ml$^{-1}$ penicillin–streptomycin. *Drosophila* S2 cells, purchased from Gibco (R69007), were grown in Schneider's *Drosophila* medium (Thermo Fisher Scientific, catalog no. 21720024) supplemented with 10% heat-inactivated FBS, 0.1% Plutonic F-68 (Gibco, 24040032) and 1% penicillin–streptomycin at 28 °C.

To generate the *Brg1*-dTAG and *Snf2h*-dTAG cell lines, mES cells were cotransfected with targeting plasmids and pX330. puro (Addgene, catalog no. 110403) using Lipofectamine 2000 (Invitrogen, 11668030). After transfection, cells were selected with puromycin (Gibco, A1113803). Colonies derived from single cells were picked and genotyped for successful integration, followed by validation and analysis. Sequences of single guide RNAs are listed in Supplementary Table 1.

To generate the *BRM*-knockout cell line, BEAS-2B cells were cotransfected with pL-CRISPR.EFS.GFP (Addgene, catalog no. 57818) and pL-CRISPR.EFS.tRFP (Addgene, catalog no. 57819) using Lipofectamine 2000. Two days after transfection, GFP and RFP double-positive cells were sorted to 96-well plates with a single cell per well. Knockout was confirmed by genotyping, followed by western blot analysis. Genotyping primers are listed in Supplementary Table 2.

For chemical treatment, mES cells and BEAS-2B were seeded in 12-well plates the day before treatment. Cells were treated with dTAG13 (100 pM to 100 nM) or AU-15330 (10 nM to 1,000 nM) by complete replacement of the culture medium. After the specified treatment period, cells were harvested using TrypLE (Gibco, 12605010) or Trypsin-EDTA (Gibco, 25200072) and resuspended in fresh culture medium for downstream analysis.

### Western blot

Cells were resuspended at a final concentration of $1 \times 10^4$ cells µl$^{-1}$ in 1× LDS buffer (Invitrogen, NP0007). Protein samples were prepared by sonication using a Bioruptor 300 (Diagnode), followed by heat denaturation. Proteins were separated by 3–8% SDS–PAGE, then transferred onto nitrocellulose or polyvinylidene fluoride (PVDF) membranes (Invitrogen, IB23002 or IB24002) using the iBlot2 system (Invitrogen). Membranes were blocked in 3% bovine serum albumin (Jackson ImmunoResearch, 001-000-162) prepared in Tris-buffered saline (Cell Signaling Technology, 12498S) with 0.1% Tween-20 (TBS-T) for 1 h at room temperature. After blocking, membranes were incubated overnight at 4 °C with primary antibodies. The next day, membranes were washed four times with TBS-T, followed by incubation with secondary antibodies conjugated to fluorescent-dye or HRP at room temperature. Then, membranes were washed four times with TBS-T, and those incubated with HRP antibodies were developed using SuperSignal West Pico PLUS Chemiluminescent Substrate (Thermo, 34580) before being imaged on an iBright 1500 system. The sources of all antibodies are listed in Supplementary Table 3.

### Immunofluorescence

Cells were seeded on a collagen-coated eight-well culture slide (Corning, 354630) and treated the following day with dTAG13 or AU-15330 by complete replacement of the culture medium. Twenty-four hours after treatment, cells were fixed in 4% paraformaldehyde (PFA) for 20 min and permeabilized with PBS containing 0.5 % Triton X-100. Cells were incubated overnight at 4 °C with primary antibodies (listed in Supplementary Table 3), followed by incubation with secondary antibody for 2 h at room temperature. They were then mounted using Vectashield (Vector Laboratories, H-1200-10) and imaged with a confocal microscope (Zeiss, LSM800). The relative signal level was calculated by normalizing to the mean intensity of the dimethyl sulfoxide (DMSO)-treated control.

### CUT&RUN, ATAC-seq and RNA-seq library preparation and sequencing

CUT&RUN was performed as previously described[90,91]. In brief, 50,000 mES cells were incubated with activated Concanavalin A Magnetic Beads (Polysciences, 86057-3). Samples were then incubated overnight with the antibody, followed by incubation with pA-MNase (home-made). DNA was fragmented by incubation with CaCl$_2$ and subsequently purified using a phenol-chloroform extraction. Sequencing libraries were prepared using a NEBNext Ultra II DNA Library Preparation Kit for Illumina (New England Biolabs, E7645S).

ATAC-seq was performed as previously described[42,92]. In brief, 10,000 to 20,000 mES cells were treated with adapter-loaded Tn5 transposase (home-made) at 37 °C. The reaction was stopped by adding a buffer containing SDS, and proteins were digested with Proteinase K at 55 °C. Tween-20 was added to quench the SDS. Sequencing libraries were prepared using NEBNext High-Fidelity 2× PCR Master Mix (New England Biolabs, M0541S). RNA-seq libraries were prepared using the SMART-Seq Stranded Kit (Takara Bio, 63444) using 5,000 mES cells. All libraries were sequenced on the NextSeq 550 system (Illumina) using paired-end 75-bp reads.

### Co-immunoprecipitation

Antibody beads were prepared using a Dynabeads Co-Immunoprecipitation Kit (Thermo, 14321D). In brief, 0.4 mg of beads were coupled with 1.4 µg of antibody overnight, followed by washing with HB, LB and SB buffers. Suspensions of nuclei were prepared using a modified protocol[43]. About $2.0–3.5 \times 10^7$ cells were used for each experiment. Cells were swelled in buffer A (10 mM HEPES, pH 7.9, 1.5 mM MgCl$_2$ and 10 mM KCl) supplemented with 1 mM DTT and Complete Protease Inhibitor Cocktail (Roche, 11836170001), then lysed by homogenization using a 25-gauge needle with 5–8 strokes. Nuclei were pelleted by centrifugation at 900$g$ for 5 min and resuspended in buffer C (20 mM HEPES, pH 7.9, 20% glycerol, 420 mM NaCl, 1.5 mM MgCl$_2$ and 0.2 mM EDTA) supplemented with 1 mM DTT and protease inhibitor cocktail. After a 30-min incubation on ice, the sample was cleared by centrifugation at 21,100$g$ for 10 min. The supernatant was collected and flash-frozen in liquid nitrogen for further use.

Before immunoprecipitation, nuclear lysates were diluted with two-thirds of the original volume of 20 mM HEPES, pH 7.9, and 0.3% NP-40 to reduce the NaCl concentration. The diluted nuclear lysates were incubated with the antibody–bead solution for 3 h at 4 °C. The samples were then washed with wash buffer (50 mM Tris, pH 8.0, 150 mM NaCl, 1 mM EDTA, 10% glycerol and 0.5% Triton X-100). Proteins were eluted in 1× LDS buffer by boiling and subsequently analyzed by western blotting.

## Cell proliferation assay

Cells were seeded in 96-well plates at 300 cells per well for 6 days, 1,500 cells per well for 3 days and 5,000 cells per well for 1 day of treatment. After the specified treatment period, DNA was stained with a CyQUANT NF Cell Proliferation Assay kit (Invitrogen, C35007) and detected with a CLARIOstar Plus plate reader (BMG LABTECH). Each sample was measured in triplicate, and the mean fluorescence intensity was used for calculations. Relative signal levels were normalized to the DMSO-treated control.

## Alkaline phosphatase staining

Cells were seeded in 12-well plates at 10,000 cells per well and then treated with dTAG13 at the specified concentrations for 6 days. At the endpoint of the treatment period, cells were washed gently with phosphate-buffered saline, stained using a StemAb Alkaline Phosphatase Staining Kit II (Amsbio, AMS.00-0055) and imaged with a BZ-X810 microscope (KEYENCE). Colonies were scored manually.

## Quantitative RT–PCR analysis

Total RNA was isolated from cells using TRIzol reagent (Invitrogen). cDNA was synthesized with SuperScript IV, and RT–PCR was performed using Fast SYBR Green Master Mix (Applied Biosystems) on a QuantStudio 6 Pro (Applied Biosystems). Relative mRNA expression levels were calculated using a standard curve, with *GAPDH* as an internal control. Primer sequences are listed in Supplementary Table 2.

## Data analysis

**CUT&RUN data analysis.** Raw reads with sequencing adaptors or low-quality ends reads were trimmed using trim_galore (v.0.6.7) with '–illumina -q 20 –length 30 –paired -j 2'. Then, the clean reads were aligned to the mouse mm10 genome via bowtie2 (v.2.5.1) with default parameters. The bam files were sorted with samtools (v.1.6). Duplicates were marked and removed with Picard (v.2.26.4-0); then, MACS2 (v.2.2.7.1) was used to call significant peaks with parameters '-f BAMPE –keep-dup all -g mm -q 0.05 –broad –max-gap 200'. To obtain highly conserved BRG1 binding peaks, the peaks detected in both biological repeats were defined as binding sites. ChIPseeker (annotatePeak function) was used to annotate BRG1 binding peaks. Peaks located near transcription start sites (±1,000 base pairs) were defined as promoter binding. Peaks overlapping with CTCF-binding sites (CTCF ChIP–seq data were downloaded from GEO: GSE30206) were classified as insulator binding, and the remaining peaks were classified as enhancer binding. For Integrative Genomics Viewer (IGV) visualization of the signals, the read coverage was normalized by reads per kilobase per million (RPKM) using bamCoverage (v.3.5.5).

**ATAC-seq data analysis.** The same method as for CUT&RUN data analysis was used to remove sequencing adaptors or low-quality reads, and the clean reads were aligned to mm10 via bowtiew2 with parameter '-q -X 2000'. Then, Picard was used to remove duplicates, and MACS was used to call significant peaks with parameters '-f BAMPE –keep-dup all -g mm -B -q 0.05'.

We first tried to use *Drosophila* spike-in DNA to construct the ATAC-seq library. The sequencing reads were aligned to *Drosophila* genome dm6, and unmapped reads were further mapped to the mouse genome. The spike-in read numbers varied significantly from sample to sample; this did not reflect the actual biology but may have been due to inconsistent amounts of spike-in DNA added across different samples. Therefore, spike-in DNA could not be applied for normalization in our data. The total peak numbers showed limited differences among all samples (peak numbers: 0 nM 43,212; 0.3 nM 51,119; 1 nM 40,673; 3 nM 40,164; 10 nM 35,207; 100 nM 37,441). Most ATAC-seq peaks did not change after BRG11 depletion, especially the strong peaks. There were no global ATAC-seq signal changes. The strong peaks could be used for calculation of size factors. The upper-quartile normalization method from the calcNormFactors function in edgeR (v.3.32.1) was used to calculate 95% quantiles of the counts for each library, after removing genes that were zero in all libraries.

For differentially accessible peak detection, the ATAC-seq read counts for each BRG1-binding site were calculated using multiBamCov (v.2.30.0). The read count matrix was then loaded into edgeR. After normalization and estimation of dispersions, a generalized linear model was fitted, and likelihood ratio tests were applied (glmFit and glmLRT functions). Peaks with a false discovery rate of less than 0.05 were defined as differentially accessible peaks.

For motif enrichment analysis, the findMotifsGenome.pl function in HOMER2 (http://homer.ucsd.edu/homer/motif/) was applied to bed files. The '-mask' parameter was applied to use the repeat-masked sequence. Briefly, HOMER identifies overrepresented motifs within a set of peaks. Then, it selects random background regions as a background set. Finally, it screens its library of reliable motifs against the target and background sequences to determine enrichment. Enrichment *P* values for all motifs in all peaks were then corrected for multiple testing by calculating the Benjamini-adjusted *P* value. Motifs with an adjusted *P* value less than 0.05 were considered to be significantly enriched.

**RNA-seq data analysis.** The same method as for CUT&RUN and ATAC-seq data analyses was used to remove sequencing adaptors or low-quality reads, and the clean reads were aligned to the mouse genome using HISAT2 (v.2.2.1). To quantify gene expression, featureCounts (v.2.0.1) was used to count the sequencing reads within each gene. Then, edgeR (v.3.32.1) was employed for normalization and differential expression analysis. Genes with reads per kilobase per million less than 1 were defined as having low expression and excluded from the differential expression analysis. Then, DEGs were identified using the likelihood ratio test (glmFit and glmLRT functions in edgeR). Genes with false discovery rate less than 0.05 and absolute value of fold change greater than 1.5 were defined as DEGs. enrichR (v.3.4) was used for gene ontology enrichment analysis of DEGs.

**Modeling buffered and linear response peaks.** For each BRG1-binding peak, read counts from all dosage-treated samples were normalized to the control sample. Then, $z$-scores were calculated as follows: $z = (x - \mu)/\sigma$, where $x$ is the raw signal, $\mu$ is the mean across all samples and $\sigma$ is the standard deviation of all samples. Peaks with maximum $z$-score in any sample were excluded from modeling. Two models—a buffered model and a linear model—were then fitted to assess changes in CUT&RUN, ATAC-seq and transcriptome data relative to BRG1 protein levels as measured by western blotting. The linear model was

$$RC \sim 1 - BRG,$$

and the buffered model was

$$RC \sim 1 - BRG^3,$$

where RC is a vector of normalized read counts among all samples, and BRG is the BRG1 protein level of all samples. The residual sum of squares, RSS, was calculated for each peak in each model as follows:

$$RSS = \sum_{k=i}^{n} (y_i - \bar{y}),$$

and a delta value of residuals was computed to quantify the distance between the two models, enabling a comparison of their fit to the data:

$$\Delta = RSS_{sensitive} - RSS_{buffered}.$$

Here a positive delta value indicates that the peak is closer to the linear model than the buffered model, whereas a negative delta value indicates that the peak is closer to the buffered model.

**SE analysis.** A total of 231 SEs of mouse ES cells were obtained from R. Young's laboratory[62]. The original genomic locations of these enhancers were based on mm9 genome assembly. We converted the genome coordinates from mm9 to mm10 using the LiftOver tool from USCS utilities. The overlaps of SEs with other genomic features were measured using bedtools (v.2.30.0). The SEs of BEAS-2B cells were defined as previously described[62].

**Chromatin states enrichment analysis.** A chromatin states map for mES cells generated using ChIP–seq data for E14 cells from the ENCODE project was downloaded from https://github.com/guifengwei/Chrom-HMM_mESC_mm10. ChromHMM was used to identify chromatin states according to combinations of histone modifications. A 12-state model was trained, containing mostly promoters, enhancers, repressed Polycomb sites, transcription elongation regions, CTCF-bound sites and intergenic regions. To calculate chromatin state enrichments of given genomic regions, we used Fisher's exact test to compare enrichment to background regions. The background regions were generated by shuffling regions using bedtools (shuffleBed).

**Training of ChromBPNet models.** We used ChromBPNet to predict the chromatin accessibilities of the BRG1-binding sites. ChromBP-Net was downloaded from GitHub (https://github.com/kundajelab/chrombpnet), and we followed the tutorial provided to train models for ATAC-seq of untreated (0 nM) and treated (100 nM) mES cells. We first generated background regions for training a bias model to capture the Tn5 enzyme effect. We then used this model to regress out the effects of the enzyme from our ATAC-seq profiles. Next, bias-factorized ChromBPNet models were trained to predict the base-pair resolution chromatin accessibility profiles. Finally, we generated base-pair resolution importance scores for all BRG1-binding sites as described in the ChromBPNet tutorial. We further used the TF-MODISCO (https://github.com/kundajelab/tfmodisco) motif discovery algorithm to identify sequence motifs enriched in different enhancers. We then used a previously described method[66] to build a hidden Markov model to identify sequences with a significant positive or negative influence on chromatin accessibility within the BRG1-binding sites.

**Statistics and reproducibility.** No statistical methods were used to predetermine sample sizes, but our sample sizes are similar to those reported in previous publications[11,12,42]. The experiments were not randomized. Drosophila DNA for ATAC-seq and spike-in yeast DNA for CUT&RUN were excluded from analyses owing to substantial variation in spike-in read counts across samples, possibly caused by inconsistent spike-in DNA amounts. Three data points were excluded from the cell proliferation assay because their values exceeded the detectable range or were suspected to result from uneven cell seeding. Data distribution was assumed to be normal, but this was not formally tested. Data collection and analysis were not performed blind to the conditions of the experiments.

### Reporting summary

Further information on research design is available in the Nature Portfolio Reporting Summary linked to this article.

## Data availability

The sequencing data reported in this study are available via the Gene Expression Omnibus (GEO) with accession numbers GSE274469 and GSE294015. All public datasets used in this study are listed in Supplementary Table 4, including GSE64825 (ref. 6), GSE198517 (ref. 16), GSE87819 (ref. 48), GSE115110 (ref. 56), GSE44286 (ref. 62), GSE56098 (ref. 93), GSE155062 (ref. 94), GSE30203 (ref. 95), GSE56839 (ref. 96), GSE11431 (ref. 97) and GSE56051 (ref. 89). ChromBPNet models are available via Figshare at https://doi.org/10.6084/m9.figshare.28705430 (ref. 98). Source data are provided with this paper.

## Code availability

The code for analysis of the sequencing data is available via GitHub at https://github.com/YiZhang-lab/Brg1_dTag and Zenodo at https://doi.org/10.5281/zenodo.15951452 (ref. 99).

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

## Acknowledgements

We thank J. Yao for help with cell sorting and C. Zhou, Q. Yang and J. Ren for helpful discussions and thoughtful comments on the manuscript. This project was supported by the NIH (R01HD092465) and the Howard Hughes Medical Institute (HHMI). Y.H. is supported by a JSPS Overseas Research Fellowship from the Japan Society for the Promotion of Science (JSPS). Y.Z. is an investigator of the HHMI. This article is subject to HHMI's Open Access to Publications policy. HHMI laboratory heads have previously granted a nonexclusive CC BY 4.0 license to the public and a sublicensable license to HHMI in their research articles. Pursuant to those licenses, the author-accepted manuscript of this article can be made freely available under a CC BY 4.0 license immediately upon publication.

## Author contributions

Y.Z. supervised the project. Y.H. designed and performed experiments, and C.Z. performed bioinformatic analysis. Y.H., C.Z. and Y.Z. wrote the manuscript.

## Competing interests

The authors declare no competing interests.

## Additional information

**Extended data** is available for this paper at https://doi.org/10.1038/s41588-025-02305-z.

**Correspondence and requests for materials** should be addressed to Yi Zhang.

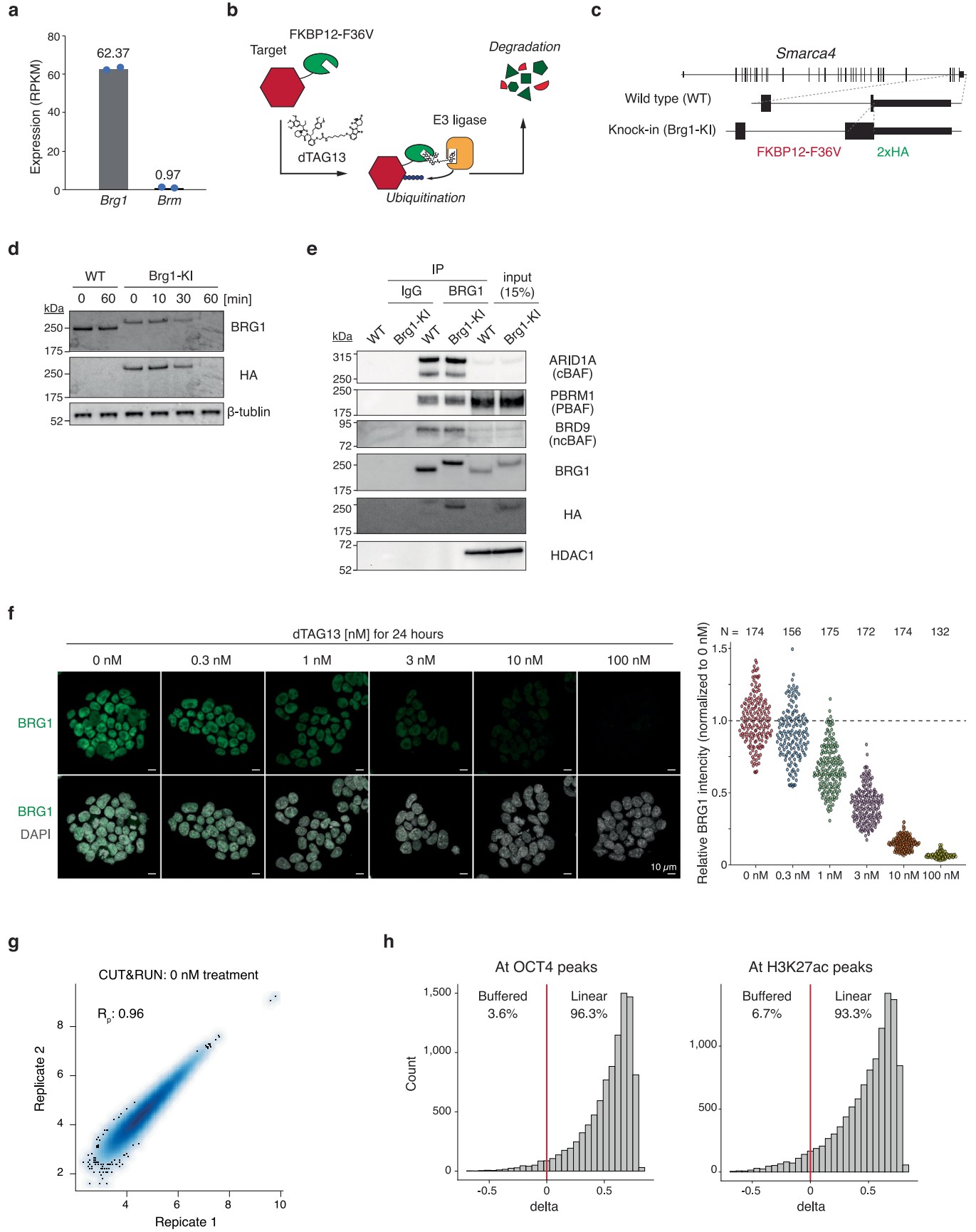

**Extended Data Fig. 1 | See next page for caption.**

**Extended Data Fig. 1 | dTAG knock-in does not perturb BRG1 complex formation. a**, mRNA levels of *Brg1* (*Smarca4*) and *Brm* (*Smarca2*) in mESCs. Data represent two independent experiments. **b**, Schematic of dTAG mediated protein degradation system. FKBP12-F36V fused protein undergoes degradation in the presence of dTAG13. **c**, Schematic illustration of FKBP12-F36V-HA knock-in at the 3′ end of *Brg1*. **d**, Western blot analysis of Brg1-KI mESCs after treatment for different times with 100 nM dTAG13 or DMSO. Data are representative of two independent experiments. **e**, BRG1 immunoprecipitation in WT or Brg1-KI mESCs, followed by western blot analysis. Data are representative of three independent experiments. **f**, Immunostaining analysis of BRG1 in the Brg1-KI mESCs after treatment with the indicated concentrations of dTAG13 for 24 hours. Representative images (left), and relative BRG1 intensity in each cell (right). The GFP intensity of individual cell with 0 nM dTAG13 treatment was set as 1. Data includes two independent replicates. **g**, Correlations of the replicates of CUT&RUN at 0 nM. $R_p$: Pearson correlation. **h**, Histograms showing delta values of BRG1 binding sites co-occupied with OCT4 (left) or H3K27ac (right). The red line indicates delta = 0.

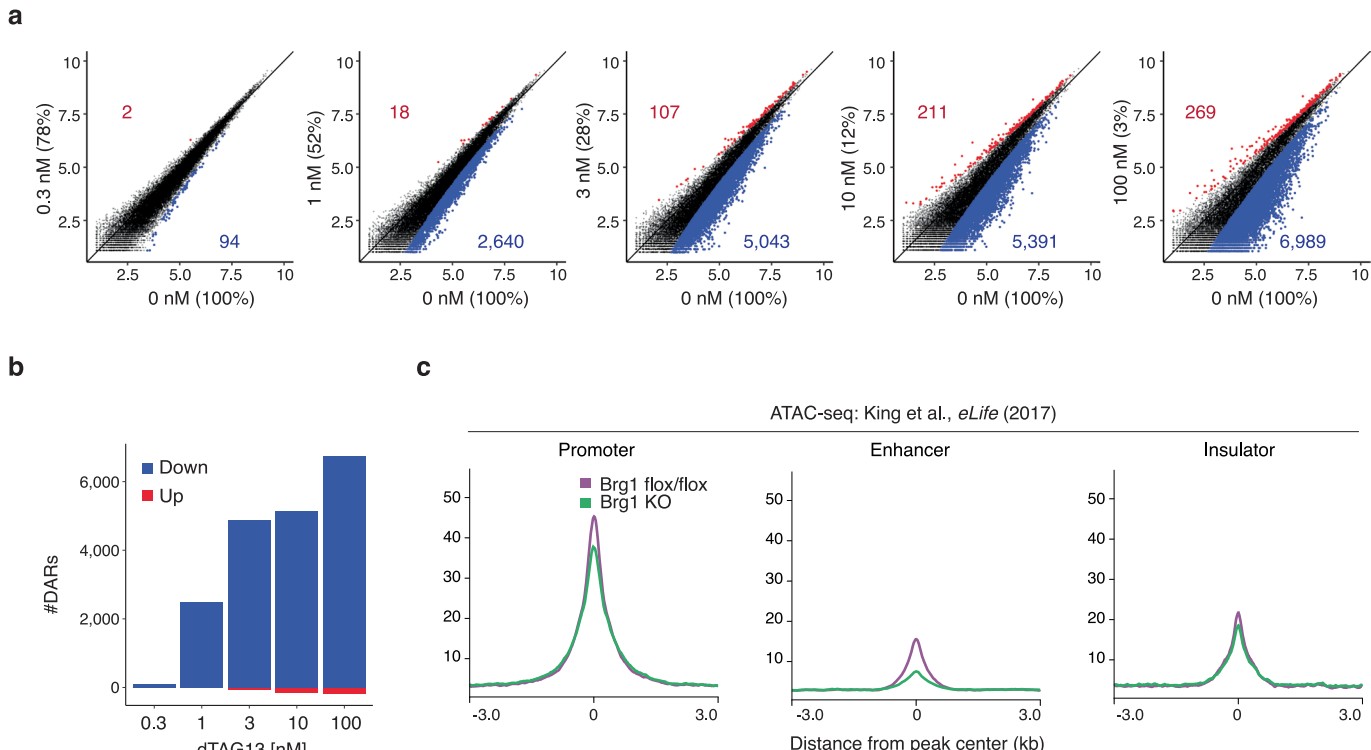

**Extended Data Fig. 2 | The number of DARs increases progressively as BRG1 degradation increases. a**, Comparison of chromatin accessibility (ATAC-seq) changes at BRG1-targeted regulatory elements following 24 hours of dTAG13 treatment at 0 nM (DMSO, x-axis) and 0.3 to 100 nM (y-axis). **b**, The numbers of differentially accessible regions (DARs) identified at varying dTAG13 concentrations. **c**, Aggregate coverage plot showing mean ATAC-seq peaks of Brg1 flox/flox (red) and Brg1-KO (green) at promoters, enhancers and insulators[48].

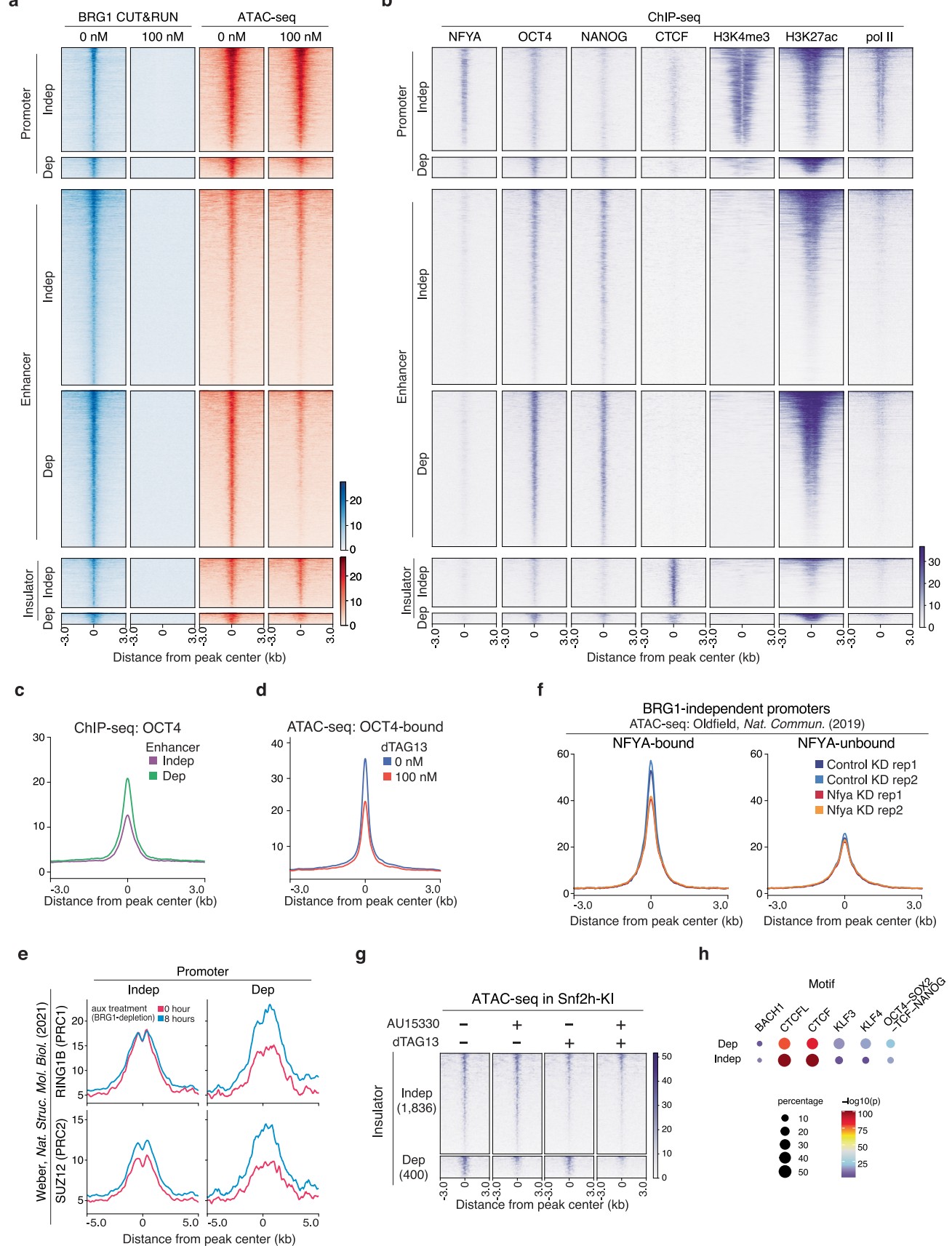

**Extended Data Fig. 3 | See next page for caption.**

**Extended Data Fig. 3 | Chromatin signature of BRG1-dependent and -independent regulatory elements. a**, Heatmap showing BRG1-binding (Blue) and ATAC-seq (Red) for BRG1-independent and -dependent regulatory elements. Indep, BRG1-independent regions. Dep, BRG1-dependent regions. **b**, Heatmap showing TFs and histone modifications for BRG1-independent and -dependent regulatory elements. **c**, Aggregate coverage plot showing mean OCT4 peaks at BRG1-dependent and -independent regulatory elements. **d**, Aggregate coverage plot showing mean ATAC-seq signal at OCT4-bound regions. **e**, Aggregate coverage plot showing mean RING1B (top) and SUZ12 (bottom) signals of 0 hour (red) and 8 hours (cyan) at BRG1-dependent and -independent promoters[14].

BRG1 was depleted by auxin-degron system for 8 hours[14]. **f**, Aggregate coverage plot showing mean ATAC-seq signal at NFYA-bound (left) and -unbound (right) regions at BRG1-independent promoters[56]. **g**, Heatmap showing ATAC-seq signals for BRG1-independent and BRG1-dependent insulators following treatment with 1 μM AU-15330 and/or 100 nM dTAG13 for 24 hours. **h**, Transcription factor motif enrichment at BRG1-independent (Indep) and BRG1-dependent (Dep) ATAC-seq peaks in insulators. The color scale represents −log10(P value), and dot size corresponds to the effect size measured as odds ratio (OR). A two-sided Fisher's exact test was performed.

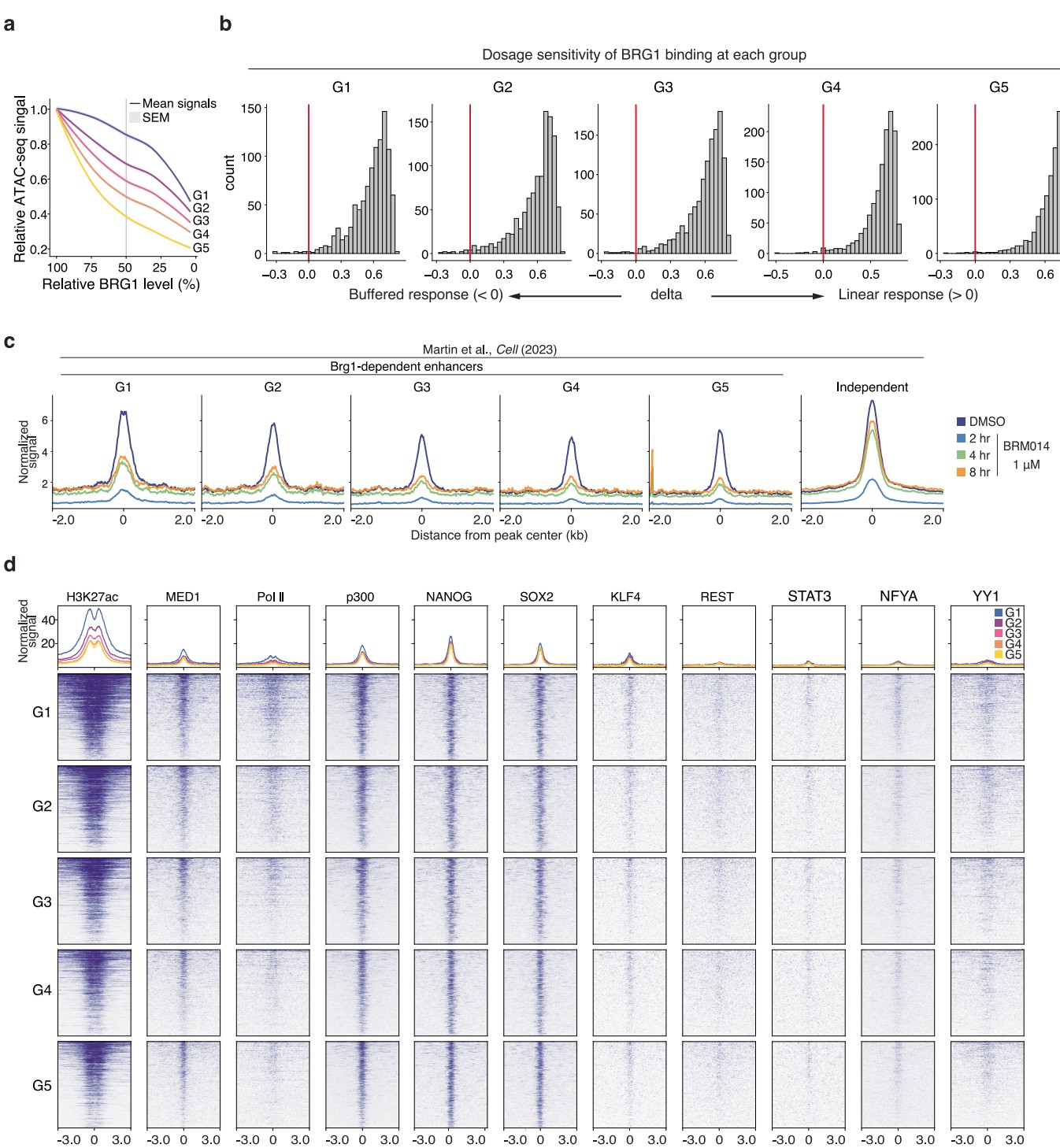

**Extended Data Fig. 4 | Characteristic of BRG1 dosage-sensitive enhancers.**
**a**, Chromatin accessibility changes for each of the five groups of BRG1-dependent enhancers at different BRG1 levels. Line showed mean of each group, shed area indicates SEM **b**, Histogram of BRG1 binding delta value for each group. Red line indicates delta = 0. **c**, Aggregate coverage plot showing mean ATAC-seq signal at each enhancer regions after BRM014, BRG1/BRM ATPase inhibitor, treatment for the indicated time on the right. **d**, Bottom: Heatmap showing TFs and histone modifications of the BRG1-dependent regions in each of the groups. Top: aggregate coverage plots showing mean enrichment of the indicated TFs and histone modifications for each group.

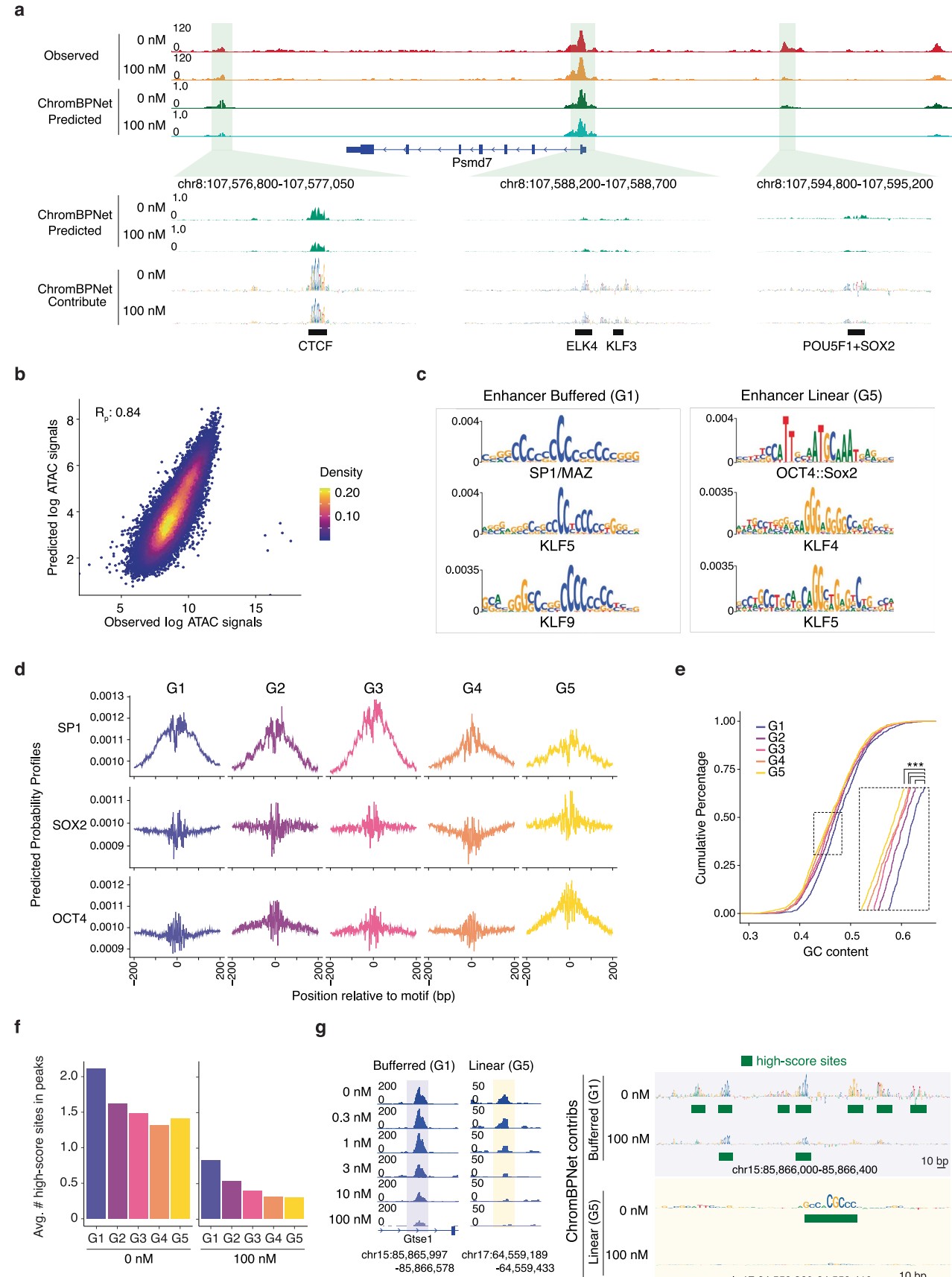

**Extended Data Fig. 5 | See next page for caption.**

**Extended Data Fig. 5 | Machine learning model predicts features of buffered and linearly sensitive enhancers. a**, Illustration of ChromBPNet predict chromatin accessibilities at *Psmd7* gene locus. Observed track: signals detected by ATAC-seq experiments; Predicted track: ChromBPNet predicted signals; Contribution track: the genome sequence scaled according to ChromBPNet contribute scores. **b**, Scatter plot showing the correlation between predicted and observed signals. **c**, Motifs identified by ChromBPNet models in G1 enhancers (left), and G5 enhancers (right). **d**, Marginal footprints of TF motifs of different groups using predicted profiles from mESC ATAC-seq ChromBPNet models. **e**, Cumulative percentage of GC content in different groups. *P* values were calculated with Wilcoxon's *t*-test (two-sided) by comparing each group with G1: $p = 2.5 \times 10^{-4}$ (G2); $p = 4.2 \times 10^{-8}$ (G3), $p = 1.3 \times 10^{-10}$ (G4), $p = 1.1 \times 10^{-14}$ (G5). *** $p < 0.001$ **f**, Histograms showing the number of ChromBPNet predicted high score sites from the 0 nM mESCs model or 100 nM dTAG13 mESCs model. **g**, Representative genome browser view of high-score site at buffered or linear enhancers.

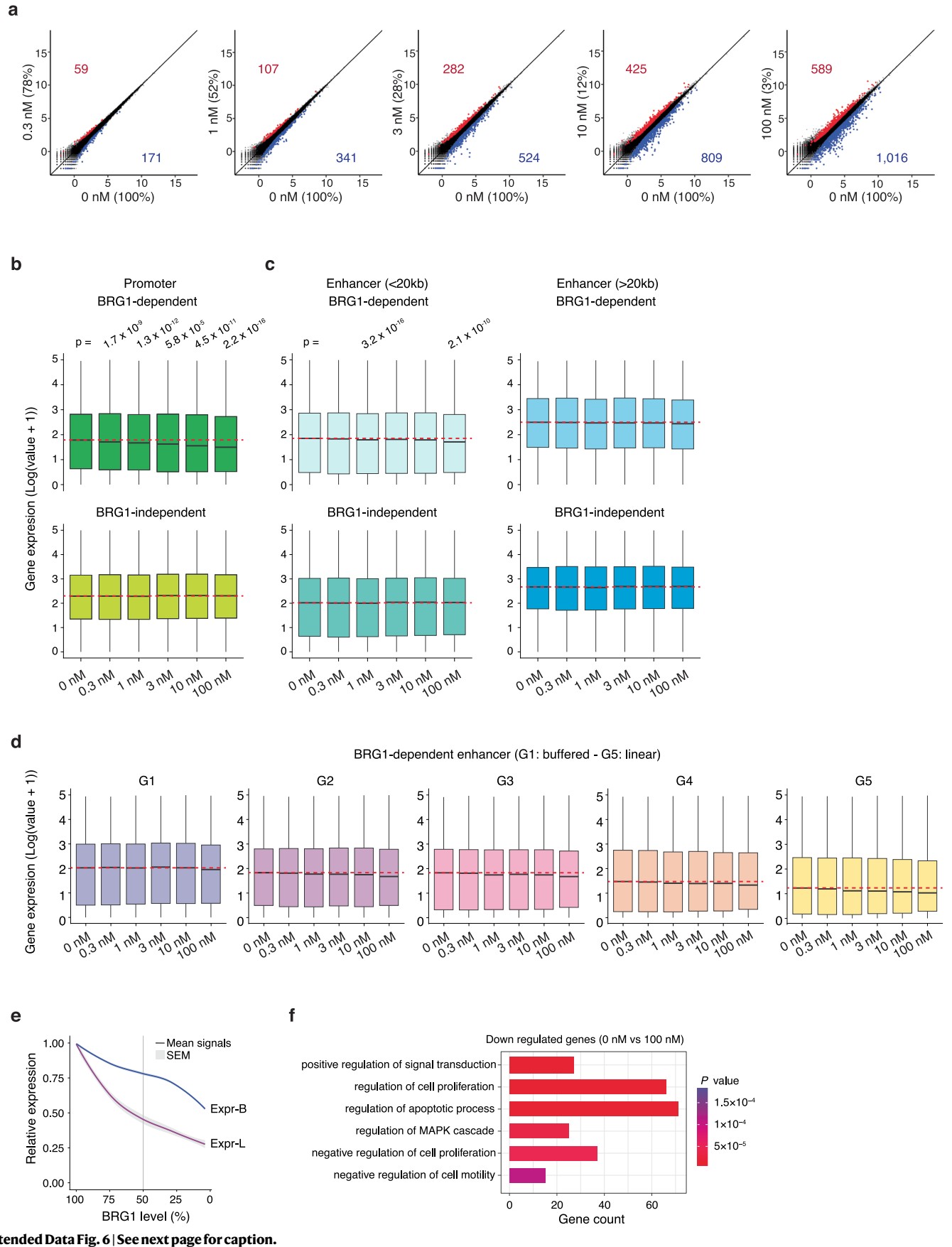

**Extended Data Fig. 6 | See next page for caption.**

**Extended Data Fig. 6 | BRG1 dosage response in transcriptome. a**, Comparison of transcription (RNA-seq) changes after 24 hours of dTAG13 treatment at concentrations of 0 nM (DMSO, x-axis) and 0.3 to 100 nM (y-axis). The numbers of differentially expressed genes under each condition are shown. **b**, Boxplot showing gene expression associated with BRG1-dependent promoter (1,128 genes) (top) and -independent promoter (6,826 genes) (bottom). Boxes are defined by the 25th to 75th percentiles, the median is shown as a horizontal black line, and whiskers extend up to 1.5 times the interquartile range (IQR) from the quartiles. Dash line indicates the mean signal at 0 nM condition. Data represent two independent experiments. *P* values were calculated with Wilcoxon's *t*-test (two-sided). **c**, Boxplot showing gene expression associated with BRG1-dependent enhancers (top) and -independent enhancers (bottom). Genes are classified by the distance of TSS-enhancer <20 kb (left) and >20 kb (right), each including 4,738 (dep, <20 kb), 7,340 (indep, <20 kb), 3,190 (dep, >20 kb) and

3,740 (indep, >20 kb) genes. Boxes are defined by the 25th to 75th percentiles, the median is shown as a horizontal black line, and whiskers extend up to 1.5 times the IQR from the quartiles. Dashed line indicates the mean signal at 0 nM condition. Data represent two independent experiments. *P* values were calculated with Wilcoxon's *t*-test (two-sided). **d**, Boxplot showing gene expression associated with each BRG1-dependent enhancer groups. Boxes are defined by the 25th to 75th percentiles, the median is shown as a horizontal black line, and whiskers extend up to 1.5 times the IQR from the quartiles. Dashed line indicates the mean signal at 0 nM condition. Each group include 1,037 (G1), 1,036 (G2), 1,037 (G3), 1,036 (G4) and 1,037 (G5) genes. Data represent two independent experiments. **e**, Line plot showing gene expression for each group. Line showed mean of each group, shed area indicates SEM **f**, Gene Ontology (GO) terms enriched for all the down-regulated genes after 100 nM dTAG13 treatment. A two-sided Fisher's exact test was performed.

**a**

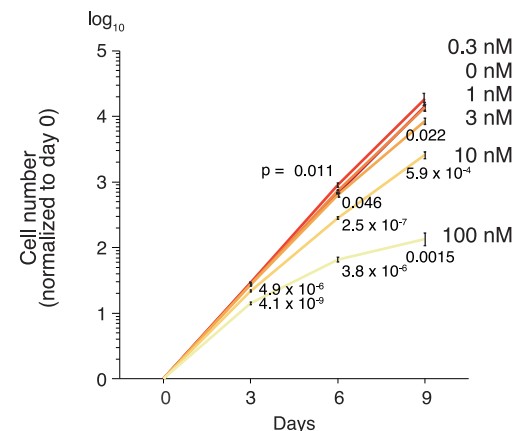

**b**

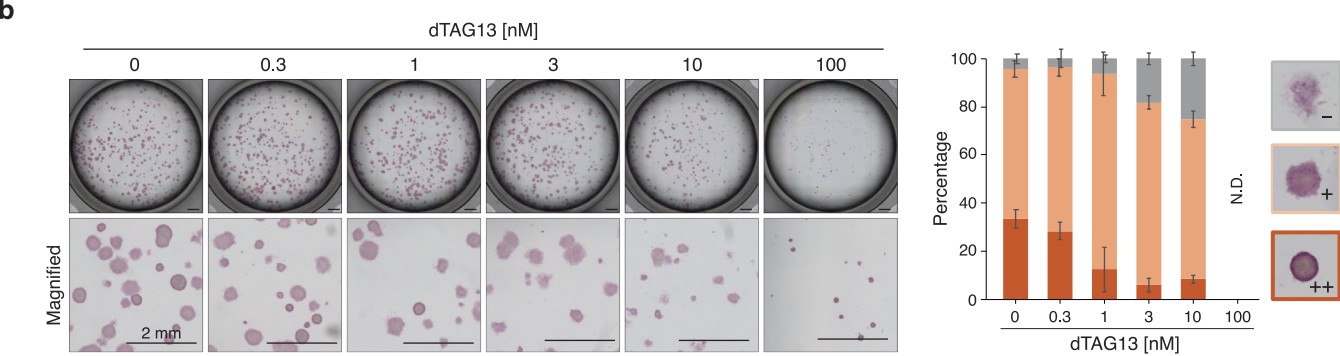

**c**

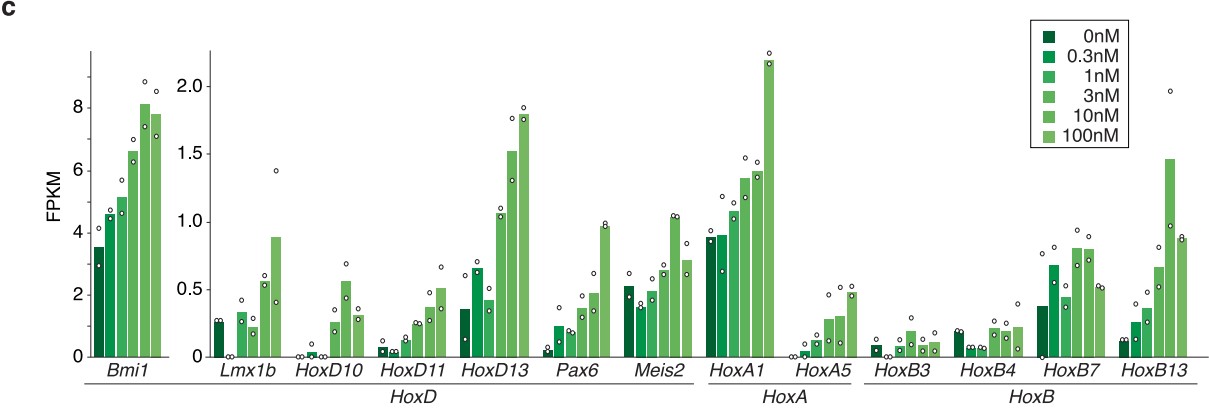

**Extended Data Fig. 7 | BRG1 dosage sensitivity in cellular homeostasis. a,** Cell proliferation assay under dTAG13 treatment. Cells are counted and passaged every 3-days. Data are presented as mean ± SEM from five independent experiments. *P* values were calculated with Student's *t*-test (two-sided). **b,** Representative images of alkaline phosphatase (AP) staining upon dTAG13 treatment at different concentrations for 6-days (left), and quantification of the AP staining (right). Colonies were classified into three groups based on AP signals and morphology. Red, orange and gray color represent undifferentiated (++), intermediate (+) and differentiated (-), respectively. Data are presented as mean ± SD from three independent experiments. **c,** Relative gene expression of Polycomb-targets at different BRG1 levels. Expression levels were normalized to the expression at 0 nM dTAG13 (DMSO). Data represent two independent experiments.

**a**

WT, 24 hour
AU15330 treatment

[nM]   0   1,000

kDa

250 ─  BRG1

250 ─  BRM

250 ─  PBRM1

52 ─  β-tublin

**b**

*SMARCA2 (BRM)*
178,274 bp

sgRNA                                    sgRNA

WT-PCR

KO-PCR

*BRM*-KO

WT  #1  #2

WT-PCR

KO-PCR

*BRM*-KO

WT  #1  #2

kDa

250 ─  BRG1

250 ─  BRM

250 ─  PBRM1

52 ─  β-tublin

**c**

WT                    *BRM*-KO (#2)

50 μm

**d**

CUT&RUN

BRG1                         BRM

WT        *BRM*-KO      WT       *BRM*-KO

Normalized signal

60
40
20
0

genes

-3.0  0  3.0  -3.0  0  3.0  -3.0  0  3.0  -3.0  0  3.0

Distance from peak center (kb)

0   15   30

**e**

*BRM*-KO

[Hours]   0   0.5   1   2

kDa

250 ─  BRG1

250 ─  PBRM1

52 ─  β-tublin

**g**

N =   138   150   144   158

1.5

1.0

0.5

Relative BRG1 intencity
(normalized to 0 nM)

0 nM   30 nM   100 nM   1,000 nM

**f**

*BRM*-KO, 24 hour AU15330 treatment

0 nM        30 nM       100 nM      1,000 nM

BRG1

DAPI

10 μm

**h**

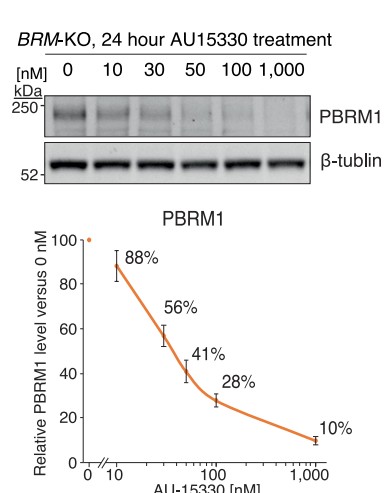

*BRM*-KO, 24 hour AU15330 treatment

[nM]   0   10   30   50   100  1,000

kDa

250 ─  PBRM1

52 ─  β-tublin

PBRM1

100
80
60
40
20
0

Relative PBRM1 level versus 0 nM

88%
56%
41%
28%
10%

0  10          100          1,000

AU-15330 [nM]

**Extended Data Fig. 8 | See next page for caption.**

**Extended Data Fig. 8 | Knockout of BRM does not perturb cell homeostasis in BEAS-2B cells. a**, Western blot analysis of mSWI/SNF components in WT BEAS-2B cells after treatment with 0 nM AU-15330 (DMSO) or 1,000 nM AU-15330 for 24 hours. Data are representative of four independent experiments. **b**, Schematic illustration of BRM-KO strategy (top). Red bar indicates sgRNA-target and green arrows indicate primer pair for genotyping. Knockout of *BRM* was confirmed by PCR genotyping (left) and Western blot (right). Data are representative of two independent experiments. **c**, Representative bright field image of WT (left) and *BRM*-KO (right) BEAS-2B cells. Data are representative of two independent experiments. **d**, Heatmap showing profiles of BRG1 (red) and BRM (blue) in WT and *BRM*-KO BEAS-2B cells. The top panels show aggregate coverage plots

with mean enrichment. **e**, Western blot analysis of BRG1 and PBRM1 in *BRM*-KO BEAS-2B cells treated with 1,000 nM AU-15330 for the indicated time. Data are representative of two independent experiments. **f**, Immunostaining of BRG1 in *BRM*-KO BEAS-2B cells after treatment with the indicated concentrations of AU-15330 for 24 hours. **g**, Relative BRG1 intensity measured from the immunostaining shown in (f). Each dot represents a cell. **h**, Western blot analysis of PBRM1 in *BRM*-KO BEAS-2B cells after treatment with the indicated concentrations of AU-15330 for 24 hours (top). Quantification of relative BRG1 abundance following AU-15330 treatment (bottom). BRG1 expression relative to Tubulin was normalized to DMSO-treated (0 nM) *BRM*-KO BEAS-2B cells. Data are presented as mean ± SEM from four independent experiments.

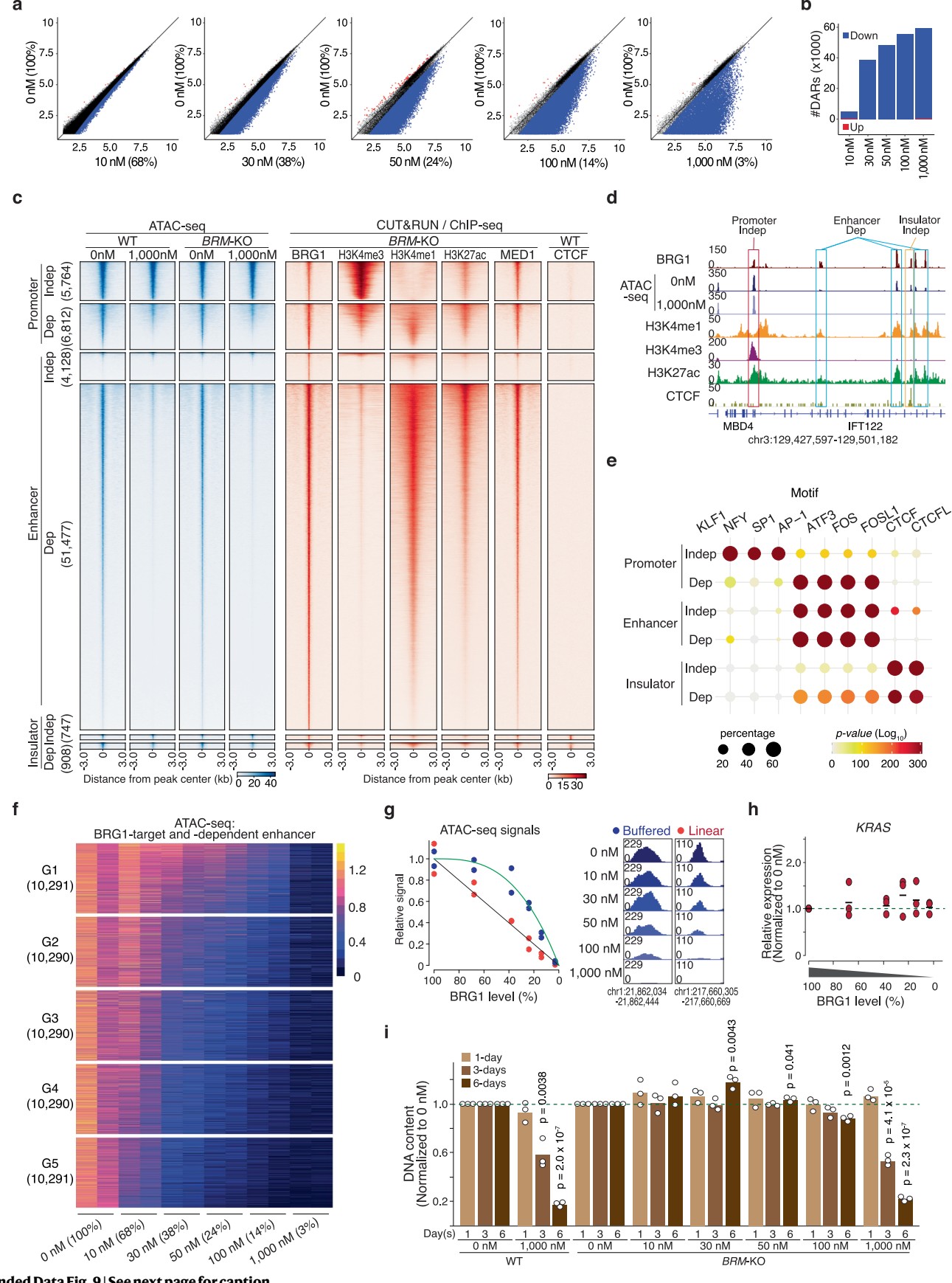

**Extended Data Fig. 9 | See next page for caption.**

**Extended Data Fig. 9 | BRG1 dosage sensitivity and effects can also be observed in the BEAS-2B lung epithelial cells. a**, Comparison of ATAC-seq changes at BRG1-targeted regions following 24 hours of AU-15330 treatment at 0 nM (DMSO, x-axis), and 10–1,000 nM (y-axis). **b**, The numbers of differentially accessible regions (DARs) identified at different AU-15330 concentrations. **c**, Heatmap showing BRG1-binding (Blue) and ATAC-seq (Red) for BRG1-independent and -dependent regulatory elements. Indep, BRG1-independent regions. Dep, BRG1-dependent regions. **d**, Genome browser view showing BRG1 binding and ATAC-seq signals across regions containing promoters, enhancers, and insulators at a representative locus. **e**, Transcription factor motif enrichment at BRG1-independent (Indep) and BRG1-dependent (Dep) ATAC-seq peaks in promoters, enhancers, and insulators. The color scale represents −log10(P value), and dot size corresponds to the effect size measured as odds ratio (OR). A two-sided Fisher's exact test was performed. **f**, Heatmap showing BRG1-

dependent enhancer regulatory elements, clustered into five groups based on delta values, ranging from buffered to linear response regions. **g**, Representative genome browser views showing linear and buffered response to BRG1 dosage (right). Red dots represent linearly sensitive regions, and blue dots represent buffered response regions (left). The linear and buffered models are illustrated in the diagram. **h**, *KRAS* expression relative to *GAPDH* at different BRG1 levels. The level is further normalized by 0 nM AU-15330 conditions. Data represent three independent experiments, each with three technical replicates. **i**, Relative DNA counts after 1, 3, and 6 days of AU-15330 treatment at the indicated concentrations. The signal was normalized to the 0 nM condition for samples at the indicated days. Data represent three independent experiments, each with three technical replicates. P values were calculated with Student's *t*-test (two-sided), comparing samples of the same genotype and treatment time.

# Reporting Summary

## Statistics

For all statistical analyses, confirm that the following items are present in the figure legend, table legend, main text, or Methods section.

| n/a | Confirmed | |
|---|---|---|
| ☐ | ☒ | The exact sample size (*n*) for each experimental group/condition, given as a discrete number and unit of measurement |
| ☐ | ☒ | A statement on whether measurements were taken from distinct samples or whether the same sample was measured repeatedly |
| ☐ | ☒ | The statistical test(s) used AND whether they are one- or two-sided *Only common tests should be described solely by name; describe more complex techniques in the Methods section.* |
| ☒ | ☐ | A description of all covariates tested |
| ☐ | ☒ | A description of any assumptions or corrections, such as tests of normality and adjustment for multiple comparisons |
| ☐ | ☒ | A full description of the statistical parameters including central tendency (e.g. means) or other basic estimates (e.g. regression coefficient) AND variation (e.g. standard deviation) or associated estimates of uncertainty (e.g. confidence intervals) |
| ☐ | ☒ | For null hypothesis testing, the test statistic (e.g. *F*, *t*, *r*) with confidence intervals, effect sizes, degrees of freedom and *P* value noted *Give P values as exact values whenever suitable.* |
| ☒ | ☐ | For Bayesian analysis, information on the choice of priors and Markov chain Monte Carlo settings |
| ☒ | ☐ | For hierarchical and complex designs, identification of the appropriate level for tests and full reporting of outcomes |
| ☒ | ☐ | Estimates of effect sizes (e.g. Cohen's *d*, Pearson's *r*), indicating how they were calculated |

*Our web collection on statistics for biologists contains articles on many of the points above.*

## Software and code

Policy information about availability of computer code

| Data collection | No software was used |
|---|---|
| Data analysis | The codes for analyzing the sequencing data are available at GitHub: https://github.com/YiZhang-lab/Brg1_dTag and at Zenodo: https://doi.org/10.5281/zenodo.15951452. |

For manuscripts utilizing custom algorithms or software that are central to the research but not yet described in published literature, software must be made available to editors and reviewers. We strongly encourage code deposition in a community repository (e.g. GitHub). See the Nature Portfolio guidelines for submitting code & software for further information.

## Data

Policy information about availability of data

All manuscripts must include a data availability statement. This statement should provide the following information, where applicable:
- Accession codes, unique identifiers, or web links for publicly available datasets
- A description of any restrictions on data availability
- For clinical datasets or third party data, please ensure that the statement adheres to our policy

The sequencing data reported in this study are available at Gene Expression Omnibus: GSE274469 and GSE294015. ChromBPNet models were uploaded to figshare: https://doi.org/10.6084/m9.figshare.28705430. Source data are provided with this paper. All other data supporting the findings of this study are available from the corresponding author on reasonable request.

## Research involving human participants, their data, or biological material

Policy information about studies with [human participants or human data](). See also policy information about [sex, gender (identity/presentation), and sexual orientation]() and [race, ethnicity and racism]().

| Reporting on sex and gender | N/A |
|---|---|
| Reporting on race, ethnicity, or other socially relevant groupings | N/A |
| Population characteristics | N/A |
| Recruitment | N/A |
| Ethics oversight | N/A |

Note that full information on the approval of the study protocol must also be provided in the manuscript.

# Field-specific reporting

Please select the one below that is the best fit for your research. If you are not sure, read the appropriate sections before making your selection.

☒ Life sciences  ☐ Behavioural & social sciences  ☐ Ecological, evolutionary & environmental sciences

For a reference copy of the document with all sections, see [nature.com/documents/nr-reporting-summary-flat.pdf]()

# Life sciences study design

All studies must disclose on these points even when the disclosure is negative.

| Sample size | No statistical method was used to predetermine sample size but our sample sizes are similar to those reported in previous publications (PMID: 39747581, 33558760, 33558757). For publicly available datasets used in the study, we analyze the relevant experimental system (mESCs, BEAS-2B). |
|---|---|
| Data exclusions | Drosophila DNA for ATAC-seq and spike-in yeast DNA for CUT&RUN are excluded from analysis due to substantial variation in spike-in read counts across samples, which is possibly caused by inconsistent spike-in DNA amounts. Three data points were excluded from the cell proliferation assay because their values exceeded the detectable range or were suspected to result from uneven cell seeding. |
| Replication | All experiments were performed 2-6 biological replicates as indicated in manuscript. All attempts are replications were successful, all samples are included in manuscript. |
| Randomization | Randomization was not considered in this cell culture-based study. Comparison were done between treated and control cells. |
| Blinding | Blinding was not considered for this study. All cells were grown in identical culture condition (+/- treatment). No subjective measurements were applied. |

# Reporting for specific materials, systems and methods

We require information from authors about some types of materials, experimental systems and methods used in many studies. Here, indicate whether each material, system or method listed is relevant to your study. If you are not sure if a list item applies to your research, read the appropriate section before selecting a response.

## Materials & experimental systems

| n/a | Involved in the study |
|---|---|
| ☐ | ☒ Antibodies |
| ☐ | ☒ Eukaryotic cell lines |
| ☒ | ☐ Palaeontology and archaeology |
| ☐ | ☒ Animals and other organisms |
| ☒ | ☐ Clinical data |
| ☒ | ☐ Dual use research of concern |
| ☒ | ☐ Plants |

## Methods

| n/a | Involved in the study |
|---|---|
| ☒ | ☐ ChIP-seq |
| ☒ | ☐ Flow cytometry |
| ☒ | ☐ MRI-based neuroimaging |

# Antibodies

| | |
|---|---|
| Antibodies used | BRG1 rabbit monoclonal (Abcam, ab110641, Lot# GR3375498-11)<br>HA-Tag rabbit monoclonal (Cell Signaling Technology, 3724S, Lot# 11)<br>HA-Tag mouse monoclonal (Cell Signaling Technology, 2367S, Lot# 5)<br>ARID1A mouse monoclonal (Novus Biologicals, NBP2-61623, Lot# MAB-03564)<br>PBRM1 rabbit polyclonal (Bethyl Laboratories, A301-591A, Lot# 5)<br>BRD9 rabbit polyclonal (Bethyl Laboratories, A303-781A, Lot# 3)<br>HDAC1 mouse monoclonal (Novus Biologicals, NBP2-52937, Lot# MAB-03647)<br>SNF2H rabbit polyclonal (Abcam, ab72499, Lot# GR255705-60)<br>Rhodamine-conjugated Tubulin (Bio-Rad, 12004165, Lot# 64512248)<br>Normal Rabbit IgG (Cell Signaling Technology, 2729S, Lot# 11)<br>BRM rabbit monoclonal (Cell Signaling Technology, 11966T, Lot# 5)<br>H3K4me1 rabbit monoclonal (Cell Signaling Technology, 5326T, Lot# 7)<br>H3K4me3 rabbit polyclonal (Cell Signaling, 9727S, Lot# 6)<br>H3K27ac rabbit polyclonal (Sigma -Aldrich, 07-360, Lot# 3935826)<br>MED1 rabbit polyclonal (Bethyl Laboratories, A300-793A-T, Lot# 11)<br>Rabbit IgG HRP-conjugated (Invitrogen, 31460, Lot# RB230194)<br>Mouse IgG HRP-conjugated (Invitrogen, 31430, Lot# QD216575)<br>Rabbit IgG Alexa 647-conjugateed (Invitrogen, 21245, Lot# 1660844)<br>Mouse IgG Alexa 800-conjugateed<br>Rabbit IgG Alexa 488-conjugateed |
| Validation | BRG1: Validated for IF, CUT&RUN, WB and IP by manufacture (https://www.abcam.com/en-us/products/primary-antibodies/brg1-antibody-epncir111a-ab110641), and for ChIP-seq in other study (PMID: 34446700). In this study, WB signal are disappeared after AU-15330 treatment, supporting antibody specificity.<br>HA-Tag (rabbit): Validated for WB, IP, IF and ChIP by manufacture (https://www.cellsignal.com/products/primary-antibodies/ha-tag-c29f4-rabbit-mab/3724?srsltid=AfmBOoqlmaMbX6xuW6Rn59VFSV89zLd5B6Gk6Ervl8gPdMCn-L_zMFLR), and for WB in other study (PMID: 33558760). In this study, CUT&RUN signal are disappeared after dTAG treatment, supporting antibody specificity.<br>HA-Tag (mouse): Validated for WB and IF by manufacture (https://www.cellsignal.com/products/primary-antibodies/ha-tag-6e2-mouse-mab/2367?srsltid=AfmBOorCUxE2zacjtsM-VdNVaDe2a6EbSDHPPiF8X4S6xwfBiHZSoaPl), and for WB in other study (PMID: 39747581)<br>ARID1A: Validated for WB and IF by manufacture (https://www.novusbio.com/products/arid1a-antibody-cl3595_nbp2-61623?srsltid=AfmBOopiaEZaLvlorrjaBkgETKIHfoi_p5nhIhNUp-dr5vPmu7qQDBab), and for IF in other study (PMID: 39013863)<br>PBRM1: Validated for IHC, IP and WB by manufacture (https://www.fortislife.com/products/primary-antibodies/rabbit-anti-pbrm1-antibody/BETHYL-A301-591), and for WB in other study (PMID: 34937944). In this study, WB signal are disappeared after AU-15330 treatment, supporting antibody specificity.<br>BRD9: Validated for WB and IP by manufacture (https://www.fortislife.com/products/primary-antibodies/rabbit-anti-brd9-antibody/BETHYL-A303-781), and for WB in other study (PMID: 32457312)<br>HDAC1: Validated for WB and IF by manufacture (https://www.novusbio.com/products/hdac1-antibody-cl0510_nbp2-52937?srsltid=AfmBOopt4ExBq3IBBcaRfcg0qhWpTpSKDwpRkq6Sx4KWEXTuqmMG413j)<br>SNF2H: Validated for IP, WB and IF by manufacture (https://www.abcam.com/en-us/products/primary-antibodies/snf2h-antibody-ab72499), and for WB in other study (PMID: 30996347)<br>Rhodamine-conjugated Tubulin: Validated for WB by manufacture (https://www.bio-rad.com/en-us/sku/12004165-hfab-rhodamine-anti-tubulin-primary-antibody-200-ul?ID=12004165), and other study (PMID: 37730997)<br>BRM: Validated for WB, IP, IF and ChIP by manufacture (https://www.cellsignal.com/products/primary-antibodies/brm-d9e8b-xp-rabbit-mab/11966?srsltid=AfmBOoqZdOajF695KLJ03dwnwvZqCJr4qmEYqefAh1rFgvaVlEi7494z), and for ChIP-seq in other study (PMID: 34446700). In this study, CUT&RUN and WB signals are disappeared in BRM-KO cells and AU-15330 treated cells, supporting antibody specificity.<br>H3K4me1: Validated for WB, IF, ChIP CUT&Tag, and CUT&RUN by manufacture (https://www.cellsignal.com/products/primary-antibodies/mono-methyl-histone-h3-lys4-d1a9-xp-rabbit-mab/5326?srsltid=AfmBOopcC2WSYuc_3WeyNYvWtAm2wTp7h20tYc3xXBiHHnRneSzkE0Es),and for CUT&Tag in other study (PMID: 39582024)<br>H3K4me3: Validated for WB and IF by manufacture (https://www.cellsignal.com/products/primary-antibodies/tri-methyl-histone-h3-lys4-antibody/9727?srsltid=AfmBOorFdwncUVJ4BZt-GA4kDvLpHiTRelgt1TUYkSXtVTi4VlwG3hRm), and for ChIP-seq in other study (PMID: 35668082)<br>H3K27ac: Validated for ChIP-seq and WB by manufacture (https://www.sigmaaldrich.com/US/en/product/mm/07360?srsltid=AfmBOopkSh7Onz1crwF-amvCtLQLs4T10iirV738zGsHG14Ud26-98lN), and for CUT&RUN in other study (PMID: 39747581)<br>MED1: Validated for IHC, IP and WB by manufacture (https://www.fortislife.com/products/primary-antibodies/rabbit-anti-med1-antibody/BETHYL-A300-793), and for ChIP-seq in other study (PMID: 26416749)<br>Rabbit IgG HRP-conjugated and Mouse IgG HRP-conjugated:Validated for WB in other study (PMID: 31209294) |

# Eukaryotic cell lines

Policy information about cell lines and Sex and Gender in Research

| | |
|---|---|
| Cell line source(s) | Male mouse ES cell line of C57BL/6J background was established in this study. Drosophila S2 cells can be obtained from Thermo Fisher Scientific (R69007). BEAS-2B cells can be obtained from American Type Culture Collection (ATCC, CRL-3588) |
| Authentication | Authentication Genotype of cell lines was tested at the level of DNA sequence and protein. |
| Mycoplasma contamination | The ES cell lines and BEAS-2B cell line were tested negative for mycoplasma contamination, Drosophila S2 cells were not tested. |

| Commonly misidentified lines (See ICLAC register) | None |
| --- | --- |

## Animals and other research organisms

Policy information about studies involving animals; ARRIVE guidelines recommended for reporting animal research, and Sex and Gender in Research

| Laboratory animals | Female (C57BL/6J, 8 week-old) were mated with Male (C57BL/6J, 10 week-old) and used to collect blastocyst stage embryos. |
| --- | --- |
| Wild animals | No wild animals were used in this study. |
| Reporting on sex | Female mouse was used for blastocyst collection and Male mouse was used for mating |
| Field-collected samples | No field-collected samples were used in this study. |
| Ethics oversight | All animal experiments were performed in accordance with the guidelines of the Institutional Animal Care and Use Committee at Harvard Medical School. All mice were kept under specific pathogen-free conditions within and environmental controlled for tempurature (20-22°C) and humidity (40–70%), and were subjected to a 12-hour light/dark cycle. |

Note that full information on the approval of the study protocol must also be provided in the manuscript.

## Plants

| Seed stocks | N/A |
| --- | --- |
| Novel plant genotypes | N/A |
| Authentication | N/A |

