## [Peer Review File · Nature Genetics]

Precise modulation of BRG1 levels reveals features of mSWI/SNF dosage sensitivity

Corresponding Author: Professor Yi Zhang

Version 0:

Decision Letter:

3rd Dec 2024

Dear Professor Zhang,

sorry for the delay in providing our decision.

Your Article, "Precise modulation of BRG1 levels reveals features of mSWI/SNF dosage sensitivity" has now been seen by 3 referees. You will see from their comments below that while they find your work of interest, some important points are raised. We are interested in the possibility of publishing your study in Nature Genetics, but would like to consider your response to these concerns in the form of a revised manuscript before we make a final decision on publication.

To guide the scope of the revisions, the editors discuss the referee reports in detail within the team, including with the chief editor, with a view to identifying key priorities that should be addressed in revision and sometimes overruling referee requests that are deemed beyond the scope of the current study. In this case, we ask you to address Reviewers' comments in full. Please do not hesitate to get in touch if you would like to discuss these issues further.

We therefore invite you to revise your manuscript taking into account all reviewer and editor comments. Please highlight all changes in the manuscript text file. At this stage we will need you to upload a copy of the manuscript in MS Word .docx or similar editable format.

*2) If you have not done so already please begin to revise your manuscript so that it conforms to our Article format instructions, available

[here](http://www.nature.com/ng/authors/article_types/index.html).

*3) Include a revised version of any required Reporting Summary: <https://www.nature.com/documents/hr-reporting-summary.pdf>

Please be aware of our [guidelines](https://www.nature.com/nature-research/editorial-policies/image-integrity) on digital image standards.

Link Redacted

We hope to receive your revised manuscript within four to eight weeks. If you cannot send it within this time, please let us know.

Nature Genetics is committed to improving transparency in authorship. As part of our efforts in this direction, we are now requesting that all authors identified as 'corresponding author' on published papers create and link their Open Researcher and Contributor Identifier (ORCID) with their account on the Manuscript Tracking System (MTS), prior to acceptance. ORCID helps the scientific community achieve unambiguous attribution of all scholarly contributions. You can create and link your ORCID from the home page of the MTS by clicking on 'Modify my Springer Nature account'. For more information please visit please visit www.springernature.com/orcid.

Sincerely,
Chiara

Chiara Anania, PhD
Associate Editor
Nature Genetics
<https://orcid.org/0000-0003-1549-4157>

Referee expertise:

Referee #1: chromatin regulation

Referee #2: chromatin, gene regulation

Referee #3: genomics

Reviewers' Comments:

Reviewer #1 (Remarks to the Author):

The manuscript „Precise modulation of BRG1 levels reveals features of mSWI/SNF dosage sensitivity“ by Hagihara et al. addresses the BRG1-dependent dosage sensitive effects on BRG1 chromatin binding, chromatin accessibility, and transcriptional regulation in mouse embryonic stem cells. This is of clinical relevance, as loss of function mutations – esp. also heterozygous mutations – of members of the SWI/SNF complexes are frequently found in cancer and can result in neurodevelopmental disorders. While this is not a novel question, they address this in a systematic manner using the dTAG system, where precise control of the protein levels are feasible. They combine their NGS approaches with modelling to define if the responses are linear or buffered relative to the protein levels. The observed effects depend on genomic region and feature (binding, remodelling activity, transcription regulation).

While some of the observations are not novel, this is a thorough study with very clean and well presented data that address the still poorly understood issue of dosage sensitivity of SWI/SNF members. It may advance the understanding of the effect of SWI/SNF dose on disease, if the results are applicable to more disease-relevant contexts, and is thus of importance.

Major comments

1. The authors try to do the analyses in a systematic manner for different genomic features (promoter, enhancer, insulator) and different readouts (binding, chromatin accessibility, transcription regulation). However, this is unfortunately not followed up in all combinations.

a) It would be of high interest to investigate if the transcription factors identified by motif analysis (SP1, NFY) or other chromatin remodellers are responsible for maintaining chromatin accessibility upon BRG1 degradation at BRG1-independent promoters.

b) They further only followed up on the gene expression response, measured by RNA-seq, on the genes whose promoter accessibility was dependent or independent of BRG1. As the major regulatory sites of SWI/SNF complexes are enhancers and they observe here a dependency on activity, it would be even more interesting to look at the expression of associated genes. These could be identified e.g. by Hi-ChIP. In addition, RNA-seq might not capture the direct responses, as the total RNA levels depend also on RNA stability and not only on new transcription. It would be of high relevance to investigate how enhancer sensitivity correlates with the transcriptional effect of target genes.

2. The sensitivity of enhancers to SWI/SNF perturbation based on their activity is not entirely new. Schick et al., Nature genetics 2021 has shown that the stronger the enhancers are, the slower was the response to SMARCA4 degradation and only complete perturbation of all SWI/SNF ATPases also perturb super-enhancers. This is consistent with the observation that BRG1 levels also show similar dependence. This similarity between kinetics after BRG1 degradation and BRG1 levels

could be discussed.

Further, figure 4e shows nicely the enhancer activity decrease from G1 to G5. However, figures 4d and 4f also show that the G2 to G5 regions can be strong enhancers and super-enhancers. Are there differences between the super enhancers falling into the different groups?

3. The authors showed linear relationship of BRG1 protein level to co-occupancy of OCT4 and H3K27ac. Is this also the case for sites with low and high OCT4/H3K27ac occupancy?
4. The authors conclude on page 9 "This suggests that BRG1 has a preference for promoters and insulators and thus a lower concentration is enough to maintain the accessibility of these genomic elements, while a higher concentration of BRG1 is needed to maintain the accessibility of enhancers." The data do not explain why these relationships occur, and as alternative explanations are plausible, possible scenarios should be discussed.
5. It would be great to additionally or instead of showing representative examples of linear and buffered regions (red and blue dots) to plot the distribution of regions falling into the respective categories.
6. How do the authors explain the increase in Nanog (and Klf4) upon lower BRG1 levels?
7. The inverse correlation between BRG1 and Myc expression is of high cancer relevance. It would be interesting to look into cancer data, if this holds true beyond mESCs.

Minor comments

8. In the introduction the authors write that esBAF equals cBAF. This is not entirely correct, as this is only the case for embryonic stem cells (ES) and in the original publication, the term esBAF was used for all ES-specific complexes and not only ES-specific cBAF. This should be rewritten or removed.
9. It would be nice to mention for the reader in the text that AU-15330 is a PROTAC targeting BRG1 and the alternative ATPase BRM, though the latter is hardly expressed in mESC.
10. Please indicate for clarity the timepoints when analyses have been done (e.g. 24h after treatment).
11. Line 310 – intolerant instead of intolerance.
12. Lot numbers of antibodies are missing.
13. Extended data Fig. 5e requires further description.
14. Reporting summary:
Stating that "two biological replicates are accepted practice in the genomics field" is not fully correct.
Antibodies: lot numbers are missing, it would be nice to indicate for what they were used and how much was used (e.g. it is also not provided in the Material and Method section, how much HA antibody was used for the CUT&RUN), only provider pages were indicated for antibody validations
15. Editorial Policy Checklist:
Where is the "inclusion & ethics" statement?

Reviewer #2 (Remarks to the Author):

In their submission, "Precise modulation of BRG1 levels reveals features of mSWI/SNF dosage sensitivity." Hagihara, Zhang, and Zhang utilize dTAG degradation technology in mouse embryonic stem cells (mESCs) to explore how altered dosage of BRG1 (the only expressed ATPase subunit of the BAF complex in mESCs) impacts genome-wide chromatin accessibility. From a mixture of genomics assays, including ATAC-seq, CUT&RUN, and RNA-seq, the authors conclude that BRG1 occupancy across the genome is linearly correlated with protein dose, that accessibility is mostly regulated at the level of enhancers (as opposed to promoters or insulators), and that as a result transcriptional consequences are 'buffered.' The authors' approach is interesting – only one published study comes to mind that leverages dTAG technology to systematically measure the consequences of altered factor dosage on genomic phenomena (PMID37024583); however, the study is largely confirmatory given the multiple recent studies combining degradation technology (PMIDs 34117481; 33558760), chemical inhibition (PMIDs 33558757; 37922899), or genetic manipulation (PMID30996347) with genomic readouts to probe regulation by BRG1. As a result, my enthusiasm for the manuscript from a novelty perspective is significantly diminished. Below, I detail some experiments that could render a revised study more impactful (considering the existing literature).

Major Comments

- Validation of altered dosage within single cells. The manuscript critically depends on a set of experiments / observations made by Western blot: that altered dTAG compound dosage results in fractional modulation of BRG1 at the protein level. However, Western blot is necessarily based on population-averaging, raising the possibility that certain cells within a population have complete loss of BRG1, and certain cells have higher per-cell levels of the protein (thus resulting in a fractional decrease in signal by Western). The authors must validate their dosing strategy in an orthogonal way. I suggest two possible routes: (1) performing experiments in a cell line with a fluorescent protein knock-in and validating % degradation at the protein level by FACS and then carrying out flow-enriched bulk epigenomics assays; and / or (2) performing genomics experiments at the single-cell level through single-cell ATAC-seq, scRNA-seq, or scCUT&TAG. Ideally, the authors would perform both of these sets of experiments to pinpoint the per-cell impact of BRG1 knockdown, especially at dose-dependent enhancers as defined by the authors via bulk ATAC-seq.
- Deeper analysis of enhancer dose-dependency. The authors note that a fraction of enhancers exhibit dose-dependency, but the analysis of what sequence elements guide this dose-dependency remains incomplete. The authors should leverage recent advances in sequence-to-function model training to train computational models (for instance, based on BPNet / ChromBPNet or Enformer) that can distinguish between dose-specific accessibility peaks in their data, and present the resulting models and predictions as a resource for readers to highlight sequence features explaining the data.

- Expanding on the cooperativity between ISWI and SWI/SNF. The double degradation experiments presented in Figure 3 are intriguing, but are too preliminary. Work from multiple groups (see work from Owen-Hughes & Schubeler groups) has already demonstrated that ISWI (not SWI/SNF) is the major driver of CTCF motif accessibility and CTCF binding. Is the phenomenon the authors observe specific to a subset of CTCF sites? Are there sequence elements that determine this 'cooperative' interaction? The authors must explain this more if such a figure is to remain in any revised manuscript.
- Addressing prior literature linking BRG1 dose to PRC1 activity. The well-established balance between polycomb repression and SWI/SNF activity is insufficiently discussed in this manuscript. Work from the Crabtree lab using a combined auxin- and dTAG-degradation of two separate complexes (PMID34117481) showed that the dose of BRG1 is critical for regulating PRC1 repression in mESCs, as derepression of HOX locus genes consequent to BRG1 loss can be rescued by co-depletion of RING1B, the catalytic subunit of the PRC1 complex. The authors must integrate their results with the findings from this published study.
- Moving beyond mESCs. Extension of the study to a cell-type beyond mESCs would tremendously increase impact. The majority of studies of BRG1 have focused on mESCs, and scarce few have extended to the study of cell-fate transitions (that are almost certainly regulated by BRG1 and its paralog BRM). If the authors were able to generate a double-degron system targeting both of these factors, and extend their dose-specific perturbation and genomic analyses to a well-characterized differentiation system, the impact of the study would be significantly improved.

Minor Comments

- Recently published work (PMID37922899) has suggested that the TIP60 remodeler can compensate for BRG1 catalytic inhibition at a subset of regulated sites. This work is not discussed in the manuscript currently and must be addressed in a revised discussion.

Reviewer #3 (Remarks to the Author):

In this manuscript, Hagihara et al. explore the dosage-dependent effects of BRG1 on gene regulatory elements. For this, they use a combination of degron tags, genomic assays and modelling. While BRG1 depletion effects have already been studied in other contexts, the major novelty here comes from assessing BRG1 dosage sensitivity across different regulatory elements and genes. This question is important, given that dosage defects are known to play a role in disease. The study is elegant and well executed with high data quality, and I have only a few points that would strengthen the major message of the manuscript. I would fully support its publication should the authors address the comments below:

1. The manuscript provides convincing evidence of different sensitivity to BRG1 levels for both accessibility (as proxy of activity) and transcription at different types of elements. However, there is some discrepancy between promoter accessibility and transcriptional responses (see point 2). This suggests that while some of the genes may be directly regulated by BRG1 at their TSSs, other genes may be regulated by BRG1-dependent enhancers/insulators. In addition, broader questions would be: How is the dosage response at enhancers related to the transcriptional and accessibility response at their target gene promoters? For instance, do buffered enhancers regulate buffered genes? Could the authors provide additional analyses to estimate this and integrate enhancer accessibility with transcriptional response?

2. A really interesting finding are the types of transcriptional effects elicited by different dosages of BRG1, suggesting that there are genes whose expression is more or less sensitive to its dosage. Thereby the majority of the genes shows a buffered response to BRG1 dosage. However, I notice a discrepancy between the percentage of downregulated genes with a buffered response (85%) and the percentage of promoters dependent on BRG1 for accessibility with a buffered response (71%). If my calculations are correct, also the number of BRG1-accessibility-dependent genes is lower (~775) than the number of downregulated genes (~1000), suggesting that some of the observed transcriptional effects may be secondary. Is there a way to disentangle this? An additional method, such as nascent transcription profiling (PRO-Seq, 4Su-Seq) upon a shorter depletion period could be helpful to rule this out.

3. The authors initially attempted to calibrate ATAC-Seq using drosophila DNA spike-ins, but had to resort to calibrating using top ATAC peaks due to technical problems with drosophila spike-ins variation. This is a valid option in principle, but depends on the assumption that the strongest ATAC-Seq peaks do not globally change in their accessibility. However, given that BRG1 depletion has a strong effect on highly accessible SEs, it is likely that strong ATAC-Seq peaks change their accessibility globally. To rule this out, did the authors use strong ATAC-Seq peaks that do not overlap with BRG1 peaks for calibration? Moreover, since it appears that the authors have not used any calibration for the transcriptional profiling, the transcriptional response (downregulation) may actually be underestimated. How would this affect the reported findings?

Version 1:

Decision Letter:

24th Apr 2025

Dear Professor Zhang,

hope this email finds you well.

Your Article, "Precise modulation of BRG1 levels reveals features of mSWI/SNF dosage sensitivity" has now been seen by 3 referees. You will see from their comments below that Reviewers #2 and #3 are now fully satisfied, however Reviewer #1 has a few remaining points. We are interested in the possibility of publishing your study in Nature Genetics, but would like to consider your response to these final points in the form of a revised manuscript before we make a final decision on publication.

To guide the scope of the revisions, the editors discuss the referee reports in detail within the team, including with the chief editor, with a view to identifying key priorities that should be addressed in revision and sometimes overruling referee requests that are deemed beyond the scope of the current study. In this case, we kindly ask you to address the remaining requests by Reviewer #1. Please do not hesitate to get in touch if you would like to discuss these issues further.

We therefore invite you to revise your manuscript taking into account all reviewer and editor comments. Please highlight all changes in the manuscript text file. At this stage we will need you to upload a copy of the manuscript in MS Word .docx or similar editable format.

*2) If you have not done so already please begin to revise your manuscript so that it conforms to our Article format instructions, available

[here](http://www.nature.com/ng/authors/article_types/index.html).

*3) Include a revised version of any required Reporting Summary: <https://www.nature.com/documents/nr-reporting-summary.pdf>

EXTENDED DATA FIGURES

Link Redacted

We hope to receive your revised manuscript within four to eight weeks. If you cannot send it within this time, please let us know.

Nature Genetics is committed to improving transparency in authorship. As part of our efforts in this direction, we are now requesting that all authors identified as 'corresponding author' on published papers create and link their Open Researcher and Contributor Identifier (ORCID) with their account on the Manuscript Tracking System (MTS), prior to acceptance. ORCID helps the scientific community achieve unambiguous attribution of all scholarly contributions. You can create and link your ORCID from the home page of the MTS by clicking on 'Modify my Springer Nature account'. For more information please visit please visit www.springernature.com/orcid.

Sincerely,
Chiara

Chiara Anania, PhD
Associate Editor
Nature Genetics
<https://orcid.org/0000-0003-1549-4157>

Reviewers' Comments:

Reviewer #1 (Remarks to the Author):

Overall assessment

The manuscript by Hagihara and Zhang entitled "Precise modulation of BRG1 levels reveals features of mSWI/SNF dosage sensitivity" investigates the dependency of BRG1 chromatin binding, chromatin accessibility and transcriptional output on BRG1-dosage. Since their first submission the authors were able to supplement their work by additional evidence, which significantly improved the quality of the manuscript. By confirming their findings in an additional cell line and thereby broadening their applicability, the significance of the results was strengthened. This allowed the authors to suggest potential clinical relevance. Furthermore, they provide additional details on the features of BRG1-dependent and independent promoters and insulators, as well as on the features of buffered and linearly sensitive enhancers, which increases the novelty of their findings.

In summary, this manuscript presents relevant, high-quality data, and the authors have addressed nearly all the key points raised in the initial review. Therefore, I support its publication.

Comments

Related to extended data figure 4c

This analysis demonstrates that BRG1-dependent enhancers lose chromatin accessibility at 2h after 1 μ M BRM014 treatment and that there is only a partial recovery of accessibility at later time points. Furthermore, the authors use these data to demonstrate that the temporal dynamics of G1 and G5 enhancers are similar. However, what is also noticeable in this figure is that also BRG1-independent enhancers lose chromatin accessibility at 2h following treatment, which in contrast to BRG1-dependent enhancers, is completely recovered at 4h after treatment. How do the authors explain it? Perhaps this should be discussed in the text.

The figure legend states that BRG1-dependent and BRG1-independent enhancers were analyzed, while in the figure itself BRG1-dependent and BRG1-independent DARs are indicated. Please refine exactly which regions were included in the analysis.

Related to extended data figure 4e

The increase in binding of RING1B and SUZ12 at BRG1-dependent promoters after BRG1-degradation indicates that loss of accessibility at BRG1-dependent promoters is partially due to PRC mediated repression. While these data suggest that PRC complex-mediated mechanisms are involved in differential chromatin accessibility changes after BRG1-depletion it does not prove a direct causative effect. It would have been interesting to see if PRC-inhibition (e.g. with EZH2 inhibitors) would render the BRG1-dependent promoters BRG1-independent.

Related to figures 5 e-g and extended data figure 9 h-1

The authors demonstrate that upon BRG1 depletion there is a subset of genes which's expression is significantly down-or upregulated. They further show that the transcriptional response to BRG1 dosage is mostly buffered. Through a few handpicked examples, they suggest that transcriptional changes are related to stem cell pluripotency, proliferation and in the context of BEAS-2B cells, tumorigenicity. Perhaps a more systematic approach, such as GO term analysis, would be more suitable to demonstrate the transcriptional effects and to elucidate the cellular effects of BRG1 dosage. In the methods section they write that GO term analysis was performed, why is it not shown?

Furthermore, to strengthen the clinical relevance of their findings, human cancer data sets could be analyzed to demonstrate the correlation between BRG1-loss and increased MYC expression.

Related to extended data figure 7b-c

Alkaline phosphatase staining is not described in the methods section. Please provide the missing information.

Minor comments

Editing suggestions

Line 34: dosage-sensitive effects by precise control of the protein levels of BRG1

Line 50: Depending

Line 61: Since these new approaches overcome the limitations of conventional genetic depletion methods

Line 61-63: please revise sentence for better comprehensibility

Line 70: humans

Line 71: affecting

Line 74: humans

Line 81: In this study we, take advantage

Line 82: and generated BRG1 dTAG knock-in mouse embryonic stem cells (mESCs).

Line 85: suggesting that genome-wide mSWI/SNF
Line 86: We show
Line 86: We further show
Line 91-92: please revise sentence for better comprehensibility
Line 209: analyzed
Line 388: Notably
Line 402: Notably, SEs were preferentially enriched in the G1 group,
Line 426-428: please revise sentence for better comprehensibility
Line 438: showing
Line 439: after mSWI/SNF inhibition, with strong enhancers respond slowly
Line 440: our data indicates that buffered

Reviewer #2 (Remarks to the Author):

I am satisfied with the responses of the authors and have no further comments. This manuscript will be of great interest to the Nature Genetics readership.

Reviewer #3 (Remarks to the Author):

I thank to the authors for their thoughtful response to my questions. I have no further requests.

Version 2:

Decision Letter:

Our ref: NG-A66871R1

9th May 2025

Dear Dr. Zhang,

Thank you for submitting your revised manuscript "Precise modulation of BRG1 levels reveals features of mSWI/SNF dosage sensitivity" (NG-A66871R1). We find that the paper has improved in revision, and therefore we'll be happy in principle to publish it in Nature Genetics, pending minor revisions to comply with our editorial and formatting guidelines.

Thank you again for your interest in Nature Genetics. Please do not hesitate to contact me if you have any questions.

Congratulations!

Sincerely,
Chiara

Chiara Anania, PhD
Associate Editor
Nature Genetics
<https://orcid.org/0000-0003-1549-4157>

Point-by-point Responses to the Reviewers

We thank all the reviewers for their constructive comments. We address their comments point-by-point below.

Reviewer #1 (Remarks to the Author):

The manuscript „Precise modulation of BRG1 levels reveals features of mSWI/SNF dosage sensitivity“ by Hagihara et al. addresses the BRG1-dependent dosage sensitive effects on BRG1 chromatin binding, chromatin accessibility, and transcriptional regulation in mouse embryonic stem cells. This is of clinical relevance, as loss of function mutations – esp. also heterozygous mutations – of members of the SWI/SNF complexes are frequently found in cancer and can result in neurodevelopmental disorders. While this is not a novel question, they address this in a systematic manner using the dTAG system, where precise control of the protein levels are feasible. They combine their NGS approaches with modelling to define if the responses are linear or buffered relative to the protein levels. The observed effects depend on genomic region and feature (binding, remodelling activity, transcription regulation). While some of the observations are not novel, this is a thorough study with very clean and well presented data that address the still poorly understood issue of dosage sensitivity of SWI/SNF members. It may advance the understanding of the effect of SWI/SNF dose on disease, if the results are applicable to more disease-relevant contexts, and is thus of importance.

Response: We thank this reviewer for nicely summarizing our study and for the encouraging comments.

Major comments

1. The authors try to do the analyses in a systematic manner for different genomic features (promoter, enhancer, insulator) and different readouts (binding, chromatin accessibility, transcription regulation). However, this is unfortunately not followed up in all combinations.

a) It would be of high interest to investigate if the transcription factors identified by motif analysis (SP1, NFY) or other chromatin remodellers are responsible for maintaining chromatin accessibility upon BRG1 degradation at BRG1-independent promoters.

Response: We thank this reviewer for raising the question. To address the question, we reanalyzed the ATAC-seq datasets of *Nfya* knockdown in mESCs¹, and found chromatin accessibility is specifically decreased at NFYA-targets among the BRG1-independent promoters. In contrast, no obvious change is detected in the NFYA unbound promoters (Fig. R1). This result, together with the fact that NFYA motif and binding is significantly enriched in BRG1-independent promoter region (Fig. 3a, 3b), indicates that NFYA has a critical role in maintaining the chromatin accessibility of the BRG1-independent promoters. This new result has been added to the revised manuscript as Extended Data Fig. 3f.

Fig. R1. NFYA-KD reduces the accessibility of the NFYA-bound, BRG1-independent promoter. Aggregate coverage plot showing mean ATAC-seq signal at the NFYA-bound (left) and -unbound (right) regions in BRG1-independent promoters.

b) They further only followed up on the gene expression response, measured by RNA-seq, on the genes whose promoter accessibility was dependent or independent of BRG1. As the major regulatory sites of SWI/SNF complexes are enhancers and they observe here a dependency on activity, it would be even more interesting to look at the expression of associated genes. These could be identified e.g. by Hi-ChIP. In addition, RNA-seq might not capture the direct responses, as the total RNA levels depend also on RNA stability and not only on new transcription. It would be of high relevance to investigate how enhancer sensitivity correlates with the transcriptional effect of target genes.

Response: We thank the reviewer for the suggestion. To further analyze the enhancer-mediated transcriptional regulation, we integrated Hi-ChIP of H3K4me3 datasets². Using the loops identified by Hi-ChIP, we linked enhancers to their target genes. We found that genes linked to BRG1-dependent enhancers showed a slight decrease in expression in response to BRG1 depletion, whereas independent enhancers showed no significant alteration on target gene expression (Fig. R2a). However, we observed that 60% of ATAC-seq peaks were located within 20 kb of the transcription start site (TSS) (Fig. R2b, blue), but the Hi-ChIP loops span beyond 20 kb (Fig. R2b, red), suggesting that the enhancer-promoter linkage via chromatin looping was unable to capture the enhancers located in the TSS-proximal regions (<20 kb), as exemplified by the data showing in Fig. R2c. To address this issue, we assigned each enhancer to its nearest promoter based on genomic proximity. Genes with TSSs located within 20 kb of enhancers exhibited a greater decrease in expression (Fig. R2d left) compared to genes located farther from enhancers (>20 kbp) (Fig. R2d, right). However, the transcriptional changes in enhancer-associated genes were smaller than those in BRG1-dependent promoter-associated genes (Fig. R2e), suggesting that promoter regulation plays a more important role in transcriptional control.

While RNA-seq measures the total RNA levels, which are influenced by both transcription and RNA stability, previous time-course studies have demonstrated that RNA-seq can effectively capture transcriptional changes within a short time period, such as 30 minutes to 24 hours³, or 2 hours to 72 hours⁴. These previous studies suggest that at our chosen 24-hour time point, RNA-seq can detect gene expression changes caused by BRG1-dependent regulatory mechanisms. The new results are now presented as Extended Data Fig. 6b, c.

Fig. R2. BRG1-dependent enhancers and promoters in transcriptional regulation. **a**, Boxplot showing dynamics of gene expression associated with BRG1-dependent enhancer (top) and -independent enhancer (bottom). Genes were annotated by H3K4me3 Hi-ChIP data set. **b**, Histogram showing distribution of genomic distance between enhancers to their target gene TSS. Distances derived from H3K4me3 Hi-ChIP loops are shown in red, while distances based on the closest TSS to each enhancer are shown in blue. **c**, Genome browser view showing BRG1 binding and ATAC-seq signals. **d**, Boxplot showing gene expression associated with BRG1-dependent enhancer (top) and -independent enhancer (bottom). Genes are classified by the distance of TSS-enhancer <20 kb (left) and >20 kb (right). Dash line indicates the mean signal at 0 nM condition. *** $p < 0.001$. **e**, Boxplot showing gene expression that is associated with BRG1-dependent promoters (top) and -independent promoters (bottom). Dashed line indicates the mean signal at 0 nM condition. *** $p < 0.001$.

2. The sensitivity of enhancers to SWI/SNF perturbation based on their activity is not entirely new. Schick et al., Nature genetics 2021 has shown that the stronger the enhancers are, the slower was the response to SMARCA4 degradation and only complete perturbation of all SWI/SNF ATPases also perturb super-enhancers. This is consistent with the observation that BRG1 levels also show similar dependence. This similarity between kinetics after BRG1 degradation and BRG1 levels could be discussed.

Response: We thank this reviewer for the suggestion. As suggested, we compared the temporal dynamics of BRG1 inhibition in mESCs⁵ (the Schick study used different cell types). Consistent with their results, we found that chromatin accessibility decreases at 2 hours and partially recovers by 4 hours following

inhibitor treatment (Fig. R3). Notably, despite different characteristics between G1-enriched strong enhancers and G5-enriched weak enhancers, the G1 and G5 temporal dynamics were similar (Fig. R3). This suggests that dosage sensitivity and temporal dynamics are not entirely the same. We have added this comparative analysis as Extended Data Fig. 4c and discussed the potential causes of the differences between the temporal dynamics and sensitivity in the revised manuscript.

Fig. R3. Temporal dynamics of buffered and linearly sensitive enhancers. Aggregate coverage plot showing mean ATAC-seq signal at each enhancer regions after BRM014, BRG1/BRM ATPase inhibitor, treatment for the indicated time.

Further, figure 4e shows nicely the enhancer activity decrease from G1 to G5. However, figures 4d and 4f also show that the G2 to G5 regions can be strong enhancers and super-enhancers. Are there differences between the super enhancers falling into the different groups?

Response: Since the number of super-enhancers (SEs) within each group is relatively small, especially in G2 to G5 (97, 43, 26, 16, and 14 SEs in G1 to G5, respectively), we were unable to perform a reliable chromatin analysis to systematically compare SEs across these groups. The limited sample size restricted our ability to draw statistically conclusions on the potential differences between SEs in different groups.

3. The authors showed linear relationship of BRG1 protein level to co-occupancy of OCT4 and H3K27ac. Is this also the case for sites with low and high OCT4/H3K27ac occupancy?

Response: We thank the reviewer for raising this question. To address the question, we evenly divided the BRG1-binding sites into two groups (low and high) based on the OCT4/H3K27ac ChIP-seq signals, and analyzed BRG1-binding sensitivity in regions with both low and high OCT4/H3K27ac occupancy. We found that BRG1 binding exhibits a consistently linear response to its dosage, regardless of the enrichment levels of H3K27ac or OCT4 (Fig. R4). This further supports our notion that BRG1 occupancy to these sites is not dependent on the binding of TFs or histone modifications.

Fig. R4. BRG1-binding sensitivity is not affected by H3K27ac or OCT4 levels. Histogram of delta values for BRG1 binding at H3K27ac low (1st panel), H3K27ac high (2nd panel), OCT4 low (3rd panel) and OCT4 high (4th panel). The red line indicates delta = 0.

4. The authors conclude on page 9 “This suggests that BRG1 has a preference for promoters and insulators and thus a lower concentration is enough to maintain the accessibility of these genomic elements, while a higher concentration of BRG1 is needed to maintain the accessibility of enhancers.” The data do not explain why these relationships occur, and as alternative explanations are plausible, possible scenarios should be discussed.

Response: We have refined our statement in lines 239–241 as “This indicates that a lower concentration is enough to maintain the accessibility of promoter and insulator, while a higher concentration of BRG1 is needed to maintain the accessibility of enhancers.”.

Additionally, we have expanded our discussion for possible explanations to the differential sensitivities. One potential mechanism involves the compensatory roles of other chromatin remodelers: EP400-TIP60 has been shown to compensate for the loss of mSWI/SNF at promoters⁵, while ISWI plays a major role in maintaining insulator function⁶. These alternative pathways might explain why promoters and insulators are less dependent on high levels of BRG1 for chromatin accessibility. We have incorporated this discussion in lines 455–463.

5. It would be great to additionally or instead of showing representative examples of linear and buffered regions (red and blue dots) to plot the distribution of regions falling into the respective categories.

Response: We have modified the figures to better visualize the distributions of regions falling into the respective categories. We have now included the mean values for each group in new Fig. 6h and Extended Data Fig. 4a, 6e.

6. How do the authors explain the increase in Nanog (and Klf4) upon lower BRG1 levels?

Response: Previous studies have shown that short-term *Brg1* knockdown (72 hours) resulted in *Nanog* upregulation, likely through histone deacetylation at the *Nanog* promoter by HDAC1 in naïve mESCs⁷. However, long-term knockdown results in a decrease in *Nanog* levels⁸. Since we analyzed the effects at 24 hours after acute depletion, our data capture the early, short-term effects of BRG1 loss.

7. The inverse correlation between BRG1 and Myc expression is of high cancer relevance. It would be interesting to look into cancer data, if this holds true beyond mESCs.

Response: We thank the reviewer for raising this important issue. The relationship between BRG1 and *Myc* expression is context-dependent, and the relevance of this inverse correlation to cancer has been explored in several studies. In a BRG1-deficient non-small cell lung cancer (NSCLC) cell line, rescuing BRG1 leads to a decrease in *MYC* expression⁹. Conversely, in another NSCLC line with wild-type BRG1, its knockdown results in *MYC* upregulation¹⁰. Given the frequent mutations in BRG1 observed in lung cancer^{11,12}, and the well-established oncogenic role of *MYC*, we extended our mESC studies to a lung epithelial cell line BEAS-2B to further investigated *MYC* dosage sensitivity.

We manipulated mSWI/SNF dosage by knocking out BRM and degrading BRG1 using AU-15330 (Fig. 6a, Extended Data Fig. 8b). We found that *MYC* expression was upregulated by 1.9-fold when 38% of BRG1 still remaining (new Fig. 6k). Notably, defects in cell proliferation were only observed when mSWI/SNF was completely depleted (new Extended Data Fig. 9i). These findings suggest that a partial decrease in mSWI/SNF dosage can upregulate *MYC* without severely impairing cell viability. These data not only demonstrate the conservation of the BRG1 dosage sensitivity in human and beyond mESCs, but

also suggest that BRG1 dosage decrease can upregulate *MYC* which may initiate oncogenesis in lung cells. Data generated using the lung epithelia cells are now presented in Fig. 6 and Extended Data Figs. 8 and 9, which further emphasize the cancer relevance of BRG1 in regulating *MYC* expression.

Minor comments

8. In the introduction the authors write that esBAF equals cBAF. This is not entirely correct, as this is only the case for embryonic stem cells (ES) and in the original publication, the term esBAF was used for all ES-specific complexes and not only ES-specific cBAF. This should be rewritten or removed.

Response: We thank the reviewer for pointing this out. As suggested, we have removed the term "esBAF" from the introduction.

9. It would be nice to mention for the reader in the text that AU-15330 is a PROTAC targeting BRG1 and the alternative ATPase BRM, though the latter is hardly expressed in mESC.

Response: We thank the reviewer for the suggestion. As recommended, we have updated the text to clarify that AU-15330 is a PROTAC that targets BRG1 and BRM. The updated information can be found in line 218-219.

10. Please indicate for clarity the timepoints when analyses have been done (e.g. 24h after treatment).

Response: We thank the reviewer for the suggestion. To provide greater clarity, we have updated the manuscript to indicate the time points when analyses were performed. For example, we specify that the analyses were conducted 24 hours after treatment, as stated in line 220.

11. Line 310 – intolerant instead of intolerance.

Response: We thank the reviewer for pointing this out, which we have corrected.

12. Lot numbers of antibodies are missing.

Response: We thank the reviewer for pointing this out. We have now included a summary of the antibodies used, along with their lot numbers and other relevant information, in the new Supplemental Table 3.

13. Extended data Fig. 5e requires further description.

Response: We thank the reviewer for this suggestion. We have updated the manuscript to provide a more detailed description of previous Extended Data Fig. 5e (moved to new Extended Data Fig. 7b). The updated description can be found in line 332, and we have also revised the figure legend accordingly in line 1,200-1,203 to ensure the clarity.

14. Reporting summary:

Stating that “two biological replicates are accepted practice in the genomics field” is not fully correct.

Antibodies: lot numbers are missing, it would be nice to indicate for what they were used and how much was used (e.g. it is also not provided in the Material and Method section, how much HA antibody was used for the CUT&RUN), only provider pages were indicated for antibody validations

Response: We thank the reviewer for the helpful comments. We have removed the statement. Regarding the antibodies, we have now provided a table summarizing the information, including lot numbers, intended use, and quantities. This includes the specific amount of HA antibody used for CUT&RUN, which was previously missing from the Materials and Methods section. We also added reference for antibodies in reporting summary.

15. Editorial Policy Checklist:

Where is the “inclusion & ethics” statement?

Response: Thank you for your comment. We appreciate the opportunity to clarify this point. Upon review, we acknowledge that the “Inclusion & Ethics” statement was mistakenly checked in the Editorial Policy Checklist. Since this study was conducted entirely within a single research group and does not involve global or human subject research, an Inclusion & Ethics statement is not applicable in this case. We have now corrected the checklist accordingly and apologize for the oversight.

Reviewer #2 (Remarks to the Author):

In their submission, “Precise modulation of BRG1 levels reveals features of mSWI/SNF dosage sensitivity.” Hagihara, Zhang, and Zhang utilize dTAG degradation technology in mouse embryonic stem cells (mESCs) to explore how altered dosage of BRG1 (the only expressed ATPase subunit of the BAF complex in mESCs) impacts genome-wide chromatin accessibility. From a mixture of genomics assays, including ATAC-seq, CUT&RUN, and RNA-seq, the authors conclude that BRG1 occupancy across the genome is linearly correlated with protein dose, that accessibility is mostly regulated at the level of enhancers (as opposed to promoters or insulators), and that as a result transcriptional consequences are ‘buffered.’ The authors’ approach is interesting – only one published study comes to mind that leverages dTAG technology to systematically measure the consequences of altered factor dosage on genomic phenomena (PMID37024583); however, the study is largely confirmatory given the multiple recent studies combining degradation technology (PMIDs 34117481; 33558760), chemical inhibition (PMIDs 33558757; 37922899), or genetic manipulation (PMID30996347) with genomic readouts to probe regulation by BRG1. As a result, my enthusiasm for the manuscript from a novelty perspective is significantly diminished. Below, I detail some experiments that could render a revised study more impactful (considering the existing literature).

Response: We appreciate this reviewer’s critical comments and thoughtful suggestions. While previous studies have mainly focused on the temporal dynamics of how rapidly BRG1 loss affects chromatin, our study takes a different approach by examining its dosage sensitivity as its role in cancer is dosage sensitive. Although some of our findings, such as the correlation with enhancer activity, may appear to be similar to previous conclusions, our study revealed some novel points with biological implications. Notably, we found no obvious differences in temporal dynamics between buffered and linearly sensitive enhancers (please refer to our response to question 2 from reviewer #1), suggesting that temporal and dosage responses are driven by different mechanisms.

Additionally, we identified NFYA and CTCF motifs are specifically enriched in BRG1-bound, but not in BRG1-independent regulatory elements. We further demonstrated their critical role in maintaining chromatin accessibility of the BRG1-independent genomic elements by analyzing existing datasets.

These findings highlight the roles of TFs and other remodelers in the accessibility of the BRG1-independent regions. Furthermore, following this reviewer's suggestion, we used machine learning model to identify genomic features distinguishing enhancers with different sensitivity (see our response to this reviewer's question 2). Finally, we also extended our study beyond mESCs to a normal human lung epithelial cell line and observed conserved features of BRG1 dosage sensitivity across species and cell type (see our response to this reviewer's question 4). We believe that these additional experiments and analyses confirm and strengthen our initial conclusions and further clarified the mechanisms underlying BRG1 dosage sensitivity.

Major Comments

1. Validation of altered dosage within single cells. The manuscript critically depends on a set of experiments / observations made by Western blot: that altered dTAG compound dosage results in fractional modulation of BRG1 at the protein level. However, Western blot is necessarily based on population-averaging, raising the possibility that certain cells within a population have complete loss of BRG1, and certain cells have higher per-cell levels of the protein (thus resulting in a fractional decrease in signal by Western). The authors must validate their dosing strategy in an orthogonal way. I suggest two possible routes: (1) performing experiments in a cell line with a fluorescent protein knock-in and validating % degradation at the protein level by FACS and then carrying out flow-enriched bulk epigenomics assays; and / or (2) performing genomics experiments at the single-cell level through single-cell ATAC-seq, scRNA-seq, or scCUT&TAG. Ideally, the authors would perform both of these sets of experiments to pinpoint the per-cell impact of BRG1 knockdown, especially at dose-dependent enhancers as defined by the authors via bulk ATAC-seq.

Response: We appreciate the reviewer's valuable suggestion. To address the reviewer's concern, we have performed immunofluorescence staining followed by semi-quantitative analysis to evaluate BRG1 levels in individual cells (Fig. R5a, b), in which BRG1 level was manipulated by dTAG13 in mESCs (a) or by AU-15330 in BEAS-2B cells (b). The results showed that BRG1 levels decrease uniformly across the cells, and we did not observe the presence of distinct subpopulations of cells with different BRG1 degradation efficiency. A similar pattern of protein level changes has also been observed in FACS analysis in a previous study using fluorescence-tagged hSOX9¹³. Thus, the dosage-dependent decrease in BRG1 levels shown in Western blot reflects a population-wide, uniform downregulation rather than a heterogeneous response of individual cells. We have presented the new data in Extended Data Fig. 1f and 8f, g of the revised manuscript.

Fig. R5. The BRG1 level is uniform at single cell level in response to dTAG or AU15330 treatment. a, b, Immunostaining of BRG1 after 24 hours of chemical treatment at the indicated concentrations in mESCs (a) and BEAS-2B cells (b). Relative BRG1 intensity was normalized to that at 0 nM samples (right). Each dot represents a cell.

2. Deeper analysis of enhancer dose-dependency. The authors note that a fraction of enhancers exhibit dose-dependency, but the analysis of what sequence elements guide this dose-dependency remains incomplete. The authors should leverage recent advances in sequence-to-function model training to train computational models (for instance, based on BPNet / ChromBPNet or Enformer) that can distinguish between dose-specific accessibility peaks in their data, and present the resulting models and predictions as a resource for readers to highlight sequence features explaining the data.

Response: We thank the reviewer for this valuable suggestion. To gain deeper insights into the sequence features influencing the dosage-dependent changes in chromatin accessibility, we applied a computational model, ChromBPNet^{14,15}, to predict chromatin accessibility in untreated (0 nM) and treated (100 nM) mESCs. The model achieved a high Pearson's correlation coefficient of $r = 0.84$ across all BRG1-bound peaks (Fig. R6a, b), which reflects its ability to capture chromatin accessibility patterns effectively. We then utilized the TF-MODISCO motif discovery algorithm to identify sequence motifs enriched in buffered and linearly sensitive enhancers and found that SP1 and MAZ motifs were enriched in buffered enhancers, whereas OCT4, SOX2, and KLF4 motifs were enriched in linearly sensitive enhancers (Fig. R6c). Additionally, footprinting analysis by ChromBPNet revealed that OCT4 and SOX2 preferentially bind to linear enhancers, while buffered enhancers had a higher GC content, suggesting that the sequence composition, particularly GC content, may influence dosage sensitivity (Fig. R6d, e). Furthermore, we quantified the number of high-score putative TF binding sites within each enhancer group, and revealed that G1 enhancers had the highest number of high-score sites (2.1 sites per enhancer) compared to the G5 enhancers (1.4 sites per enhancer), and the G1 enhancers retained more sites (0.8 sites) after BRG1 depletion when compared to the G5 enhancers (0.3 sites) (Fig. R6f, g). These findings suggest that motif abundance and sequence composition play critical roles in determining enhancer response to BRG1 depletion. We have presented these results in Extended Data Fig. 5 of the revised manuscript.

Fig. R6. Machine learning model predicts features of buffered and linearly sensitive enhancers. a, Illustration of ChromBPNet predicted chromatin accessibilities at Psmid7 gene locus. Observed track: signals detected by ATAC-seq experiments; Predicted track: ChromBPNet predicted signals; Contribution track: the genome sequence is scaled according to ChromBPNet contribute scores. **b**, Scatter plot showing the correlation between predicted and observed signals. **c**, Motif identified by ChromBPNet models in G1 enhancers (left), and G5 enhancers (right). **d**, Marginal footprints of TF motifs at different groups using predicted profiles from mESC ATAC-seq ChromBPNet models. **e**, Cumulative percentage of GC content in different groups. Wilcoxon rank sum test: G1 vs G2, $p = 0.0003$; G1 vs G3, $p < 0.0001$, G1 vs G4, $p < 0.0001$, G1 vs G5, $p < 0.0001$. **f**, Histograms showing the number of ChromBPNet predicted high score sites from the 0 nM mESCs model or 100 nM dTAG13 mESCs model. **g**, Representative genome browser view of high-score site at buffered or linear enhancers.

3. Expanding on the cooperativity between ISWI and SWI/SNF. The double degradation experiments presented in Figure 3 are intriguing, but are too preliminary. Work from multiple groups (see work from Owen-Hughes & Schubeler groups) has already demonstrated that ISWI (not SWI/SNF) is the major driver of CTCF motif accessibility and CTCF binding. Is the phenomenon the authors observe specific to a subset of CTCF sites? Are there sequence elements that determine this ‘cooperative’ interaction? The authors must explain this more if such a figure is to remain in any revised manuscript.

Response: We thank the reviewer for the comment and thoughtful suggestion. Our results showed that 82% (1836/2236) of BRG1/CTCF-bound insulators were "BRG1-independent," whereas 18% (400/2236) exhibited a "BRG1-dependent (or cooperative)" interaction. This suggests that while ISWI primarily drives CTCF motif accessibility, BRG1 can cooperatively regulate a subset of CTCF sites, indicating a more specialized role of BRG1 in certain regions. Further ChIP-seq and motif enrichment analysis revealed that these BRG1-independent regions are enriched for other TFs binding and motifs, including those of OCT4, NANOG, KLF3, and KLF4 (Fig. R7a, b). These findings suggest that BRG1-dependent (cooperative) regions are occupied by more TFs than the BRG1-independent regions. These results are now presented in Extended Data Fig. 3b and 3h of the revised manuscript.

Fig. R7. Features of BRG1-dependent (cooperative) and -independent (ISWI-dependent) insulators. **a**, Heatmap showing the occupancy of TFs and histone modifications in BRG1-independent and -dependent insulators. **b**, Transcription factor motif enrichment at the BRG1-independent (Indep) and -dependent (Dep) ATAC-seq peaks in insulators. The color scale represents $-\log_{10}(P \text{ value})$, and dot size corresponds to the effect size measured as odds ratio (OR). A two-sided Fisher’s exact test was performed.

4. Addressing prior literature linking BRG1 dose to PRC1 activity. The well-established balance between polycomb repression and SWI/SNF activity is insufficiently discussed in this manuscript. Work from the Crabtree lab using a combined auxin- and dTAG-degradation of two separate complexes (PMID34117481) showed that the dose of BRG1 is critical for regulating PRC1 repression in mESCs, as derepression of HOX locus genes consequent to BRG1 loss can be rescued by co-depletion of RING1B, the catalytic subunit of the PRC1 complex. The authors must integrate their results with the findings from this published study.

Response: We thank the reviewer for pointing this out. To explore the relationship between mSWI/SNF and Polycomb, we analyzed the changes in Polycomb distribution at BRG1-dependent and -independent

regions using data from the referred study¹⁶. We found that binding of the Polycomb proteins are specifically increased in the BRG1-dependent promoters, but not in the BRG1-independent promoters, or enhancers or insulators (Fig. R8a). These results suggest that Polycomb-mediated accessibility regulation is specific to promoters. We also analyzed the effect on Polycomb-target gene expression and found that *Bmi1*, *HoxD11* are significantly upregulated and other *Hox* genes are also tend to be upregulated at 3nM dTAG13 (28% of BRG1) (Fig. R8b) although most of them are not changed before 1 hour (~20%) in the referred study¹⁶. These differences might due to insufficient time (1 hour) for gene expression change, and highlighting the importance of dosage sensitivity study. We also found that upregulation of *Hox* genes is more evident when BRG1 level is decreased, consistent with the previous conclusion that BRG1 directly affects Polycomb distribution. We have included the analyses in Extended Data Fig. 3e, 7c of the revised manuscript.

Fig. R8. Polycomb binding changes at BRG1-dependent and -independent regions and the effect of BRG1 dosage on Polycomb target gene expression. a, Aggregate coverage plot showing mean RING1B (top) and SUZ12 (bottom) signals of 0 hour (red or purple) and 8 hours (cyan or green) at BRG1-dependent and -independent REs. BRG1 was depleted by auxin-degron system for 8 hours¹⁶. **b,** Relative gene expression of *Hox* genes at different BRG1 levels.

5. Moving beyond mESCs. Extension of the study to a cell-type beyond mESCs would tremendously increase impact. The majority of studies of BRG1 have focused on mESCs, and scarce few have extended to the study of cell-fate transitions (that are almost certainly regulated by BRG1 and its paralog BRM). If the authors were able to generate a double-degron system targeting both of these factors, and extend their

dose-specific perturbation and genomic analyses to a well-characterized differentiation system, the impact of the study would be significantly improved.

Response: We thank this reviewer for the thoughtful suggestion. Since BRG1 and BRM have cooperative and antagonistic function¹⁷, and their ATPase activity might be different¹⁸, it is challenging to analyze the effect by manipulating both dosages simultaneously. Also, differentiation model would be difficult because BRG1 is important in differentiation¹⁹ and degradation efficiency might be altered in differentiating cells. Given that approximately 35% of lung cancers have altered mSWI/SNF¹¹, we thought lung cell would be a relevant model to extend our mESC studies. Therefore, we evaluate mSWI/SNF dosage effect in the BEAS-2B, a human normal lung epithelial cell line widely used in lung cancer oncogenic study²⁰⁻²².

Since we have added a new section to the revised manuscript with 3 added Figures (Fig. 6, Extended Data Figs. 8, 9), please go there for details. Briefly, we first knocked out *BRM*, and then manipulated BRG1 dosage by treating with different concentration of AU-15330, a PROTAC chemical highly specific for BRG1 and BRM degradation (Fig. 6a). We found that BRG1 has a more important role in keeping enhancer accessible than that for promoters and insulators (Fig. 6c), which is consistent with what we observed in mESCs. Among the BRG1-dependent regions, promoters and insulators showed buffered response to BRG1 dosage, whereas enhancers exhibited both buffered and linearly sensitivity (Fig. 6g), which correlates with H3K27ac (Fig. 6i), one of the markers of enhancer activity. We also observed super-enhancer enrichment in enhancers with buffered response (Fig. 6j). Finally, we analyzed *MYC* expression at different BRG1 levels and found it is upregulated even when BRG1 is at 40% of its WT level (Fig. 6k). Overall, these observations are consistent with what we observed in mESCs although there is certain degree of difference, indicating the dosage-dependent effect is conserved. Since the mSWI/SNF components are frequently mutated in lung cancers, our new data with normal lung cells shine light on understanding its role in oncogenesis that is associated with mSWI/SNF dosage perturbation due to mutation or epigenetic alteration.

Minor Comments

6. Recently published work (PMID37922899) has suggested that the TIP60 remodeler can compensate for BRG1 catalytic inhibition at a subset of regulated sites. This work is not discussed in the manuscript currently and must be addressed in a revised discussion.

Response: We thank the reviewer for pointing this out. We have now discussed this work in the revised manuscript (line 458-463).

Reviewer #3 (Remarks to the Author):

In this manuscript, Hagihara et al. explore the dosage-dependent effects of BRG1 on gene regulatory elements. For this, they use a combination of degron tags, genomic assays and modelling. While BRG1 depletion effects have already be studied in other contexts, the major novelty here comes from assessing BRG1 dosage sensitivity across different regulatory elements and genes. This question is important, given that dosage defects are known to play a role in disease. The study is elegant and well executed with high data quality, and I have only a few points that would strengthen the major message of the manuscript. I would fully support its publication should the authors address the comments below:

Response: We thank this reviewer for nicely summarizing our study and the compliment.

1. The manuscript provides convincing evidence of different sensitivity to BRG1 levels for both accessibility (as proxy of activity) and transcription at different types of elements. However, there is some discrepancy between promoter accessibility and transcriptional responses (see point 2). This suggests that while some of the genes may be directly regulated by BRG1 at their TSSs, other genes may be regulated by BRG1-dependent enhancers/insulators. In addition, broader questions would be: How is the dosage response at enhancers related to the transcriptional and accessibility response at their target gene promoters? For instance, do buffered enhancers regulate buffered genes? Could the authors provide additional analyses to estimate this and integrate enhancer accessibility with transcriptional response?

Response: We thank the reviewer for raising the question. To assess enhancer-mediated transcriptional regulation, we analyzed genes associated with enhancers using Hi-ChIP data or proximity-based assignment. We observed that genes linked to BRG1-dependent enhancers showed a slight decrease in expression; however, the magnitude of this effect was smaller compared to the genes regulated at the promoter level, suggesting that promoter regulation plays a more important role in transcriptional control. For further details, please refer to our response to question 1b from Reviewer 1 (See Fig. R2).

Furthermore, we examined the effects of G1 (buffered) to G5 (linear) enhancers (Fig. R9). Due to the limited number of associated genes, we were unable to classify genes into strictly buffered or sensitive categories. However, we observed that genes linked to buffered enhancers tend to be less affected, whereas those associated with sensitive enhancers showed more pronounced downregulation at low dTAG13 doses. Collectively, these findings suggest that while enhancers have limited impact on transcription, gene expression changes do reflect enhancer accessibility dynamics. We have included these data in Extended Data Fig. 6b-d of the revised manuscript.

Fig. R9. Buffered and linearly sensitive enhancer’s effect in gene expression. Boxplot showing gene expression association with each of the BRG1-dependent enhancer groups. Dashed line indicates the mean signal at 0 nM condition.

2. A really interesting finding are the types of transcriptional effects elicited by different dosages of BRG1, suggesting that there are genes whose expression is more or less sensitive to its dosage. Thereby the majority of the genes shows a buffered response to BRG1 dosage. However, I notice a discrepancy between the percentage of downregulated genes with a buffered response (85%) and the percentage of promoters dependent on BRG1 for accessibility with a buffered response (71%). If my calculations are correct, also the number of BRG1-accessibility-dependent genes is lower (~775) than the number of downregulated genes (~1000), suggesting that some of the observed transcriptional effects may be secondary. Is there a way to disentangle this? An additional method, such as nascent transcription profiling (PRO-Seq, 4Su-Seq) upon a shorter depletion period could be helpful to rule this out.

Response: We thank the reviewer for this insightful comment. The percentage difference arises because we focused on BRG1-bound regions for accessibility changes, whereas transcriptional analysis included all genes due to the difficulty in defining the direct targets. Nevertheless, the discrepancy between ATAC-seq and mRNA-seq data may reflect the biological compensation mechanisms that modulate transcriptional output despite chromatin state changes. For example, we observed all the BRG1 binding were abolished after BRG1 depletion, but only one-third of them showed reduced chromatin accessibilities upon BRG1 depletion, suggesting that other factors may compensate for the loss of BRG1. In fact, previous studies have demonstrated the compensation effect in promoter accessibility and gene expression by EP400-TIP60 or redistribution of Polycomb complexes can mitigate the transcriptional effect of mSWI/SNF loss^{5,16}. These regulatory changes occur within a few hours. However, we were unable to perform earlier time-point analyses because we aimed to assess dosage-sensitive effects while minimizing confounding time-dependent changes³⁻⁵. Furthermore, the number of differentially expressed genes (DEGs) depends on the cut-off settings (Fig. R10), making direct numerical comparisons difficult. Nevertheless, we observed significant gene expression changes at BRG1-associated promoters and enhancers (Fig. R9), but not those not associated with BRG1.

Fig. R10. The number of DEGs at different dTAG13 concentrations. Left: fold change >1.5; Right: fold change >2.

3. The authors initially attempted to calibrate ATAC-Seq using drosophila DNA spike-ins, but had to resort to calibrating using top ATAC peaks due to technical problems with drosophila spike-ins variation. This is a valid option in principle, but depends on the assumption that the strongest ATAC-Seq peaks do not globally change in their accessibility. However, given that BRG1 depletion has a strong effect on highly accessible SEs, it is likely that strong ATAC-Seq peaks change their accessibility globally. To rule this out, did the authors use strong ATAC-Seq peaks that do not overlap with BRG1 peaks for calibration? Moreover, since it appears that the authors have not used any calibration for the transcriptional profiling, the transcriptional response (downregulation) may actually be underestimated. How would this affect the reported findings?

Response: We apologize for the confusion. To minimize the effect of BRG1 depletion, we calibrated ATAC-seq using peaks that do not overlap with BRG1-bound regions. Among the 107,203 ATAC-seq peaks, only 20,323 (~19%) overlapped with BRG1-binding sites. As shown in a representative region (Fig. R11a), we observed stable ATAC signals in BRG1-unbound and unaffected regions.

For the RNA-seq analysis, since the chromatin accessibility at promoters remained largely unchanged (suggesting the absence of a global transcriptional alteration), we only calibrated sequencing coverage. To further validate our findings and rule out the possibility of underestimating transcriptional downregulation, we performed RT-qPCR on selected target genes. The RT-qPCR results were consistent with the RNA-seq results (Fig. R11b), showing a similar trend for the downregulated genes (*Tbx3*, *Tcl1*) and upregulated genes (*Nanog*, *Klf4*, *Myc*). Given that BRG1 depletion affects a limited number of genes,

and that RT-qPCR validation confirmed the trends observed in RNA-seq, our current analysis should be able to capture the transcriptional changes.

Fig. R11. Example of BRG1-unbound and unaffected regions and validation of gene expression by RT-qPCR. **a**, Representative genome browser view of BRG1 CUN&RUN, ATAC-seq signals. Highlighted regions show the accessibilities of BRG1-unbound regions. **b**, Gene expression of representative genes analyzed by RNA-seq (top) and RT-qPCR (bottom). Relative expression to *Gapdh* was calculated in RT-qPCR, both RNA-seq and RT-qPCR were normalized to 0 nM conditions. The red dotted line indicate the expression level at 0 nM.

Finally, we would like to thank all the reviewers for their insightful comments. Addressing their questions has helped us greatly improved our manuscript.

Sincerely,

Yi Zhang, Ph.D
Investigator, HHMI
Professor, BCH & HMS

References

1. Oldfield, A. J. *et al.* NF-Y controls fidelity of transcription initiation at gene promoters through maintenance of the nucleosome-depleted region. *Nat Commun* **10**, (2019).
2. Crispatzu, G. *et al.* The chromatin, topological and regulatory properties of pluripotency-associated poised enhancers are conserved in vivo. *Nat Commun* **12**, (2021).
3. Iurlaro, M. *et al.* Mammalian SWI/SNF continuously restores local accessibility to chromatin. *Nat Genet* **53**, 279–287 (2021).
4. Schick, S. *et al.* Acute BAF perturbation causes immediate changes in chromatin accessibility. *Nat Genet* **53**, 269–278 (2021).
5. Martin, B. J. E. *et al.* Global identification of SWI/SNF targets reveals compensation by EP400. *Cell* **186**, 5290-5307.e26 (2023).
6. Barisic, D., Stadler, M. B., Iurlaro, M. & Schübeler, D. Mammalian ISWI and SWI/SNF selectively mediate binding of distinct transcription factors. *Nature* **569**, 136–140 (2019).
7. Carey, T. S. *et al.* BRG1 Governs Nanog Transcription in Early Mouse Embryos and Embryonic Stem Cells via Antagonism of Histone H3 Lysine 9/14 Acetylation. *Mol Cell Biol* **35**, 4158–4169 (2015).
8. Ho, L. *et al.* An embryonic stem cell chromatin remodeling complex, esBAF, is essential for embryonic stem cell self-renewal and pluripotency. *Proc Natl Acad Sci U S A* **106**, 5181–5186 (2009).
9. Romero, O. A. *et al.* The tumour suppressor and chromatin-remodelling factor BRG1 antagonizes Myc activity and promotes cell differentiation in human cancer. *EMBO Mol Med* **4**, 603–616 (2012).
10. Orvis, T. *et al.* BRG1/SMARCA4 inactivation promotes non-small cell lung cancer aggressiveness by altering chromatin organization. *Cancer Res* **74**, 6486–6498 (2014).
11. Mittal, P. & Roberts, C. W. M. The SWI/SNF complex in cancer — biology, biomarkers and therapy. *Nature Reviews Clinical Oncology* vol. 17 435–448 Preprint at <https://doi.org/10.1038/s41571-020-0357-3> (2020).
12. Kadoch, C. *et al.* Proteomic and bioinformatic analysis of mammalian SWI/SNF complexes identifies extensive roles in human malignancy. *Nat Genet* **45**, 592–601 (2013).
13. Naqvi, S. *et al.* Precise modulation of transcription factor levels identifies features underlying dosage sensitivity. *Nature Genetics* **2023** 55:5 **55**, 841–851 (2023).
14. Pampari, A. *et al.* ChromBPNet: bias factorized, base-resolution deep learning models of chromatin accessibility reveal cis-regulatory sequence syntax, transcription factor footprints and regulatory variants. Preprint at <https://doi.org/10.1101/2024.12.25.630221> (2024).
15. Pratt, H. E. *et al.* *Using a Comprehensive Atlas and Predictive Models to Reveal the Complexity and Evolution of Brain-Active Regulatory Elements*. <https://www.science.org> (2024).
16. Weber, C. M. *et al.* mSWI/SNF promotes Polycomb repression both directly and through genome-wide redistribution. *Nat Struct Mol Biol* **28**, 501–511 (2021).
17. Raab, J. R., Runge, J. S., Spear, C. C. & Magnuson, T. Co-regulation of transcription by BRG1 and BRM, two mutually exclusive SWI/SNF ATPase subunits. *Epigenetics Chromatin* **10**, (2017).
18. Jancewicz, I., Siedlecki, J. A., Sarnowski, T. J. & Sarnowska, E. BRM: The core ATPase subunit of SWI/SNF chromatin-remodelling complex - A tumour suppressor or tumour-

promoting factor? *Epigenetics and Chromatin* vol. 12 Preprint at <https://doi.org/10.1186/s13072-019-0315-4> (2019).

19. Alexander, J. M. *et al.* Brg1 modulates enhancer activation in mesoderm lineage commitment. *Development (Cambridge)* **142**, 1418–1430 (2015).
20. Zhong, M. *et al.* Malignant Transformation of Human Bronchial Epithelial Cells Induced by Arsenic through STAT3/miR-301a/SMAD4 Loop. *Sci Rep* **8**, (2018).
21. Park, Y. hee, Kim, D., Dai, J. & Zhang, Z. Human bronchial epithelial BEAS-2B cells, an appropriate in vitro model to study heavy metals induced carcinogenesis. *Toxicol Appl Pharmacol* **287**, 240–245 (2015).
22. Liu, Y. *et al.* Exosomes of A549 Cells Induced Migration, Invasion, and EMT of BEAS-2B Cells Related to let-7c-5p and miR-181b-5p. *Front Endocrinol (Lausanne)* **13**, (2022).

Point-by-point Responses to the Reviewers

We thank all the reviewers for their constructive comments. We address their comments point-by-point below.

Reviewer #1 (Remarks to the Author):

Overall assessment

The manuscript by Hagihara and Zhang entitled “Precise modulation of BRG1 levels reveals features of mSWI/SNF dosage sensitivity” investigates the dependency of BRG1 chromatin binding, chromatin accessibility and transcriptional output on BRG1-dosage. Since their first submission the authors were able to supplement their work by additional evidence, which significantly improved the quality of the manuscript. By confirming their findings in an additional cell line and thereby broadening their applicability, the significance of the results was strengthened. This allowed the authors to suggest potential clinical relevance. Furthermore, they provide additional details on the features of BRG1-dependent and independent promoters and insulators, as well as on the features of buffered and linearly sensitive enhancers, which increases the novelty of their findings.

In summary, this manuscript presents relevant, high-quality data, and the authors have addressed nearly all the key points raised in the initial review. Therefore, I support its publication.

Response: We thank the reviewer for the positive and constructive feedback. We appreciate this reviewer’s statement that “In summary, this manuscript presents relevant, high-quality data, and the authors have addressed nearly all the key points raised in the initial review. Therefore, I support its publication.”. We address his/her reminding comments below:

Related to extended data figure 4c

This analysis demonstrates that BRG1-dependent enhancers lose chromatin accessibility at 2h after 1 uM BRM014 treatment and that there is only a partial recovery of accessibility at later time points. Furthermore, the authors use these data to demonstrate that the temporal dynamics of G1 and G5 enhancers are similar. However, what is also noticeable in this figure is that also BRG1-independent enhancers lose chromatin accessibility at 2h following treatment, which in contrast to Brg1-dependent enhancers, is completely recovered at 4h after treatment. How do the authors explain it? Perhaps this should be discussed in the text.

Response: In the original temporal dynamics analysis upon BRG1/BRM inhibitor¹, they identified enrichment of OCT4, SOX2 and NANOG in non-recovered enhancers and enrichment of CTCF and SNF2H in recovered enhancers. Despite the definition of enhancers is different between that study and our study (they classified all ATAC-seq peaks into promoter and enhancer, while we classified BRG1-bound ATAC-seq peaks into promoter, enhancer and insulator), we also observed TFs enrichment in BRG1-dependent enhancer and CTCF enrichment in BRG1-independent insulator as shown in Fig. 3a and ExFig. 3b, c. Since these observations align with the difference of recovered and non-recovered enhancer’s features, and since the conclusions have already been discussed in the previous study, we do not feel we need to emphasize the same statement in our manuscript.

The figure legend states that BRG1-dependent and BRG1-independent enhancers were analyzed, while in the figure itself BRG1-dependent and BRG1-independent DARs are indicated. Please refine exactly which regions were included in the analysis.

Response: We appreciate the reviewer for pointing out this. We have stated now that the analysis was performed on BRG1-dependent and -independent enhancers, and have modified the figure accordingly for consistency.

Related to extended data figure 4e

The increase in binding of RING1B and SUZ12 at BRG1-dependent promoters after BRG1-degradation indicates that loss of accessibility at BRG1-dependent promoters is partially due to PRC mediated repression. While these data suggest that PRC complex-mediated mechanisms are involved in differential chromatin accessibility changes after BRG1-depletion it does not prove a direct causative effect. It would have been interesting to see if PRC-inhibition (e.g. with EZH2 inhibitors) would render the BRG1-dependent promoters BRG1-independent.

Response: We thank the reviewer for the insightful comment. While we observed increased binding of Polycomb proteins (RING1B and SUZ12) at BRG1-dependent promoters following BRG1 depletion, previous studies have shown that Polycomb proteins themselves do not directly alter chromatin accessibility^{2,3}. This suggests that the observed chromatin accessibility changes are not driven directly by Polycomb-mediated repression. To make this clear and to avoid potential misinterpretation, we have revised the result section (lines 206–207) accordingly.

Related to figures 5 e-g and extended data figure 9 h-1

The authors demonstrate that upon BRG1 depletion there is a subset of genes which's expression is significantly down-or upregulated. They further show that the transcriptional response to BRG1 dosage is mostly buffered. Through a few handpicked examples, they suggest that transcriptional changes are related to stem cell pluripotency, proliferation and in the context of BEAS-2B cells, tumorigenicity. Perhaps a more systematic approach, such as GO term analysis, would be more suitable to demonstrate the transcriptional effects and to elucidate the cellular effects of BRG1 dosage. In the methods section they write that GO term analysis was performed, why is it not shown?

Response: We thank the reviewer for this valuable suggestion. As the number of upregulated genes was limited, we performed Gene Ontology (GO) analysis on the downregulated genes in mESCs. This analysis revealed significant enrichment of gene sets related to cell cycle and MAPK signaling pathway, both of which are critical for maintaining self-renewal and proper pluripotency in mESCs. These results are consistent with our phenotypic observations of impaired proliferation and defects in pluripotency maintenance during long-term culture (Extended Data Fig. 7a, b). The GO analysis is now shown in Extended Data Fig. 6f.

In the case of BEAS-2B cells, our analysis was limited to *MYC* and *KRAS* expression using RT-qPCR, and we did not perform global transcriptomic analysis. Therefore, GO term analysis could not be conducted for this cell type.

Furthermore, to strengthen the clinical relevance of their findings, human cancer data sets could be analyzed to demonstrate the correlation between BRG1-loss and increased MYC expression.

Response: We thank the reviewer for this important suggestion. To address the question, we analyzed publicly available transcriptomic and genomic data from the DepMap database⁴, focusing on human cancer cell lines from lung and hematopoietic lineages. We observed a negative correlation between *BRG1* and *MYC* expression levels in lung cancer cell lines particularly in those without *MYC* amplification (green line, $R_p = -0.39$) (Fig. R1a, left). Furthermore, *BRG1* and/or *BRM* mutant cell lines

exhibited significantly elevated *MYC* expression levels (Fig. 1b, left). On the other hand, *BRG1* and *MYC* expression levels were positively correlated in hematopoietic lineage, consistent with previous studies (Fig. R1a, b, right)⁵⁻⁷. Additionally, *MYC* upregulation was observed following *BRG1* knockdown in non-small cell lung cancer (NSCLC) cell line (Fig. R1c)⁸. Conversely, *BRG1* restoration in *BRG1*-deficient NSCLC results in *MYC* downregulation⁹. Overall, the correlation and genetic manipulations support the notion that *BRG1* dosage may contribute to *MYC* expression and highlight the potential relevance of this regulatory axis in human cancers, especially in lung cancer.

We choose not to show this data in the final manuscript because we feel the correlation, although statistically significant, is not high enough to make a big statement. However, if the reviewer and the editor think we think this data worth to be shown in our manuscript, we can add it to the supplemental data when we prepare for typesetting.

Fig. R1. Negative correlation of *BRG1* and *MYC* expression in human lung cancer cell lines. **a**, Scatter plots showing the expression levels of *BRG1* (x-axis) and *MYC* (y-axis) in lung (left) and hematopoietic (right) cancer cell lines. Expression and mutation data were obtained from the Cancer Dependency Map (DepMap) Public 24Q4 datasets (DepMap, 2024; <https://depmap.org/portal>). Each dot represents an individual cell line, and the lines indicate linear regression. *MYC* amplification was defined as a copy number relative to ploidy greater than 1.3. *BRG1* mutation include both damaging and hotspot variants. **b**, Boxplot showing *MYC* expression level in the presence or absence of *BRG1* and/or *BRM* mutations. Cell lines with *MYC* amplification were excluded from the analysis. **c**, Box plot showing *MYC* expression following *BRG1*-knockdown in NSCLC cell lines⁸. $**p < 0.01$; $***p < 0.001$.

Related to extended data figure 7b-c

Alkaline phosphatase staining is not described in the methods section. Please provide the missing information.

Response: We apologize for the omission. The method for alkaline phosphatase staining has now been added to the Methods section.

Minor comments

Editing suggestions

Line 34: dosage-sensitive effects by precise control of the protein levels of *BRG1*

Line 50: Depending

Line 61: Since these new approaches overcome the limitations of conventional genetic depletion methods

Line 61-63: please revise sentence for better comprehensibility

Line 70: humans

Line 71: affecting
Line 74: humans
Line 81: In this study we, take advantage
Line 82: and generated BRG1 dTAG knock-in mouse embryonic stem cells (mESCs).
Line 85: suggesting that genome-wide mSWI/SNF
Line 86: We show
Line 86: We further show
Line 91-92: please revise sentence for better comprehensibility
Line 209: analyzed
Line 388: Notably
Line 402: Notably, SEs were preferentially enriched in the G1 group,
Line 426-428: please revise sentence for better comprehensibility
Line 438: showing
Line 439: after mSWI/SNF inhibition, with strong enhancers respond slowly
Line 440: our data indicates that buffered

Response: We thank this reviewer for these helpful editing suggestions. We have now made the suggested changes accordingly (marked in red in the text).

Reviewer #2 (Remarks to the Author):

I am satisfied with the responses of the authors and have no further comments. This manuscript will be of great interest to the Nature Genetics readership.

Reviewer #3 (Remarks to the Author):

I thank to the authors for their thoughtful response to my questions. I have no further requests.

Response: We thank both reviewers for their positive comments on our revision.

Finally, we would like to thank all the reviewers for their insightful comments. Addressing their questions has helped us greatly improved our manuscript.

Sincerely,

Yi Zhang, Ph.D
Investigator, HHMI
Professor, BCH & HMS